# Hamiltonian Descent Algorithms for Optimization: Accelerated Rates via Randomized Integration Time

**Qiang Fu**
Department of Computer Science
Yale University
qiang.fu@yale.edu

**Andre Wibisono**
Department of Computer Science
Yale University
andre.wibisono@yale.edu

## Abstract

We study the *Hamiltonian flow for optimization* (HF-opt), which simulates the Hamiltonian dynamics for some integration time and resets the velocity to $0$ to decrease the objective function; this is the optimization analogue of the Hamiltonian Monte Carlo algorithm for sampling. For short integration time, HF-opt has the same convergence rates as gradient descent for minimizing strongly and weakly convex functions. We show that by randomizing the integration time in HF-opt, the resulting *randomized Hamiltonian flow* (RHF) achieves accelerated convergence rates in continuous time, similar to the rates for accelerated gradient flow. We study a discrete-time implementation of RHF as the *randomized Hamiltonian gradient descent* (RHGD) algorithm. We prove that RHGD achieves the same accelerated convergence rates as Nesterov's accelerated gradient descent (AGD) for minimizing smooth strongly and weakly convex functions. We provide numerical experiments to demonstrate that RHGD is competitive with classical accelerated methods such as AGD across all settings and outperforms them in certain regimes.

## 1 Introduction

Optimization plays a central role in machine learning, with algorithms such as gradient descent (GD) and accelerated gradient descent (AGD) [Nesterov, 1983] serving as essential tools for optimizing objective functions. A growing body of work has explored optimization algorithms in the framework of continuous-time dynamical systems, which provides insights into algorithmic behaviors and convergence properties. In this paper, we develop a novel family of accelerated optimization algorithms that are designed based on the Hamiltonian flow.

Hamiltonian flow (HF) originates from classical mechanics, describing the continuous-time evolution of physical systems. At a fundamental level, Hamiltonian flow governs how the positions and momenta of moving bodies evolve while conserving energy. Beyond its roots in physics, Hamiltonian flows have inspired computational algorithms such as Hamiltonian Monte Carlo (HMC) [Duane et al., 1987], a classical method widely employed for sampling from complex, high-dimensional probability distributions [Neal et al., 2011, Betancourt, 2017, Hoffman et al., 2014]. Due to its effectiveness, HMC has found extensive applications in Bayesian inference, statistical physics, and machine learning [Gelman et al., 1995, Kruschke, 2014, Lelievre and Stoltz, 2016].

There has been growing interest in exploring the connections between optimization and sampling, as they share deep theoretical foundations and many algorithmic similarities. Notably, the Langevin dynamics for sampling can be viewed as the gradient flow for minimizing the relative entropy or Kullback–Leibler (KL) divergence in the space of probability distributions [Jordan et al., 1998]. Many works build on this perspective to use techniques from optimization to analyze Langevin-based algorithms [Wibisono, 2018, Bernton, 2018, Durmus et al., 2019, Ma et al., 2021] or develop novel sampling algorithms inspired by optimization [Salim et al., 2020, Lee et al., 2021, Lambert et al.,

39th Conference on Neural Information Processing Systems (NeurIPS 2025).

2022, Chen et al., 2022, Diao et al., 2023, Das and Nagaraj, 2023, Suzuki et al., 2023, Fu and Wilson, 2024, Chen et al., 2025]. In this paper, we strengthen the links between optimization and sampling in the opposite direction, by studying how to translate the Hamiltonian Monte Carlo (HMC), a classical sampling algorithm, to design new optimization algorithms, particularly for accelerated methods.

In the links between optimization and sampling, there is a significant theoretical gap regarding *acceleration*. In optimization, it is well known that greedy methods such as GD can be outperformed by accelerated methods such as AGD [Nesterov, 1983] which have faster and optimal convergence rates with square-root dependence on the condition number for minimizing smooth and strongly convex functions under the standard first-order oracle model; see Appendix C.1 for a review. A similar acceleration phenomenon in sampling is still elusive, but there are some promising candidates. The underdamped (or kinetic) Langevin dynamics has an accelerated convergence rate in continuous time [Cao et al., 2023], but in discrete time, algorithms based on its discretization still do not have the desired accelerated rates [Ma et al., 2021, Zhang et al., 2023]. Another candidate is the *randomized* Hamiltonian Monte Carlo (RHMC) [Bou-Rabee and Sanz-Serna, 2017], which is obtained by randomizing the integration time in HMC, and has been conjectured to have an accelerated convergence rate for sampling [Jiang, 2023]. In continuous time, the idealized RHMC indeed has an accelerated convergence rate in $\chi^2$-divergence for log-concave target distributions [Lu and Wang, 2022]. On the algorithmic side, recent works have shown that for a Gaussian target distribution, HMC with carefully chosen integration time, either determined by the roots of Chebyshev polynomials or randomly drawn from exponential distributions, indeed achieves an accelerated mixing time with square-root dependence on the condition number [Wang and Wibisono, 2023a, Jiang, 2023, Apers et al., 2024]. However, the proof that RHMC achieves acceleration in discrete time for a general target distribution remains missing. In this work, we study the optimization analogue of this question, by designing a new accelerated optimization algorithm based on the randomized Hamiltonian flow.

While Hamiltonian flows have found great success in sampling, their direct use in designing optimization algorithms is still relatively limited. Most existing Hamiltonian-based optimization methods can be seen as the discretization of the accelerated gradient flow (AGF) [Su et al., 2016, Wibisono et al., 2016], which is the combination of HF with a damping term for dissipating energy; this is different from the HF we study in this work which does not have damping. We provide additional discussion of related work in Appendix A. Pure HF without damping terms have rarely been studied for optimization. In fact, it obeys the law of energy conservation rather than dissipation, which is opposite to optimization tasks. Notable prior works include Teel et al. [2019], Diakonikolas and Jordan [2021], De Luca et al. [2023], Wang [2024]. Particularly, Wang [2024] show that Hamiltonian flows with velocity refreshment and Chebyshev-based integration times can achieve accelerated convergence for strongly convex quadratic functions, which is comparable to AGF with refreshment. Beyond quadratic functions, we demonstrate that HF with short-time integration and periodic velocity refreshment achieves the same non-accelerated convergence rates as GD up to constants (see Theorem 1). This naturally prompts a question:

> *Can we develop accelerated optimization methods based on the Hamiltonian flow?*

In this work, we answer this question affirmatively and demonstrate that HF with randomized integration time and its discretization yield accelerated convergence rates for minimizing strongly and weakly convex functions. Our principal contributions are:

- We propose the *randomized Hamiltonian flow* (RHF) for optimization as an analogue of RHMC. We establish its accelerated convergence rates of $O(\exp(-\sqrt{\alpha/5}\,t))$ for $\alpha$-strongly convex functions and $O(1/t^2)$ for weakly convex functions. These rates match the optimal accelerated convergence rates of AGF [Su et al., 2016, Wibisono et al., 2016] up to constants.

- We study the *randomized Hamiltonian gradient descent* (RHGD) which is a discretization of RHF. Under $L$-smoothness assumption, RHGD achieves the overall iteration complexity of $\tilde{O}(\sqrt{L/\alpha})$ for $\alpha$-strongly convex functions and $O(\sqrt{L/\varepsilon})$ for weakly convex functions to generate an $\varepsilon$-accurate solution in expectation, matching the optimal accelerated rates of AGD [Nesterov, 1983, 2018] and its randomized variant [Even et al., 2021].

**Organization**  The remainder of this work is organized as follows. Section 2 presents notations, definitions and a review of the Hamiltonian flow (HF) for designing optimization algorithms. Section 3 proposes the definition of the randomized Hamiltonian flow (RHF) and its accelerated convergence

rates. Section 4 describes how to discretize continuous-time RHF into an implementable optimization algorithm RHGD and establishes its accelerated convergence rates. Section 5 presents numerical experiments validating the effectiveness of our proposed methods. Section 6 concludes the paper and discusses its limitations.

## 2 Preliminaries and reviews

### 2.1 Notations and definitions

Let $\|\cdot\| := \|\cdot\|_2$ denote the Euclidean norm on the $d$-dimensional Euclidean space $\mathbb{R}^d$. A differentiable function $f : \mathbb{R}^d \to \mathbb{R}$ is $\alpha$-strongly convex if $f(y) \geq f(x) + \langle \nabla f(x), y - x \rangle + \frac{\alpha}{2}\|y - x\|^2$ for any $x, y \in \mathbb{R}^d$, where $f$ is (weakly) convex if $\alpha = 0$. We say $f$ is $\alpha$-gradient-dominated if $\|\nabla f(x)\|^2 \geq 2\alpha(f(x) - f(x^*))$ for any $x \in \mathbb{R}^d$ where $x^* = \arg\min_{x \in \mathbb{R}^d} f(x)$ is a minimizer of $f$. We say $f$ is $L$-smooth if $f(y) \leq f(x) + \langle \nabla f(x), y - x \rangle + \frac{L}{2}\|y - x\|^2$ or equivalently $\|\nabla f(x) - \nabla f(y)\| \leq L\|x - y\|$ for any $x, y \in \mathbb{R}^d$. We assume $\alpha \leq 1 \leq L$. We say $\hat{x}$ is an $\varepsilon$-accurate solution in expectation if $\mathbb{E}[f(\hat{x}) - f(x^*)] \leq \varepsilon$. Let $\kappa := L/\alpha$ denote the condition number. We use $\dot{g}_t := \frac{d}{dt}g_t$ to denote the time derivative of a time-dependent quantity $g_t$. We define the flow map $\mathsf{HF}_\eta : \mathbb{R}^d \times \mathbb{R}^d \to \mathbb{R}^d \times \mathbb{R}^d$ as $\mathsf{HF}_\eta(X_0, Y_0) = (X_\eta, Y_\eta)$, which is the solution to Hamiltonian flow (HF) at time $\eta$ starting from $(X_0, Y_0)$. Given a time-dependent random variable $Z_t$, we use $\rho_t^Z$ to denote its probability distribution. We identify probability distributions with their density functions. Let $\mathrm{Exp}(\gamma)$ denote the exponential distribution with mean $1/\gamma$ for $\gamma > 0$. Let $[n] := \{1, 2, ..., n\}$. We use $a = O(b)$ to denote $a \leq Cb$ for constant $C > 0$ and use $a = \tilde{O}(b)$ to denote $a = O(b)$ up to logarithmic factors. We use $a = \Theta(b)$ to denote $a = Cb$ for constants $C > 0$.

**Problem setting.** Our goal is to solve the following optimization problem:

$$\min_{x \in \mathbb{R}^d} f(x), \tag{1}$$

where $f : \mathbb{R}^d \to \mathbb{R}$ is a differentiable function, and $x^* = \arg\min_{x \in \mathbb{R}^d} f(x)$ is a minimizer of $f$.

### 2.2 Hamiltonian flow for optimization

The *Hamiltonian flow* (HF) is a system of ordinary differential equations for $(X_t, Y_t) \in \mathbb{R}^d \times \mathbb{R}^d$:

$$\dot{X}_t = Y_t, \quad \dot{Y}_t = -\nabla f(X_t). \tag{HF}$$

Define the *energy* (or Hamiltonian) function $H(x, y) := f(x) + \frac{1}{2}\|y\|^2$. A fundamental property of the Hamiltonian flow (HF) is that it conserves energy; see Appendix B.1 for the proof.

**Lemma 1** (Energy Conservation). *Along* (HF)*, $H(X_t, Y_t) = H(X_0, Y_0)$ for all $t \geq 0$.*

We can exploit the conservation property of the Hamiltonian flow to design an optimization algorithm by periodically refreshing the velocity to 0. This idea results in the following **Hamiltonian flow for optimization** (HF-opt) algorithm, which was also proposed and studied by Teel et al. [2019], Wang [2024]. Below, $\Pi_1(x, y) = x$ is the projection operator to the first component.

---

**Algorithm 1 Hamiltonian Flow for Optimization** (HF-opt)

---

1: Initialize $x_0 \in \mathbb{R}^d$. Choose integration time $\eta_k > 0$ for $k \geq 0$.
2: **for** $k = 0, 1, \ldots, K - 1$ **do**
3: $\quad x_{k+1} = \Pi_1 \circ \mathsf{HF}_{\eta_k}(x_k, 0)$ $\qquad \triangleright$ (evolve (HF) for time $\eta_k$ and project to first component)
4: **end for**
5: **return** $x_K$

---

Lemma 1 implies the following descent lemma of HF-opt; see Appendix B.1 for the proof.

**Lemma 2.** *For any $k$ and $\eta_k > 0$, HF-opt (Algorithm 1) satisfies $f(x_{k+1}) \leq f(x_k)$.*

HF-opt is an instance of a new optimization principle, the *"Lift-Conserve-Project" (LCP) scheme*, which is the same principle that underlies HMC for sampling; see Appendix B for more details.

HF-opt is an idealized algorithm since it assumes we can solve the Hamiltonian flow (HF) exactly. We study its convergence properties in this and the next sections, and we study how to implement it as a concrete discrete-time algorithm in Section 4. When $f$ is smooth, we show that HF-opt with short integration time $\eta_k$ has the following convergence rates under gradient domination and weak convexity in Theorem 1. Note that the first conclusion in Theorem 1 also holds under $\alpha$-strong convexity since it implies $\alpha$-gradient domination. We provide the proof in Appendix D.1.

**Theorem 1.** *Assume $f$ is $L$-smooth. Along Algorithm 1 with $\eta_k = h \leq \frac{1}{\sqrt{L}}$, from any $x_0 \in \mathbb{R}^d$:*

1. *If $f$ is $\alpha$-gradient-dominated, then $f(x_k) - f(x^*) \leq \left(1 - \frac{1}{2}\alpha h^2\right)^k (f(x_0) - f(x^*))$.*

2. *If $f$ is weakly convex, then $f(x_k) - f(x^*) \leq \dfrac{34\|x_0 - x^*\|^2}{h^2 k}$.*

Up to constants, the results in Theorems 1 match the convergence rates of GD under the same assumptions (see Theorem 8 in Appendix C.1.1 and Theorem 12 in Appendix C.1.2) and are derived based on the exact simulation of (HF). If we replace $\mathsf{HF}_{\eta_k}$ with a one-step leapfrog integrator [Sanz-Serna, 1992] for implementation, then HF-opt recovers exactly the GD algorithm (see Appendix D.2), and thus inherits the same convergence guarantees as GD.

In this paper, we aim to achieve accelerated convergence rates analogous to Nesterov's accelerated gradient descent (AGD). Wang [2024] show that HF-opt achieves the accelerated convergence rate for minimizing strongly convex quadratic functions when the integration time $\eta_k$ is selected based on the roots of Chebyshev polynomials. Inspired by the randomized Hamiltonian Monte Carlo (RHMC) algorithm for sampling [Bou-Rabee and Sanz-Serna, 2017] where the integration time is independently drawn from an exponential distribution, we study its optimization counterpart to explore accelerated convergence rates for a broader class of objectives beyond quadratic functions.

## 3 Randomized Hamiltonian flow for optimization

We propose a new optimization counterpart of RHMC, called the **randomized Hamiltonian flow for optimization** (RHF-opt), where the integration time is drawn from an exponential distribution.

---

**Algorithm 2 Randomized Hamiltonian Flow for Optimization** (RHF-opt)

---

1: Initialize $x_0 \in \mathbb{R}^d$. Specify $\gamma(t) > 0$ for all $t \geq 0$.
2: **for** $k = 0, 1, \ldots, K - 1$ **do**
3:      Set the current time $T_k = \sum_{i=0}^{k-1} \tau_i$    (set $T_0 = 0$)
4:      Independently sample $\tau_k \sim \mathrm{Exp}\left(\gamma(T_k)\right)$
5:      Set $x_{k+1} = \Pi_1 \circ \mathsf{HF}_{\tau_k}(x_k, 0)$      $\triangleright$ (evolve (HF) for time $\tau_k$ and project to first component)
6: **end for**
7: **return** $x_K$

---

In Algorithm 2, the $k$-th integration time $\tau_k$ is a random variable drawn from an exponential distribution with mean $1/\gamma(T_k)$, where $T_k$ is the current time. Note that $\gamma(t)$ can depend on time. Below, we choose $\gamma(t)$ to be a constant when $f$ is strongly convex, and $\gamma(t) \propto 1/t$ when $f$ is weakly convex.

### 3.1 Reformulation of RHF-opt as a continuous-time process

To rigorously state convergence rates of Algorithm 2 (RHF-opt), we first describe an equivalent formulation of RHF-opt as the following piecewise deterministic continuous-time process that we refer to as the **randomized Hamiltonian flow** (RHF):

1. Evolve (HF) between velocity refreshment events.

2. At random jump times governed by an inhomogeneous Poisson process with rate $\gamma(t)$, we refresh the velocity to 0, and continue evolving (HF).

In the continuous-time perspective, $t \geq 0$ denotes the actual time variable. The sequence $\{T_k\}_{k \geq 0}$ in Algorithm 2 (RHF-opt) represents the random refreshment times generated by cumulative sums

of independent exponential random variables with rates $\gamma(T_k)$. Equivalently, the continuous-time process described above can be modeled as the following stochastic process:

$$\begin{cases} \mathrm{d}X_t = Y_t \, \mathrm{d}t, \\ \mathrm{d}Y_t = -\nabla f(X_t) \, \mathrm{d}t - Y_t \, \mathrm{d}N_t, \end{cases} \tag{RHF}$$

where $\mathrm{d}N_t := \sum_{k \geq 1} \delta_{T_k}(\mathrm{d}t)$ is the Poisson point process with rate $\gamma(t)$, and $T_k$ is the $k$-th time an event happens. Let $Y_{t^-}$ be the left limit of $Y_t$. At each random time $T_k$, the second line in (RHF) updates $Y_{T_k} - Y_{T_k^-} = -Y_{T_k^-}$, which refreshes the velocity to $Y_{T_k} = 0$. See also Even et al. [2021, Appendix C] for a review of the Poisson point measure and the left limit update described above.

### 3.2 Accelerated convergence rates of the randomized Hamiltonian flow

We establish the accelerated convergence rates of (RHF) for minimizing strongly and weakly convex functions. Our proofs use the continuity equation along (RHF); see Lemma 13 in Appendix F.

#### 3.2.1 For strongly convex functions

We show the following convergence rate of (RHF) under strong convexity; see Appendix F.1 for the proof. In the result below, the expectation is over the randomness in $(X_t, Y_t) \in \mathbb{R}^{2d}$, which comes from the random integration times in (RHF).

---

**Theorem 2.** *Assume $f$ is $\alpha$-strongly convex. Let $(X_t, Y_t)$ evolve following* (RHF) *with the choice $\gamma(t) = \sqrt{\frac{16\alpha}{5}}$, from any $X_0 \in \mathbb{R}^d$ with $Y_0 = 0$. Then for any $t \geq 0$, we have*

$$\mathbb{E}\left[f(X_t) - f(x^*)\right] \leq \exp\left(-\sqrt{\frac{\alpha}{5}} t\right) \mathbb{E}\left[f(X_0) - f(x^*) + \frac{\alpha}{10} \|X_0 - x^*\|^2\right].$$

---

Compared with HF-opt with short integration time (Theorem 1), RHF achieves faster convergence for strongly convex functions without smoothness assumption. Recall that the convergence rates for minimizing $\alpha$-strongly convex functions are $O(\exp(-2\alpha t))$ for the gradient flow (GF) and $O(\exp(-\sqrt{\alpha} t))$ for the accelerated gradient flow (AGF) [Wibisono et al., 2016] (see Theorems 6 and 7 in Appendix C.1.1). In comparison, RHF achieves a faster convergence rate than GF when $\alpha$ is small, and it matches the accelerated rate of AGF up to constants, albeit in expectation.

#### 3.2.2 For weakly convex functions

We show the convergence rate of (RHF) under weak convexity; see Appendix F.2 for the proof.

---

**Theorem 3.** *Assume $f$ is weakly convex. Let $(X_t, Y_t)$ evolve following* (RHF) *with the choice $\gamma(t) = \frac{6}{t+1}$, from any $X_0 \in \mathbb{R}^d$ with $Y_0 = 0$. Then for any $t \geq 0$, we have*

$$\mathbb{E}\left[f(X_t) - f(x^*)\right] \leq \frac{5 \cdot \mathbb{E}\left[f(X_0) - f(x^*) + \|X_0 - x^*\|^2\right]}{(t+1)^2}.$$

---

Compared with HF-opt with short integration time (Theorem 1), RHF achieves faster convergence for weakly convex functions without smoothness assumption. Recall the convergence rates for minimixing weakly convex functions are $O(1/t)$ for GF and $O(1/t^2)$ for AGF [Su et al., 2016, Wibisono et al., 2016] (see Theorems 10 and 11 in Appendix C.1.2). In this case as well, RHF improves upon the rate of GF and matches the accelerated rate of AGF, albeit in expectation.

The convergence guarantees in Theorems 2 and 3 are still idealized because they assume we can exactly simulate Hamiltonian flow (HF). In Section 4, we discuss a practical implementation of RHF.

## 4 Randomized Hamiltonian gradient descent

We study the discretization and implementation of the randomized Hamiltonian flow (RHF) as a discrete-time algorithm. We consider two sources of approximation in the discretization process.

**Approximate Poisson process.** In RHF, velocity is refreshed at random times governed by a Poisson process with rate $\gamma(t)$. For a small time increment $h > 0$, the probability of a refreshment event occurring in $[t, t+h]$ is approximately $\gamma(t) \cdot h$, with the probability of multiple events occurring in the same interval being negligible (order $o(h)$). Thus, given $x_0 \in \mathbb{R}^d$ and $y_0 = 0$, RHF can be approximated by alternating between a deterministic integration step of (HF) over time $h$ to generate a proposal and a probabilistic accept-refresh step for $k \geq 0$:

1. **Generate proposal**: $(x_{k+1}, \tilde{y}_{k+1}) = \mathsf{HF}_h(x_k, y_k)$

2. **Accept-refresh**: $y_{k+1} = \begin{cases} \tilde{y}_{k+1} & \text{with probability } 1 - \min(\gamma(kh) \cdot h, 1) \\ 0 & \text{with probability } \min(\gamma(kh) \cdot h, 1) \end{cases}$

As $h \to 0$, the process above recovers RHF.

**Approximate Hamiltonian flow.** In practice, we need to simulate the Hamiltonian flow (HF) using a numerical integrator, such as the leapfrog integrator [Leimkuhler and Reich, 2004, Sanz-Serna, 1992, Bou-Rabee and Sanz-Serna, 2018]. Accordingly, we replace the exact flow map $\mathsf{HF}_h$ with a discrete-time integrator $\mathsf{T}_h : \mathbb{R}^d \times \mathbb{R}^d \to \mathbb{R}^d \times \mathbb{R}^d$ given stepsize $h$. As a first step, we consider $\mathsf{T}_h$ to be the implicit (backward Euler) integrator. The update for $(x_{k+1}, \tilde{y}_{k+1}) = \mathsf{T}_h(x_k, y_k)$ satisfies the following system of implicit equations:

$$x_{k+1} - x_k = h\tilde{y}_{k+1}, \tag{2a}$$
$$\tilde{y}_{k+1} - y_k = -h\nabla f(x_{k+1}). \tag{2b}$$

By substituting $\tilde{y}_{k+1}$ in (2a) with $y_k - h\nabla f(x_{k+1})$ from (2b), updates (2) can be reformulated as

$$x_{k+1} = \mathrm{Prox}_{h^2 f}(x_k + hy_k), \tag{3a}$$
$$\tilde{y}_{k+1} = y_k - h\nabla f(x_{k+1}). \tag{3b}$$

where $\mathrm{Prox}_{h^2 f}(x) = \arg\min_{y \in \mathbb{R}^d}\left\{f(y) + \frac{1}{2h^2}\|y - x\|^2\right\}$ is the proximal operator. If we can implement the proximal operator for $f$, then the updates (3) above yield a concrete algorithm that we call the **randomized proximal Hamiltonian descent** (RPHD); see Appendix G.1 for more details on RPHD and its convergence analysis. However, the proximal step (3a) is not explicit for general $f$, and thus we make one further approximation to turn it into a concrete algorithm.

**Algorithm.** Let $x_{k+\frac{1}{2}} := x_k + hy_k$. We approximate the proximal step (3a) by gradient descent:

$$x_{k+1} = x_{k+\frac{1}{2}} - h^2\nabla f(x_{k+\frac{1}{2}}). \tag{4}$$

This modification leads to a practical algorithm that we call the **randomized Hamiltonian gradient descent** (RHGD), summarized in Algorithm 3. Note that as $h \to 0$, RHGD recovers RHF.

---

**Algorithm 3 Randomized Hamiltonian Gradient Descent** (RHGD)

---
1: Initialize $x_0 \in \mathbb{R}^d$ and $y_0 = 0$. Choose stepsize $h > 0$ and refreshment rate $\gamma_k > 0$.
2: **for** $k = 0, 1, \ldots, K - 1$ **do**
3:      $x_{k+\frac{1}{2}} = x_k + hy_k$
4:      $x_{k+1} = x_{k+\frac{1}{2}} - h^2\nabla f(x_{k+\frac{1}{2}})$
5:      $\tilde{y}_{k+1} = y_k - h\nabla f(x_{k+1})$
6:      $y_{k+1} = \begin{cases} \tilde{y}_{k+1} & \text{with probability } 1 - \min(\gamma_k \cdot h, 1) \\ 0 & \text{with probability } \min(\gamma_k \cdot h, 1) \end{cases}$
7: **end for**
8: **return** $x_K$

---

## 4.1 Accelerated convergence rates of RHGD

RHGD serves as a practical implementation of the randomized Hamiltonian flow (RHF). In the following, we analyze the convergence rates of RHGD under both strong and weak convexity.

### 4.1.1 For strongly convex functions

We show the following accelerated convergence rate of RHGD for minimizing smooth and strongly convex functions. The proof is deferred to Appendix G.3.1.

> **Theorem 4.** *Assume $f$ is $\alpha$-strongly convex and $L$-smooth. Then for all $k \geq 0$, RHGD (Algorithm 3) with $h \leq \frac{1}{4\sqrt{L}}$, $\gamma_k = \sqrt{\alpha}$, and from any $x_0 \in \mathbb{R}^d$ satisfies*
>
> $$\mathbb{E}[f(x_k) - f(x^*)] \leq \left(1 + \frac{\sqrt{\alpha}h}{6}\right)^{-k} \mathbb{E}\left[f(x_0) - f(x^*) + \frac{\alpha}{72}\|x_0 - x^*\|^2\right].$$

**Corollary 1.** *Assume $f$ is $\alpha$-strongly convex and $L$-smooth. To generate $x_K$ satisfying $\mathbb{E}[f(x_K) - f(x^*)] \leq \varepsilon$, it suffices to run Algorithm 3 with $h = \frac{1}{4\sqrt{L}}$, $\gamma_k = \sqrt{\alpha}$, and from any $x_0 \in \mathbb{R}^d$ for*

$$K \geq (24\sqrt{\kappa} + 1) \cdot \log\left(\frac{\mathbb{E}\left[f(x_0) - f(x^*) + \frac{\alpha}{72}\|x_0 - x^*\|^2\right]}{\varepsilon}\right).$$

Corollary 1 shows that RHGD requires $O(\sqrt{\kappa}\log(1/\varepsilon))$ iterations to generate an $\varepsilon$-accurate solution in expectation under smoothness and strong convexity. Recall under the same assumptions, GD achieves the iteration complexity of $O(\kappa \log(1/\varepsilon))$, whereas AGD achieves the improved iteration complexity of $O(\sqrt{\kappa}\log(1/\varepsilon))$ (see Corollaries 3 and 4 in Appendix C.1.1). In comparison, RHGD is faster than GD, and matches the accelerated rate of AGD, albeit in expectation.

### 4.1.2 For weakly convex functions

We show the following convergence rate of RHGD for minimizing smooth and weakly convex functions. The proof is deferred to Appendix G.3.3.

> **Theorem 5.** *Assume $f$ is weakly convex and $L$-smooth. Then for all $k \geq 0$, RHGD (Algorithm 3) with $h \leq \frac{1}{7\sqrt{L}}$, $\gamma_k = \frac{17}{2(k+9)h}$, and from any $x_0 \in \mathbb{R}^d$ satisfies*
>
> $$\mathbb{E}[f(x_k) - f(x^*)] \leq \frac{14 \cdot \mathbb{E}\left[\|x_0 - x^*\|^2\right]}{h^2(k + 8)^2}.$$

**Corollary 2.** *Assume $f$ is weakly convex and $L$-smooth. To generate $x_K$ satisfying $\mathbb{E}[f(x_K) - f(x^*)] \leq \varepsilon$, it suffices to run Algorithm 3 with $h = \frac{1}{7\sqrt{L}}$, $\gamma_k = \frac{17}{2(k+9)h}$ and any $x_0 \in \mathbb{R}^d$ for*

$$K \geq \sqrt{\frac{686L \cdot \mathbb{E}\left[\|x_0 - x^*\|^2\right]}{\varepsilon}}.$$

Corollary 2 shows that RHGD requires $O(\sqrt{L/\varepsilon})$ iterations to generate an $\varepsilon$-accurate solution in expectation under $L$-smoothness and weak convexity. Recall under the same assumptions, GD achieves the iteration complexity of $O(L/\varepsilon)$, whereas AGD achieves the improved iteration complexity of $O(\sqrt{L/\varepsilon})$ (see Corollaries 5 and 6 in Appendix C.1.2). In comparison, RHGD is faster than GD, and matches the accelerated rate of AGD, albeit in expectation.

### 4.2 Discussion

Unlike the convergence rates of GD and AGD, which hold deterministically for $f(x_k) - f(x^*)$, the convergence rate of RHGD holds in expectation, i.e., $\mathbb{E}[f(x_k) - f(x^*)]$ due to the random refreshment. Nevertheless, convergence in expectation can still imply high-probability bounds for $f(x_k) - f(x^*)$ via Markov's inequality. We also remark that the continuized version of AGD (CAGD) proposed by Even et al. [2021] and studied by Wang and Wibisono [2023b], where the two variables continuously mix following a linear ordinary differential equation and take gradient steps at random times, similarly achieves an accelerated convergence rate in expectation.

**Proof Sketch.** We first establish the convergence of the ideal algorithm RPHD (Algorithm 5). Using a Lyapunov function $E_k$, we show that it preserves the accelerated convergence via $E_{k+1} \leq E_k$ (see Theorems 14 and 15). The analysis for the practical algorithm RHGD follows similarly but accounts for the approximation of the proximal step (3a) using gradient descent (4). We bound the resulting error, which depends on the gradient norm (see Proposition 1), and then incorporate it into the Lyapunov decrease. Unlike prior works (e.g.,[Wilson et al., 2021]), our analysis avoids explicitly tracking intermediate iterates, enabling flexibility in the choice of approximation for (3a).

## 5 Numerical experiments

In this section, we validate the empirical effectiveness of RHGD (Algorithm 3) through numerical experiments on two canonical convex optimization problems: (1) quadratic minimization and (2) logistic regression. We compare our proposed algorithm RHGD with GD, AGD, and its continuized version CAGD [Even et al., 2021], whose pseudocodes are listed in Appendix H.3 as Algorithms 6, 7, and 8, respectively. We evaluate their performance under various condition numbers with fine-tuned or adaptive stepsizes. For the quadratic problem, we also evaluate the robustness of RHGD to the misspecification of strong convexity constant $\alpha$. We denote the stepsize of GD, AGD and CAGD by $\eta$ and the stepsize of RHGD by $h$ to reflect their different scales in smoothness constant $L$. See Appendix H for full details on our experiments. Code is available at `https://github.com/QiangFu09/RHGD`.

### 5.1 Minimizing quadratic functions

Consider the quadratic optimization problem: $\min_{x \in \mathbb{R}^d} \left\{ f(x) = \frac{1}{2} x^\top A x \right\}$, where $A \in \mathbb{R}^{d \times d}$ is a positive semi-definite matrix with the smallest eigenvalue $0 \leq \alpha \leq 1$ and the largest eigenvalue $L \geq 1$. In our experiments, we set $d = 100$, select the stepsize for each algorithm via a grid search and choose the largest value that ensures convergence and stability. Larger stepsizes result in numerical instability or divergence. Specifically, we optimize $\eta$ over geometric sequence $\mathcal{S}_\eta^L := \{c/L : c = 2^n, \, n \in \mathbb{Z}\}$ with ratio 2, and optimize $h$ over geometric sequence $\mathcal{S}_h^L := \{\sqrt{c/L} : c = 2^n, \, n \in \mathbb{Z}\}$ with ratio $\sqrt{2}$. See Tables 2 and 3 in Appendix H.1.3 for the optimal stepsizes via grid search of each setting.

We consider both strongly convex ($\alpha > 0$) and weakly convex ($\alpha = 0$) quadratic minimization. For the strongly convex case, we fix $L = 500$ and test three condition numbers $\kappa \in \{10^3, 10^5, 10^7\}$ with $\alpha = L/\kappa$. Since AGD, CAGD, and RHGD require $\alpha$ to compute momentum parameters and refreshment rates, we first compare all algorithms using the exact value of $\alpha$. Notably, when the condition number $\kappa$ is large, the strong convexity constant $\alpha$ can be extremely small (e.g., $\kappa = 10^7$, $\alpha = 5 \times 10^{-5}$), which makes an accurate estimation of $\alpha$ challenging in practice. To evaluate robustness, we also test with misspecified values $\hat{\alpha} \in \{0.01, 0.1, 1\} > \alpha$. In this section, we report results for $\kappa = 10^7$ and $\hat{\alpha} = 0.01$; additional results are deferred to Appendix H.1.1. For the weakly convex case, we evaluate the same algorithms with appropriately configured parameters (see Appendix H.3). In particular, RHGD uses decaying $\gamma_k = \frac{17}{2(k+9)h}$ instead of a constant.

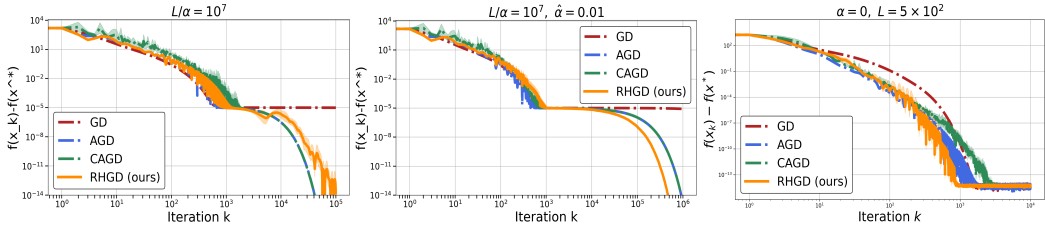

Figure 1: Comparison between GD, AGD, CAGD, and RHGD (ours) on minimizing quadratic functions with $L = 500$ in three settings: (1) $\kappa = 10^7$ with exact $\alpha$ (left); (2) $\kappa = 10^7$ with misspecified $\hat{\alpha} = 0.01$ (middle); (3) $\alpha = 0$ (right). We use optimal stepsizes via grid search for each setting. Each plot shows results averaged over 5 runs.

**Summary of experimental results.** Figure 1 presents results for the strongly convex setting with exact $\alpha$ (left), misspecified $\hat{\alpha} = 0.01$ (middle), and the weakly convex setting (right). In the left plot, all accelerated methods clearly outperform GD. While slower than AGD and CAGD, which are known to be optimal under exact parameter knowledge, RHGD remains highly competitive during the early iterations. The middle plot highlights the robustness of RHGD to misspecification. The momentum parameters of AGD and CAGD, and the refreshment rate of RHGD are computed using overestimated values $\hat{\alpha} = 0.01$, which are significantly larger than the true value $\alpha = 5 \times 10^{-5}$. While AGD and CAGD degrade significantly when using overestimated $\hat{\alpha}$, RHGD maintains stable performance and outperforms them in later stages. The right plot shows that RHGD achieves the fastest convergence among all algorithms, consistently improving over AGD and CAGD throughout the iterations, demonstrating its advantage in the weakly convex setting.

## 5.2 Minimizing logistic regression loss

We construct a synthetic binary classification task using logistic regression with $\ell_2$-regularization:

$$\min_{x \in \mathbb{R}^d} \left\{ f(x) = \frac{1}{n} \sum_{i=1}^{n} \log \left( 1 + \exp(-b_i a_i^\top x) \right) + \frac{\alpha}{2} \|x\|^2 \right\},$$

where we set $d = 100$ and $n = 500$. Details on objective generation is provided in Appendix H.2. We consider the strongly convex ($\alpha \in \{10^{-3}, 10^{-4}, 10^{-5}\}$) and weakly convex ($\alpha = 0$) settings. We report results for $\alpha \in \{10^{-4}, 0\}$ and defer the others to Appendix H.2. Since the smoothness constant $L$ is unknown, we evaluate GD, AGD and CAGD using adaptive stepsizes the same way as in Hinder et al. [2020] through line search (see Algorithms 9, 10 and 11 in Appendix H.3). For implementation of RHGD, we also adopt a similar adaptive stepsize strategy inspired by the line search in Hinder et al. [2020] (see Algorithm 12 in Appendix H.3). We initialize the stepsize $h = 1$. At each iteration, $h$ is updated based on checking the condition: $f(\tilde{x}_{k+1}) < f(x_{k+\frac{1}{2}}) - \frac{h^2}{2}\|\nabla f(x_{k+\frac{1}{2}})\|^2$. If the condition is satisfied, then we increase the stepsize by setting $h \leftarrow \sqrt{1.1} \cdot h$ and accept $\tilde{x}_{k+1}$ as $x_{k+1}$; otherwise, we decrease the stepsize by setting $h \leftarrow \sqrt{0.6} \cdot h$ and reject $\tilde{x}_{k+1}$ by setting $x_{k+1} \leftarrow x_k$.

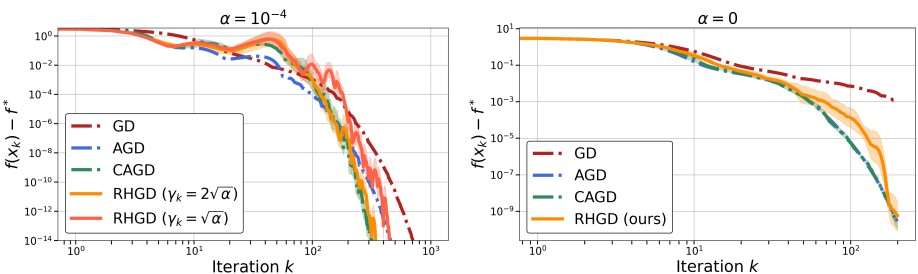

Figure 2: Comparison of GD, AGD, CAGD and RHGD on logistic regression with $\alpha \in \{10^{-4}, 0\}$. We run each algorithm using adaptive stepsizes. For RHGD, we evaluate $\gamma_k = \sqrt{\alpha}$ and $\gamma_k = 2\sqrt{\alpha}$ for $\alpha > 0$, and evaluate $\gamma_k = 17/(2k+18)h$ for $\alpha = 0$.

**Summary of experimental results.** Figure 2 presents the convergence behavior of all tested algorithms on a logistic regression task with $\alpha \in \{10^{-4}, 0\}$ and adaptive stepsizes. We observe from the left plot that using a slightly larger refreshment rate, $\gamma_k = 2\sqrt{\alpha}$, results in faster and more stable convergence, while Theorem 4 suggests setting the refreshment rate to $\gamma_k = \sqrt{\alpha}$. Our proposed algorithm RHGD with $\gamma_k = 2\sqrt{\alpha}$ and $\gamma_k = \sqrt{\alpha}$ are faster than GD in the late iterations and comparable to AGD and CAGD. The right plot presents the results in the weakly convex setting with adaptive stepsizes. For RHGD, we choose the decaying refreshment rate $\gamma_k = \frac{17}{2(k+9)h}$. All accelerated methods consistently outperform GD, and RHGD exhibits comparable performance to baseline accelerated algorithms AGD and CAGD.

# 6   Conclusion and limitations

In this work, we propose the randomized Hamiltonian flow (RHF) and its discretization, the randomized Hamiltonian gradient descent (RHGD), and establish their accelerated convergence guarantees for both strongly and weakly convex functions. Numerical experiments on quadratic minimization and logistic regression tasks demonstrate that RHGD is faster than GD, and consistently matches or outperforms baseline accelerated algorithms such as AGD and CAGD.

We outline several promising directions for future research that build upon and extend this work. First, it would be valuable to explore other integrators for discretization, such as the leapfrog integrator, which is commonly used in Hamiltonian Monte Carlo (HMC) and may improve numerical stability or efficiency in the optimization context.

Second, a natural extension is to study whether our framework and convergence analysis can be generalized to broader classes of non-convex functions, such as quasar-convex functions [Hardt et al., 2018], for which recent works have shown potential for accelerated convergence [Hinder et al., 2020, Fu et al., 2023, Wang and Wibisono, 2023b].

Moreover, an important direction is the development of a stochastic variant of RHGD, where gradients are estimated using mini-batches or variance reduction. This would make the method more suitable for large-scale optimization tasks where computing full gradients is computationally prohibitive. Finally, we hope that our analysis of randomized Hamiltonian flows for optimization can provide theoretical tools and insights toward validating a long-standing conjecture in sampling: randomized Hamiltonian Monte Carlo can achieve accelerated mixing time in KL divergence for sampling from strongly log-concave distributions.

**Limitations.**   Similar to CAGD [Even et al., 2021], our convergence guarantees for RHF and RHGD hold in expectation due to their inherent randomness. While this differs from deterministic algorithms like GD and AGD, we can extend our results to high-probability bounds via concentration inequality. Additionally, while our rates $O(\exp(-\sqrt{\alpha/5}\,t))$ for RHF and $O((1 + \sqrt{\alpha}h/6)^{-k})$ for RHGD exhibit the same $\sqrt{\alpha}$ dependence as optimal accelerated methods (AGF and AGD, see Appendix C.1.1), the constants are somewhat larger. Nevertheless, our algorithm RHGD is still faster than GD and competitive with AGD in both theory and practice.

## Acknowledgment

The authors thank the anonymous reviewers for their thoughtful feedback and Jun-Kun Wang and Molei Tao for insightful discussions.

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

# Contents

## A   Related work

**Hamiltonian Monte Carlo (HMC).**   HMC and its variants are widely-used sampling algorithms implemented as the default samplers in popular Bayesian inference packages such as Stan [Carpenter et al., 2017] and PyMC3 [Salvatier et al., 2016]. Notably, the No-U-Turn Sampler (NUTS) [Hoffman et al., 2014], a celebrated HMC-based algorithm that automatically tunes its parameters, serves as the default sampling method in Stan. However, establishing rigorous theoretical guarantees, such as non-asymptotic convergence rates and developing principled acceleration methods, remain active areas of research [Talay, 2002, Pakman and Paninski, 2013, Beskos et al., 2013, Betancourt et al., 2014, Seiler et al., 2014, Durmus et al., 2017, Mangoubi and Smith, 2017, Lee et al., 2018, Bou-Rabee and Sanz-Serna, 2018, Lee and Vempala, 2018, Mangoubi and Vishnoi, 2018, Mangoubi and Smith, 2019, Bou-Rabee et al., 2020, Chen et al., 2020, Bou-Rabee and Eberle, 2021, Mangoubi and Smith, 2021, Lu and Wang, 2022, Wang and Wibisono, 2023a, Monmarché, 2022, Jiang, 2023, Chen and Gatmiry, 2023, Camrud et al., 2023, Bou-Rabee and Schuh, 2023, Bou-Rabee and Eberle, 2023, Monmarché, 2024, Bou-Rabee and Marsden, 2025]. Many recent works focus on understanding the non-asymptotic behavior of HMC [Wang and Wibisono, 2023a, Jiang, 2023, Monmarché, 2024, Bou-Rabee and Marsden, 2025]. These analyses not only quantify convergence rates to the target distribution but also provide guidance on selecting optimal hyperparameters and developing robust stopping criteria, thereby offering further insights for algorithmic improvement. A variant of HMC,

known as Randomized Hamiltonian Monte Carlo (RHMC), has been studied in [Bou-Rabee and Sanz-Serna, 2017]. An intriguing connection between RHMC and underdamped Langevin dynamics is explored in [Riou-Durand and Vogrinc, 2022, Cao et al., 2023, Jiang, 2023, Leimkuhler et al., 2024]: specifically, the stochastic differential equation (SDE) characterizing RHMC reduces to the underdamped Langevin dynamics in the limit [Cheng et al., 2018, Cao et al., 2020, Zhang et al., 2023]. Additionally, several variants of HMC have been proposed to address different challenges. Some works introduce stochastic gradients to scale HMC to large datasets [Chen et al., 2014, Zou and Gu, 2021]. Others explore non-Euclidean geometry [Brofos and Lederman, 2021] or altered dynamics, such as magnetic [Tripuraneni et al., 2017], reflection-based [Mohasel Afshar and Domke, 2015], or discontinuous dynamics [Nishimura et al., 2020]. There are also efforts on adaptation and tuning [Hoffman et al., 2019, Hoffman and Sountsov, 2022], as well as decentralized or parallel implementations [Gürbüzbalaban et al., 2021]. Recent work has further proposed application-specific improvements [Monnahan et al., 2017, Robnik et al., 2023]. Additional developments also include continuous-time formulations [Zhang et al., 2012], probabilistic integrators [Dinh et al., 2017], nonparametric extensions [Mak et al., 2021], and entropy-aware methods [Hirt et al., 2021].

**Accelerated gradient flow for optimization.** Classical accelerated optimization algorithms are the heavy-ball method proposed by Polyak [1964] and accelerated gradient descent (AGD) proposed by Nesterov [1983]. The continuous-time limit of these two methods corresponds to the accelerated gradient flow (AGF), which is the combination of (HF) with a damping term for dissipating energy (see Appendix C.1 for a review), and is different from the pure HF we study in this work. Most existing Hamiltonian-based optimization methods can be seen as the discretization of (AGF). Moreover, Su et al. [2016], Krichene et al. [2015], Wibisono et al. [2016], Wilson et al. [2021], Suh et al. [2022] study continuous-time limits of (AGD), and Hu and Lessard [2017], Maddison et al. [2018], Muehlebach and Jordan [2019], O'Donoghue and Maddison [2019], Hinder et al. [2020], Wilson et al. [2021], Even et al. [2021], Attouch et al. [2022], Shi et al. [2022], Fu et al. [2023], Wang and Wibisono [2023b] design and analyze the accelerated methods as the discretization of (AGF) even beyond convexity.

**Hamiltonian flow for optimization.** To the best of our knowledge, only a few previous works have explored optimization algorithms explicitly based on energy-conserving Hamiltonian flow of the form (HF). Teel et al. [2019] studied optimization methods derived from Hamiltonian flow with a velocity refreshment strategy. Specifically, they refresh the velocity to zero whenever the iterate approaches the boundary of a region defined as $\{(x, y) \in \mathbb{R}^d \times \mathbb{R}^d : \langle \nabla f(x), y \rangle \leq 0, \|y\|^2 \geq \|\nabla f(x)\|^2 / L\}$ where $L$ is the smoothness constant, or after a fixed timer expires. They established uniform global stability and convergence guarantees for minimizing smooth and strongly convex functions. Diakonikolas and Jordan [2021] analyzed generalized Hamiltonian dynamics characterized by a time-dependent Hamiltonian $H(x, y, \tau) = h(\tau)f(x/\tau) + \psi^*(y)$, where $h(\tau)$ is a positive function of scaled time $\tau$ and $\psi^*(\cdot)$ is strongly convex and differentiable. They showed that along these Hamiltonian flows, the average gradient norm $\|\frac{1}{t} \int_0^t \nabla f(x_\tau)d\tau\|$ decreases at an $O(1/t)$ rate. Moreover, they demonstrated that a broad family of momentum methods, applicable in both Euclidean and non-Euclidean spaces, can be derived from these generalized dynamics. This family includes classical methods such as Nesterov's accelerated gradient method [Nesterov, 1983, 2018], Heavy-Ball method [Polyak, 1964], and other well-known accelerated methods [Wibisono et al., 2016, Wilson et al., 2021] as special cases, which are different from what we consider in this paper. De Luca et al. [2023] propose a Hamiltonian-based method with the Hamiltonian defined as $H(x, y) = \lambda \log(F(x) - F_0) + \log(\|y\|^2)$ where $\lambda > 0$ and $F_0$ are user-specified parameters. They show that the proposed method together with some heuristics is competitive with Adam [Kingma and Ba, 2015] in some deep learning experiments, but no convergence-rate analysis was performed in the work. Wang [2024] studies the Hamiltonian flow with velocity refreshment for minimizing strongly convex quadratic functions $f(x) = \frac{1}{2}x^\top A x - b^\top x$ with $A \in \mathbb{R}^{d \times d}$ and $\alpha I \preceq A \preceq LI$; importantly, (HF) can be solved exactly when $f$ is quadratic, which facilitates the analysis. By choosing the integration time $\eta_k$ related to the roots of Chebyshev polynomials, Wang [2024] shows that HF-opt achieves the convergence rate of $O((1 - \Theta(1/\sqrt{\kappa}))^k)$ and the total integration time $\Theta(1/\sqrt{\alpha})$, which matches the total integration of accelerated gradient flow with refreshment [Suh et al., 2022].

**Randomization in optimization.** The technique of randomly selecting integration times has been explored in optimization contexts. Even et al. [2021] study the "continuized" version of Nesterov acceleration, which also considers random update times to combine continuous-time and discrete-time

perspectives. In their framework, the algorithmic variables follow a linear ODE and take gradient steps at random intervals, maintaining the simplicity of continuous-time methods and allowing for efficient discrete-time implementations. While based on a linear ODE rather than Hamiltonian flow, this continuized approach shares a similar spirit with our randomized integration time perspective: both demonstrate how randomness in timing can yield rigorous theoretical guarantees and practical acceleration in optimization.

# B  Lift-Conserve-Project scheme

We describe a new methodology to reason about optimization tasks based on conservation rather than dissipation properties, which we call the *Lift-Conserve-Project* (LCP) scheme, and which serves as a unifying principle across optimization and sampling.

Suppose we want to solve the following optimization problem:

$$\min_{x \in \mathcal{X}} f(x), \tag{5}$$

where $\mathcal{X}$ is the domain of optimization; for example, $\mathcal{X}$ can be $\mathbb{R}^d$, a constraint set, or the space of probability distributions $\mathcal{P}(\mathbb{R}^d)$. Suppose we are currently at a state $x_0 \in \mathcal{X}$. We describe a procedure to update $x_0$ to a new state with smaller function value. We need the following ingredients:

- A space $\mathcal{Z}$ and an energy function $\mathcal{H} \colon \mathcal{Z} \to \mathbb{R}$.
- A lifting map $\mathbf{L} \colon \mathcal{X} \to \mathcal{Z}$ which preserves $f$, i.e., $f(x) = \mathcal{H}(\mathbf{L}(x))$.
- A conservation map $\mathbf{C} \colon \mathcal{Z} \to \mathcal{Z}$ which preserves $\mathcal{H}$, i.e., $\mathcal{H}(z) = \mathcal{H}(\mathbf{C}(z))$.
- A projection map $\mathbf{P} \colon \mathcal{Z} \to \mathcal{X}$ which reduces $f$, i.e., $\mathcal{H}(z) \geq f(\mathbf{P}(z))$.

The LCP scheme is an algorithm $\mathbf{LCP} \colon \mathcal{X} \to \mathcal{X}$ that applies the lifting $\mathbf{L}$, conservation $\mathbf{C}$ and projection $\mathbf{P}$ maps sequentially:

$$\mathbf{LCP} := \mathbf{P} \circ \mathbf{C} \circ \mathbf{L}.$$

**Lemma 3.** *The* LCP *scheme is a* descent method, *i.e.,* $f(\mathbf{LCP}(x)) \leq f(x)$.

*Proof.* By construction, $f(\mathbf{LCP}(x)) = f(\mathbf{P} \circ \mathbf{C} \circ \mathbf{L}(x)) \leq \mathcal{H}(\mathbf{C} \circ \mathbf{L}(x)) = \mathcal{H}(\mathbf{L}(x)) = f(x)$. $\quad\square$

## B.1  LCP scheme for optimization: Hamiltonian flow for optimization

Before describing HF-opt (Algorithm 1) as a LCP scheme, we first prove Lemmas 1 and 2 below.

**Lemma 1.** *Along* (HF), *for all* $t \geq 0$, $H(X_t, Y_t) = H(X_0, Y_0)$.

*Proof.* Applying the chain rule and the (HF) dynamics, we obtain

$$\frac{\mathrm{d}}{\mathrm{d}t} H(X_t, Y_t) = \langle \nabla_x H(X_t, Y_t), \dot{X}_t \rangle + \langle \nabla_y H(X_t, Y_t), \dot{Y}_t \rangle$$
$$= \langle \nabla f(X_t), Y_t \rangle + \langle Y_t, -\nabla f(X_t) \rangle = 0.$$

Thus, $H(X_t, Y_t)$ is invariant along (HF) and $H(X_t, Y_t) = H(X_0, Y_0)$. $\quad\square$

**Lemma 2.** *For any $k$ and $\eta_k > 0$, Algorithm 1 satisfies $f(x_{k+1}) \leq f(x_k)$.*

*Proof.* In one step of Algorithm 1, we run (HF) from $(X_0, Y_0) = (x_k, 0)$ to reach $(X_{\eta_k}, Y_{\eta_k}) = (x_{k+1}, y_{k+1})$ for some $y_{k+1} \in \mathbb{R}^d$. By Lemma 1, $f(x_{k+1}) = f(x_k) - \frac{1}{2}\|y_{k+1}\|^2 \leq f(x_k)$. $\quad\square$

Since (HF) is a conservative flow by Lemma 1, we can choose $\mathcal{X} = \mathbb{R}^d$, $\mathcal{Z} = \mathbb{R}^d \times \mathbb{R}^d$, $\mathcal{H} = H$, which is the Hamiltonian function defined by $H(x, y) = f(x) + \frac{1}{2}\|y\|^2$, $\mathbf{L}(x) = (x, 0)$, $\mathbf{C} = \mathsf{HF}_\eta$ which is the Hamiltonian flow map with integration time $\eta > 0$, and $\mathbf{P} = \Pi_1$, which is the projection to the first coordinate. Given $x_0 \in \mathbb{R}^d$, the LCP scheme is described as

1. **Lift:** Lift the point $x_0 \in \mathbb{R}^d$ to the augmented phase space $\mathbb{R}^d \times \mathbb{R}^d$ via

$$\mathbf{L}(x_0) = (x_0, 0).$$

Note here $y_0 = 0$ is the minimizer of the kinetic energy $\frac{1}{2}\|y\|^2$.

2. **Conserve:** Evolve the Hamiltonian flow (HF) starting from $(x_0, 0)$ for a duration $\eta$ to obtain

$$(x_\eta, y_\eta) = \mathsf{HF}_\eta(x_0, 0).$$

3. **Project:** Project $(x_\eta, y_\eta)$ back to the original space by discarding the velocity:

$$\Pi_1(x_\eta, y_\eta) = x_\eta.$$

By Lemma 3, we have $f(x_\eta) \leq f(x_0)$. This is exactly the HF-opt (Algorithm 1).

## B.2 LCP scheme for sampling: Hamiltonian Monte Carlo

In sampling, the goal is to generate a random variable $X \in \mathbb{R}^d$ from a target distribution $\nu \in \mathcal{P}(\mathbb{R}^d)$. This is a fundamental problem that frequently arises in Bayesian statistics. Sampling algorithms typically employ random walks, where each iteration applies a stochastic update to the current point. When the target distribution has the form $\nu \propto \exp(-f)$, the sampling problem can be equivalently formulated as minimizing the relative entropy (Kullback–Leibler (KL) divergence) with respect to $\nu$ over the space of probability distributions $\mathcal{P}(\mathbb{R}^d)$:

$$\text{Sample } X \sim \nu \propto \exp(-f) \qquad \Longleftrightarrow \qquad \min_{\rho \in \mathcal{P}(\mathbb{R}^d)} \mathsf{KL}(\rho \,\|\, \nu), \tag{6}$$

where $\mathsf{KL}(\rho \,\|\, \nu) = \int_{\mathbb{R}^d} \rho(x) \log \frac{\rho(x)}{\nu(x)}\, dx$ is the KL divergence between $\rho$ and $\nu$. Many sampling algorithms can be interpreted as implementing optimization methods to solve the optimization problem (6) in the space of distributions; for example, the Langevin dynamics is the gradient flow for minimizing KL divergence under the Wasserstein geometry [Jordan et al., 1998, Wibisono, 2018]. In contrast, another classical and widely used sampling algorithm is the Hamiltonian Monte Carlo (HMC, Algorithm 4) [Duane et al., 1987, Neal et al., 2011], which often achieves better performance in practice compared to Langevin-based methods.

---

**Algorithm 4 Hamiltonian Monte Carlo (HMC)**

---

1: Initialize $x_0 \in \mathbb{R}^d$. Choose integration time $\eta_k > 0$ for $k \geq 0$.
2: **for** $k = 0, 1, \ldots, K - 1$ **do**
3:     Sample velocity $\xi \sim \mathcal{N}(0, I_d)$.
4:     Set $(x_{k+1}, v_{k+1}) = \mathsf{HF}_{\eta_k}(x_k, \xi)$.
5: **end for**
6: **return** $x_K$

---

### B.2.1 Distributional properties of Hamiltonian flow

Define an auxiliary probability distribution $\tilde{\nu}(x, y) \propto \exp\left(-f(x) - \frac{1}{2}\|y\|^2\right)$ on the phase space $\mathbb{R}^d \times \mathbb{R}^d$. The following lemma demonstrates that KL divergence in the phase space with respect to $\tilde{\nu}$ is conserved along HF. In the proof below, for simplicity we suppress the arguments in the integrals.

**Lemma 4.** *Let* $(x_t, y_t) = \mathsf{HF}_t(x_0, y_0)$ *denote the solution to* (HF) *at time* $t$ *where* $(x_t, y_t) \sim \rho_t$. *Then we have*

$$\mathsf{KL}(\rho_t \,\|\, \tilde{\nu}) = \mathsf{KL}(\rho_0 \,\|\, \tilde{\nu}). \tag{7}$$

*Proof.* Since $(x_t, y_t)$ satisfies $(\dot{x}_t, \dot{y}_t) = (y_t, -\nabla f(x_t))$, using the continuity equation in Lemma 14 with the Hamiltonian vector field $v_t(x, y) = (y, -\nabla f(x))^\top$ yields

$$\partial_t \rho_t(x, y) + \nabla \cdot \left( \rho_t(x, y) \begin{pmatrix} y \\ -\nabla f(x) \end{pmatrix} \right) = 0.$$

Computing the time derivative of $\mathsf{KL}(\rho_t \| \tilde{\nu})$ along (HF), we obtain

$$\frac{\mathrm{d}}{\mathrm{d}t}\mathsf{KL}(\rho_t \| \tilde{\nu}) = \frac{\mathrm{d}}{\mathrm{d}t}\int \rho_t \log \frac{\rho_t}{\tilde{\nu}_t} = \int \partial_t \rho_t \log \frac{\rho_t}{\tilde{\nu}_t} + \int \rho_t \frac{\partial_t \rho_t}{\rho_t}$$

$$\overset{(7)}{=} -\int \nabla \cdot \left(\rho_t \begin{pmatrix} y \\ -\nabla f(x) \end{pmatrix}\right) \log \frac{\rho_t}{\tilde{\nu}_t} + \underbrace{\int \partial_t \rho_t}_{=0}$$

$$= \int \left\langle \nabla \log \frac{\rho_t}{\tilde{\nu}}, \begin{pmatrix} y \\ -\nabla f(x) \end{pmatrix}\right\rangle \rho_t$$

$$= \int \left\langle \nabla \log \rho_t, \begin{pmatrix} y \\ -\nabla f(x) \end{pmatrix}\right\rangle \rho_t + \underbrace{\int \left\langle \begin{pmatrix} \nabla f(x) \\ y \end{pmatrix}, \begin{pmatrix} y \\ -\nabla f(x) \end{pmatrix}\right\rangle \rho_t}_{=0}$$

$$= \int \left\langle \nabla \rho_t, \begin{pmatrix} y \\ -\nabla f(x) \end{pmatrix}\right\rangle = -\int \nabla \cdot \begin{pmatrix} y \\ -\nabla f(x) \end{pmatrix} \rho_t = 0,$$

where $\nabla := (\nabla_x, \nabla_y)^\top$, and the third and the last equations follow from integration by part. Thus $\frac{\mathrm{d}}{\mathrm{d}t}\mathsf{KL}(\rho_t \| \tilde{\nu}) = 0$, which implies $\mathsf{KL}(\rho_t \| \tilde{\nu}) = \mathsf{KL}(\rho_0 \| \tilde{\nu})$. $\qquad\square$

The next lemma shows the chain rule decomposition of KL divergence. For a joint distribution $\rho \in \mathcal{P}(\mathbb{R}^d \times \mathbb{R}^d)$, let $\rho^X(x) = \int \rho(x,y)\,\mathrm{d}y$ be the $X$-marginal of $\rho$, and $\rho^{Y|X}(y \mid x) = \frac{\rho(x,y)}{\rho^X(x)}$ the conditional distribution, so we have the factorization $\rho(x,y) = \rho^X(x)\rho^{Y|X}(y \mid x)$.

**Lemma 5.** *For any joint distributions $\rho, \pi \in \mathcal{P}(\mathbb{R}^d \times \mathbb{R}^d)$, we have*

$$\mathsf{KL}(\rho \,\|\, \pi) = \mathsf{KL}(\rho^X \,\|\, \pi^X) + \mathbb{E}_{\rho^X}\left[\mathsf{KL}(\rho^{Y|X} \,\|\, \pi^{Y|X})\right]. \tag{8}$$

*Proof.* Using the definition of KL divergence and the conditional distribution, we obtain

$$\mathsf{KL}(\rho \,\|\, \pi) = \int \rho \log \frac{\rho}{\pi} = \int \rho^X \rho^{Y|X} \log \frac{\rho^X \rho^{Y|X}}{\pi^X \pi^{Y|X}}$$

$$= \int \rho^X \rho^{Y|X} \log \frac{\rho^X}{\pi^X} + \int \rho^X \rho^{Y|X} \log \frac{\rho^{Y|X}}{\pi^{Y|X}}$$

$$= \int \rho^X \log \frac{\rho^X}{\pi^X} + \int \rho^X \rho^{Y|X} \log \frac{\rho^{Y|X}}{\pi^{Y|X}}$$

$$= \mathsf{KL}(\rho^X \,\|\, \pi^X) + \mathbb{E}_{\rho^X}\left[\mathsf{KL}(\rho^{Y|X} \,\|\, \pi^{Y|X})\right].$$

$\qquad\square$

### B.2.2 Hamiltonian Monte Carlo as an LCP scheme

We now describe that HMC is an instance of the LCP scheme on the space of probability distributions. Note one step of HMC from $x_k$ to $x_{k+1}$ can be described as follows:

1. **Lift:** lift $\rho_0^X \in \mathcal{P}(\mathbb{R}^d)$ to $\rho_0(x,y) \propto \rho_0^X(x) \cdot \exp\left(-\frac{1}{2}\|y\|^2\right) \in \mathcal{P}(\mathbb{R}^d \times \mathbb{R}^d)$.

2. **Conserve:** run Hamiltonian flow (HF) from an initial point $(x_0, y_0) \sim \rho_0$ with an integration time $\eta$ to get $(x_\eta, y_\eta) := \mathsf{HF}_\eta(x_0, y_0) \sim \rho_\eta$.

3. **Project:** marginalize $\rho_\eta(x,y)$ to get $\rho_\eta^X(x) := \int \rho_\eta(x,y)\mathrm{d}y$.

In the above, if $x_k \sim \rho_0^X$, then along HMC, $x_{k+1} \sim \rho_\eta^X$ where $\eta = \eta_k$ is the integration time.

In the initial lifting, since $\rho_0^{Y|X} = \tilde{\nu}^{Y|X} = \mathcal{N}(0, I)$, by Lemma 5 we have $\mathsf{KL}(\rho_0^X \,\|\, \nu) = \mathsf{KL}(\rho_0 \,\|\, \tilde{\nu})$. Then we have

$$\mathsf{KL}(\rho_\eta^X \,\|\, \nu) \le \mathsf{KL}(\rho_\eta \,\|\, \tilde{\nu}) = \mathsf{KL}(\rho_0 \,\|\, \tilde{\nu}) = \mathsf{KL}(\rho_0^X \,\|\, \nu),$$

where the first inequality follows from Lemma 5 and the next equality follows from Lemma 4.

This shows HMC is an instance of the LCP scheme, and that it is a descent method in KL divergence.

# C   Optimization and sampling: algorithms and connections

In this section, we review foundational algorithms in convex optimization and log-concave sampling, with a focus on their theoretical convergence behavior and structural similarities. We first review the convergence guarantees of optimization algorithms for minimizing strongly and weakly convex functions. We then review sampling algorithms commonly used for generating samples from log-concave distributions. Based on these algorithms, we draw connections between optimization and sampling, emphasizing both their algorithmic parallels and the fundamental gap in convergence guarantees—most notably, the presence of acceleration in optimization and its absence in current sampling algorithms. These insights motivate our development of randomized Hamiltonian-based methods that aim to unify and extend ideas from both domains.

## C.1   Review of optimization algorithms with convergence rates

In this section, we revisit some classical optimization methods (for minimizing $f$) along with their convergence rates. The classical greedy methods in optimization are *gradient flow* (GF) in continuous-time and *gradient descent* (GD) in discrete-time with stepsize $\eta > 0$, given by

$$\dot{X}_t = -\nabla f(X_t), \tag{GF}$$

$$x_{k+1} = x_k - \eta \nabla f(x_k). \tag{GD}$$

The classical accelerated methods in optimization are *accelerated gradient flow* (AGF) in continuous-time [Su et al., 2016, Wibisono et al., 2016] and *accelerated gradient descent* (AGD) in discrete-time [Nesterov, 1983, 2018] with stepsize $\eta > 0$, given by

$$\ddot{X}_t + \beta(t)\dot{X}_t + \nabla f(X_t) = 0, \tag{AGF}$$

$$\begin{cases} x_{k+1} = y_k - \eta \nabla f(y_k), \\ y_{k+1} = x_{k+1} + \beta_k(x_{k+1} - x_k), \end{cases} \tag{AGD}$$

where $\beta(t) > 0$ is the damping parameter and $\beta_k > 0$ is the momentum parameter.

### C.1.1   Convergence under strong convexity or gradient domination

Now we review the convergence guarantees of the methods mentioned above for strongly convex or gradient-dominated functions. Notably, strong convexity implies gradient domination, and thus any convergence guarantee under gradient domination also holds under strong convexity.

**Continuous-time flows.**

**Theorem 6.** *Assume $f \colon \mathbb{R}^d \to \mathbb{R}$ is $\alpha$-gradient-dominated. Let $X_t \in \mathbb{R}^d$ evolve following (GF) from any $X_0 \in \mathbb{R}^d$. Then for any $t \geq 0$, we have*

$$f(X_t) - f(x^*) \leq \exp(-2\alpha t)(f(X_0) - f(x^*)).$$

*Proof.* Taking the time derivative of $f(X_t) - f(x^*)$, we obtain

$$\frac{\mathrm{d}}{\mathrm{d}t}(f(X_t) - f(x^*)) = \langle \nabla f(X_t), \dot{X}_t \rangle = -\|\nabla f(X_t)\|^2 \leq -2\alpha(f(X_t) - f(x^*)),$$

where the inequality follows from $\alpha$-gradient domination. Applying Grönwall's inequality completes the proof. □

**Theorem 7.** *Assume $f \colon \mathbb{R}^d \to \mathbb{R}$ is $\alpha$-strongly convex. Let $X_t \in \mathbb{R}^d$ evolve following (AGF) with $\beta(t) = 2\sqrt{\alpha}$ from any $X_0 \in \mathbb{R}^d$. Then for any $t \geq 0$, we have*

$$f(X_t) - f(x^*) \leq \exp(-\sqrt{\alpha}t)\left(f(X_0) - f(x^*) + \frac{\alpha}{2}\left\|X_0 + \frac{1}{\sqrt{\alpha}}\dot{X}_0 - x^*\right\|^2\right).$$

*Proof.* Define the continous-time Lyapunov function:

$$\mathcal{E}_t = \exp(\sqrt{\alpha}t)\left(f(X_t) - f(x^*) + \frac{\alpha}{2}\left\|X_t + \frac{1}{\sqrt{\alpha}}\dot{X}_t - x^*\right\|^2\right).$$

We can show $\dot{\mathcal{E}}_t \leq 0$ along (AGF) with $\beta(t) = 2\sqrt{\alpha}$ for $t \geq 0$. For a complete proof, see the proof of Wilson et al. [2021, Proposition 3]. $\qquad \square$

**Remark 1.** *Comparing Theorem 6 and Theorem 7, we observe that* (GF) *converges at a rate of* $\exp(-2\alpha t)$ *under $\alpha$-gradient domination, which also holds under $\alpha$-strong convexity. In contrast,* (AGF) *achieves a faster rate of* $\exp(-\sqrt{\alpha} t)$ *under $\alpha$-strong convexity. Since $\sqrt{\alpha} > 2\alpha$ for small $\alpha < \frac{1}{4}$, this demonstrates the acceleration effect of* (AGF) *compared to* (GF).

### Discrete-time algorithms.

**Theorem 8.** *Assume $f \colon \mathbb{R}^d \to \mathbb{R}$ is $\alpha$-gradient-dominated and $L$-smooth. Then for all $k \geq 0$, along* (GD) *with $\eta \leq \frac{1}{L}$ from any $x_0 \in \mathbb{R}^d$, we have*

$$f(x_k) - f(x^*) \leq (1 - \alpha\eta)^k (f(x_0) - f(x^*)).$$

*Proof.* Define the discrete-time Lyapunov function: $E_k = (1-\alpha\eta)^{-k}(f(x_k)-f(x^*))$. We can show $E_{k+1} \leq E_k$ for all $k \geq 0$ along (GD). See Wilson [2018, Appendix B.1.1] for more details. $\qquad \square$

**Corollary 3.** *Assume $f \colon \mathbb{R}^d \to \mathbb{R}$ is $\alpha$-gradient-dominated and $L$-smooth. To generate $x_K$ satisfying $f(x_K) - f(x^*) \leq \varepsilon$, it suffices to run* (GD) *with $\eta = \frac{1}{L}$ and from any $x_0 \in \mathbb{R}^d$ for*

$$K \geq \kappa \cdot \log\left(\frac{f(x_0) - f(x^*)}{\varepsilon}\right).$$

**Theorem 9.** *Assume $f \colon \mathbb{R}^d \to \mathbb{R}$ is $\alpha$-strongly convex and $L$-smooth. Then for all $k \geq 0$, along* (AGD) *with $\beta_k = \frac{1-\sqrt{\alpha\eta}}{1+\sqrt{\alpha\eta}}$, $\eta \leq \frac{1}{L}$ from any $x_0 = y_0 \in \mathbb{R}^d$, we have*

$$f(x_k) - f(x^*) \leq (1 - \sqrt{\alpha\eta})^k \left(f(x_0) - f(x^*) + \frac{\alpha}{2}\|x_0 - x^*\|^2\right).$$

*Proof.* A complete proof can be found in the proof of d'Aspremont et al. [2021, Corollary 4.1.5]. $\qquad \square$

**Corollary 4.** *Assume $f \colon \mathbb{R}^d \to \mathbb{R}$ is $\alpha$-strongly convex and $L$-smooth. To generate $x_K$ satisfying $f(x_K) - f(x^*) \leq \varepsilon$, it suffices to run* (AGD) *with $\beta_k = \frac{\sqrt{\kappa}-1}{\sqrt{\kappa}+1}$, $\eta = \frac{1}{L}$ and from any $x_0$, $y_0 \in \mathbb{R}^d$ for*

$$K \geq \sqrt{\kappa} \cdot \log\left(\frac{f(x_0) - f(x^*) + \frac{\alpha}{2}\|x_0 - x^*\|^2}{\varepsilon}\right).$$

**Remark 2.** *Comparing Corollary 3 and Corollary 4, we observe that* (AGD) *improves upon* (GD) *by reducing the iteration complexity from $O(\kappa \log(1/\varepsilon))$ to $O(\sqrt{\kappa} \log(1/\varepsilon))$, where $\kappa = L/\alpha$ is the condition number. This demonstrates the acceleration effect of* (AGD) *over* (GD) *for strongly convex functions.*

### C.1.2  Convergence under weak convexity

Now we review convergence guarantees of the aforementioned methods for weakly convex functions.

### Continuous-time flows.

**Theorem 10.** *Assume $f \colon \mathbb{R}^d \to \mathbb{R}$ is weakly convex. Let $X_t \in \mathbb{R}^d$ evolve following* (GF) *from any $X_0 \in \mathbb{R}^d$. Then for any $t \geq 0$, we have*

$$f(X_t) - f(x^*) \leq \frac{\|X_0 - x^*\|^2}{2t}.$$

*Proof.* Define the continuous-time Lyapunov function: $\mathcal{E}_t = \frac{1}{2}\|X_t - x^*\|^2 + t(f(X_t) - f(x^*))$. We can show $\dot{\mathcal{E}}_t \leq 0$ along (GF) for $t \geq 0$. See Wilson [2018, Appendix B.2.2] for more details. $\qquad \square$

**Theorem 11.** *Assume $f \colon \mathbb{R}^d \to \mathbb{R}$ is weakly convex. Let $X_t \in \mathbb{R}^d$ evolve following* (AGF) *with $\beta(t) = \frac{3}{t}$ from any $X_0 \in \mathbb{R}^d$ and $\dot{X}_0 \in \mathbb{R}^d$. Then for any $t \geq 0$, we have*

$$f(X_t) - f(x^*) \leq \frac{2\|X_0 - x^*\|^2}{t^2}.$$

*Proof.* Define the continuous-time Lyapunov function:

$$\mathcal{E}_t = t^2(f(X_t) - f(x^*)) + 2\left\|X_t + \frac{t}{2}\dot{X}_t - x^*\right\|^2.$$

We can show $\dot{\mathcal{E}}_t \leq 0$ along (AGF) with $\beta(t) = 3/t$ for $t \geq 0$. A complete proof can be found in the proof of Su et al. [2016, Theorem 3]. $\qquad\square$

**Remark 3.** *Comparing Theorem 10 and Theorem 11, we observe that* (GF) *achieves a convergence rate of $O(1/t)$, while* (AGF) *improves this to $O(1/t^2)$. This confirms the acceleration effect of* (AGF) *over* (GF) *for weakly convex functions.*

**Discrete-time algorithms.**

**Theorem 12.** *Assume $f : \mathbb{R}^d \to \mathbb{R}$ is weakly convex and $L$-smooth. Then for all $k \geq 0$, along* (GD) *with $\eta \leq \frac{1}{L}$ from any $x_0 \in \mathbb{R}^d$, we have*

$$f(x_k) - f(x^*) \leq \frac{\|x_0 - x^*\|^2}{2\eta k}.$$

*Proof.* Define the discrete-time Lyapunov function: $E_k = \frac{1}{2}\|x_k - x^*\|^2 + \eta k(f(x_k) - f(x^*))$. We can show $E_{k+1} \leq E_k$ along (GD) for all $k \geq 0$. A complete proof can be found in Wilson [2018, Section 2.1.2]. $\qquad\square$

**Corollary 5.** *Assume $f : \mathbb{R}^d \to \mathbb{R}$ is weakly convex and $L$-smooth. To generate $x_K$ satisfying $f(x_K) - f(x^*) \leq \varepsilon$, it suffices to run* (GD) *with $\eta = \frac{1}{L}$ from any $x_0 \in \mathbb{R}^d$ for*

$$K \geq \frac{L\|x_0 - x^*\|^2}{\varepsilon}.$$

**Theorem 13.** *Assume $f : \mathbb{R}^d \to \mathbb{R}$ is weakly convex and $L$-smooth. Then for all $k \geq 0$, along* (AGD) *with $\beta_k = \frac{k-1}{k+2}$, $\eta \leq \frac{1}{L}$ from any $x_0$, $y_0 \in \mathbb{R}^d$, we have*

$$f(x_k) - f(x^*) \leq \frac{2\|x_0 - x^*\|^2}{\eta k^2}.$$

*Proof.* A complete proof can be found in the proof of Su et al. [2016, Theorem 6]. $\qquad\square$

**Corollary 6.** *Assume $f : \mathbb{R}^d \to \mathbb{R}$ is weakly convex and $L$-smooth. To generate $x_K$ satisfying $f(x_K) - f(x^*) \leq \varepsilon$, it suffices to run* (AGD) *with $\beta_k = \frac{k-1}{k+2}$, $\eta = \frac{1}{L}$ and any $x_0$, $y_0 \in \mathbb{R}^d$ for*

$$K \geq \sqrt{\frac{2L\|x_0 - x^*\|^2}{\varepsilon}}.$$

**Remark 4.** *Comparing Corollary 5 and Corollary 6, we observe that to reach an $\varepsilon$-accurate solution,* (GD) *requires $O(L/\varepsilon)$ iterations, while* (AGD) *only needs $O(\sqrt{L/\varepsilon})$ iterations. This confirms the accelerated convergence of* (AGD) *for smooth and weakly convex functions.*

### C.2  Review of sampling algorithms

A natural greedy dynamics for sampling $\nu \propto e^{-f}$ is the *Langevin dynamics* (LD) in continuous-time:

$$\mathrm{d}X_t = -\nabla f(X_t)\mathrm{d}t + \sqrt{2}\mathrm{d}\mathrm{B}_t, \tag{LD}$$

where $\mathrm{B}_t$ is the standard Brownian motion. A simple discretization of (LD) is called the *unadjusted Langevin algorithm* (ULA):

$$x_{k+1} = x_k - \eta\nabla f(x_k) + \sqrt{2\eta}\xi_k, \tag{ULA}$$

where $\xi_k \sim \mathcal{N}(0, I)$ is an independent Gaussian noise, and $\eta > 0$ is stepsize. However, (ULA) is a biased algorithm, which means for each fixed step size $\eta$, it converges to a biased limiting distribution [Roberts and Tweedie, 1996]. There have been many results on the biased convergence guarantee of ULA [Dalalyan, 2017a,b, Durmus and Moulines, 2017, Cheng and Bartlett, 2018,

Durmus et al., 2019, Vempala and Wibisono, 2019, Chewi et al., 2024], but due to the bias, it does not have a matching convergence rate with the continuous-time Langevin dynamics.

An alternative, *unbiased* discretization of (LD) is the *Proximal Sampler* (Prox-S) [Lee et al., 2021]. Given the stepsize $\eta > 0$, the update is given by:

$$\begin{cases} y_k \mid x_k \sim \tilde{\nu}^{Y|X}(\cdot \mid x_k) = \mathcal{N}(x_k, \eta I), \\ x_{k+1} \mid y_k \sim \tilde{\nu}^{X|Y}(\cdot \mid y_k), \end{cases} \quad \text{(Prox-S)}$$

where $\tilde{\nu}(x, y) \propto \exp\left(-f(x) - \frac{1}{2\eta}\|x - y\|^2\right)$. The Proximal Sampler has been shown to have convergence guarantees that match the Langevin dynamics convergence rates from continuous time; see e.g. Lee et al. [2021], Chen et al. [2022], Mitra and Wibisono [2025], Wibisono [2025]. Note that HMC (Algorithm 4) with $\eta_k = h$ is also a greedy sampling method in discrete-time assuming we can simulate (HF) exactly.

Accelerated methods for sampling remain an active area of research. A natural candidate is the *underdamped Langevin dynamics* (ULD):

$$\begin{cases} \mathrm{d}X_t = Y_t \, \mathrm{d}t, \\ \mathrm{d}Y_t = -\beta Y_t \, \mathrm{d}t - \nabla f(X_t) \, \mathrm{d}t + \sqrt{2\beta} \, \mathrm{d}B_t, \end{cases} \quad \text{(ULD)}$$

where $\beta > 0$ is the damping parameter. Lu and Wang [2022] show that (ULD) has accelerated convergence rate in $\chi^2$-divergence in continuous time. However, establishing similar acceleration in KL divergence remains challenging. Furthermore, standard discretizations of (ULD) suffer from being biased [Ma et al., 2021, Zhang et al., 2023] and thus do not exhibit acceleration.

Another candidate to achieve the same acceleration as (AGF) in KL divergence and 2-Wasserstein distance is the *accelerated information gradient flow* (AIG) [Wang and Li, 2022, Chen et al., 2025]:

$$\begin{cases} \dot{X}_t = Y_t, \\ \dot{Y}_t = -\beta Y_t - \nabla f(X_t) - \nabla \log \mu_t(X_t), \end{cases} \quad \text{(AIG)}$$

where $\mu_t$ is the law of $X_t$. However, discretizing the AIG flow as a concrete algorithm poses significant challenges: the implementation is hindered by the unknown score function $\nabla \log \mu_t$, and the analysis is further complicated by the non-smoothness property of the entropy functional under the Wasserstein metric. Thus, both the development of accelerated sampling methods and the improved analysis of existing methods to show acceleration are still largely open.

## C.3 Connection and comparison between optimization and sampling

Langevin dynamics (LD) can be interpreted as the Wasserstein gradient flow of the Kullback–Leibler (KL) divergence. More precisely, consider the target distribution $\nu(x) \propto \exp(-f(x))$. Then the Fokker-Planck equation of (LD), which corresponds to the evolution of the law of (LD) is

$$\partial_t \mu_t = \nabla \cdot \left(\mu_t \nabla \log \frac{\mu_t}{\nu}\right),$$

where $\mu_t = \mathsf{Law}(X_t)$ and $\nabla_{\mathcal{W}_2} \mathsf{KL}(\mu \,\|\, \nu) = -\nabla \cdot \left(\mu \nabla \log \frac{\mu}{\nu}\right)$ is the Wasserstein gradient of the functional $\mathsf{KL}(\mu\|\nu)$ with respect to the 2-Wasserstein metric in the space of probability distributions $\mathcal{P}_2(\mathbb{R}^d)$. This interpretation, originally developed by Jordan et al. [1998] and later formalized in the language of optimal transport by Otto [2001], provides a variational characterization of Langevin dynamics and connects sampling algorithms with the theory of gradient flows in metric spaces; see also [Wibisono, 2018] for a discussion on the algorithmic guarantees.

We summarize the convergence and mixing rates of the aforementioned optimization and sampling methods in Table 1 under the following assumptions:

- **(SC)** $f$ is $\alpha$-strongly convex $\iff \nu$ is $\alpha$-strongly log-concave $\iff \mathsf{KL}(\cdot \,\|\, \nu)$ is $\alpha$-strongly convex.
- **(WC)** $f$ is weakly convex $\iff \nu$ is weakly log-concave $\iff \mathsf{KL}(\cdot \,\|\, \nu)$ is weakly convex.
- **(Sm)** $f$ is $L$-smooth $\iff \nu$ is $L$-log-smooth.

| Method | Assumption | Optimization | Sampling |
|--------|------------|--------------|----------|
| Greedy (cont.) | **(WC)** | GF: $1/t$ | LD: $1/t$ |
|  | **(SC)** | GF: $\exp(-2\alpha t)$ | LD: $\exp(-2\alpha t)$ |
| Greedy (disc.) | **(WC)+(Sm)** | GD: $1/(\eta k)$ | Prox-S: $1/(\eta k)$ |
|  | **(SC)+(Sm)** | GD: $(1-\alpha\eta)^k$ | HMC: $\left(1-\alpha h^2/4\right)^k$ |
| Accel. (cont.) | **(WC)** | AGF: $1/t^2$ | AIG: $1/t^2$ |
|  | **(SC)** | AGF: $\exp(-\sqrt{\alpha}t)$ | AIG: $\exp(-\sqrt{\alpha}t)$ |
| Accel. (disc.) | **(WC)+(Sm)** | AGD: $1/(\eta k^2)$ | —: *missing* |
|  | **(SC)+(Sm)** | AGD: $(1-\sqrt{\alpha\eta})^k$ | —: *missing* |

Table 1: Comparison of convergence rates for optimization and sampling methods under different assumptions omitting constants, where $t$ denotes the continuous time; $k$ denotes the iteration count; $\eta$ denotes the stepsize and $h$ denotes integration time of HMC.

Under Assumption **(A1)**, the convergence rates are given in squared distance for optimization (e.g., $\|x_k - x^*\|^2$) and 2-Wasserstein distance for sampling (e.g., $\mathcal{W}_2^2(\mu_k, \nu)$). Under Assumption **(A2)**, the convergence rates are given in sub-optimality gap for optimization (e.g. $f(x_k) - f(x^*)$) and KL divergence for sampling (e.g., $\mathsf{KL}(\mu_k\|\nu)$).

Table 1 highlights the structural parallels between optimization and sampling methods under various convexity and smoothness assumptions. In both continuous and discrete-time settings, classical greedy methods exhibit matching convergence behavior: gradient flow (GF) corresponds to Langevin dynamics (LD), and gradient descent (GD) aligns with the proximal sampler (Prox-S) or Hamiltonian Monte Carlo (HMC). Similarly, accelerated gradient flow (AGF) and the accelerated information flow (AIG) achieve analogous accelerated rates in continuous time. However, an essential gap emerges in the discrete-time setting: while accelerated optimization methods such as accelerated gradient descent (AGD) are well established and enjoy fast convergence rates, their counterparts in sampling are notably absent.

## D   Hamiltonian flow with short-time integration

In this section, we show that HF-opt (Algorithm 1) achieves the non-accelerated convergence rate with short integration time $\eta_k$ for gradient-dominated and weakly convex functions. These rates matches the convergence rate of gradient descent (GD) (with $\eta = h^2$) under the same assumptions. We first show a descent lemma for HF-opt similar to that of GD.

**Lemma 6.** *Assume $f$ is $L$-smooth. Then Algorithm 1 with $\eta_k = h \le \frac{1}{\sqrt{L}}$ satisfies*

$$f(x_{k+1}) \le f(x_k) - \frac{h^2}{4}\|\nabla f(x_k)\|^2.$$

Before proving Lemma 6, we introduce the following lemmas that are useful to the proof.

**Lemma 7** (Lee et al. [2018] Lemma A.5). *Given a continuous function $y(t)$ and positive scalars $m$, $L$ such that $0 \le y(t) \le m + L\int_0^t(t-s)y(s)\,\mathrm{d}s$, we have $y(t) \le m\cosh(\sqrt{L}t)$.*

We can show the following bound on the velocity along Hamiltonian flow.

**Lemma 8.** *Assume $f$ is $L$-smooth. Then along (HF) from any $X_0 \in \mathbb{R}^d$ with $Y_0 = 0$, we have*

$$\|Y_t\| \le t\|\nabla f(X_0)\|\cosh(\sqrt{L}t).$$

*Proof.* Note that along (HF), we have

$$\frac{\mathrm{d}}{\mathrm{d}t}\|Y_t\| = \frac{-\langle Y_t, \nabla f(X_t)\rangle}{\|Y_t\|} \leq \frac{\|Y_t\|\|\nabla f(X_t)\|}{\|Y_t\|} = \|\nabla f(X_t)\|. \tag{9}$$

Furthermore, we have

$$\frac{\mathrm{d}}{\mathrm{d}t}\|\nabla f(X_t)\| = \frac{\langle \nabla f(X_t), \nabla^2 f(X_t)Y_t\rangle}{\|\nabla f(X_t)\|} \leq \|\nabla^2 f(X_t)Y_t\| \leq L\|Y_t\|. \tag{10}$$

Integrating (10) yields

$$\|\nabla f(X_t)\| \leq \|\nabla f(X_0)\| + L\int_0^t \|Y_s\|\,\mathrm{d}s. \tag{11}$$

Integrating (9) and applying (11) yields

$$\|Y_t\| \leq \|Y_0\| + \int_0^t \|\nabla f(X_s)\|\,\mathrm{d}s \leq t\|\nabla f(X_0)\| + L\int_0^t (t-s)\|Y_s\|\,\mathrm{d}s.$$

Suppose $t \leq h$. Then we have $\|Y_t\| \leq h\|\nabla f(X_0)\| + L\int_0^t (t-s)\|Y_s\|\,\mathrm{d}s$. By Lemma 7, we conclude that for $0 \leq t \leq h$:

$$\|Y_t\| \leq h\|\nabla f(X_0)\|\cosh(\sqrt{L}t).$$

Choosing $t = h$ completes the proof. $\qquad\square$

We can also show the following lower bound on velocity along Hamiltonian flow for short time.

**Lemma 9.** *Assume $f$ is $L$-smooth. Then along* (HF) *with $0 \leq t \leq \frac{1}{\sqrt{L}}$ from any $X_0 \in \mathbb{R}^d$ with $Y_0 = 0$, we have $\|Y_t\| \geq \frac{t}{\sqrt{2}}\|\nabla f(X_0)\|$.*

*Proof.* Note that along (HF), we have

$$\dot{X}_t = Y_t, \quad \dot{Y}_t = -\nabla f(X_t), \quad \ddot{Y}_t = -\nabla^2 f(X_t)\,Y_t.$$

Therefore, by Taylor expansion around $t = 0$, we have

$$Y_t = Y_0 + t\dot{Y}_0 + R_t = -t\nabla f(X_0) + R_t,$$

where $R_t = Y_t - (Y_0 + t\dot{Y}_0)$ is the remainder term which can be written as

$$R_t = \int_0^t \int_0^s \ddot{Y}_r\,\mathrm{d}r\,\mathrm{d}s = \int_0^t (t-s)\ddot{Y}_s\,\mathrm{d}s = -\int_0^t (t-s)\nabla^2 f(X_s)\,Y_s\,\mathrm{d}s.$$

Thus we have

$$\|R_t\|_2 \leq \int_0^t (t-s)\left\|\nabla^2 f(X_s)\,Y_s\right\|\,\mathrm{d}s \leq L\int_0^t (t-s)\|Y_s\|\,\mathrm{d}s$$

Applying Lemma 8 with $\|Y_t\| \leq t\|\nabla f(X_0)\|\cosh(\sqrt{L}t)$, we obtain

$$\|R_t\| \leq L\int_0^t (t-s)s\|\nabla f(X_0)\|\cosh(\sqrt{L}s)\,\mathrm{d}s$$

$$\leq L\|\nabla f(X_0)\|\cosh(\sqrt{L}t)\int_0^t (t-s)s\,\mathrm{d}s$$

$$\leq \frac{1}{6}t^3 L\|\nabla f(X_0)\|\cosh(\sqrt{L}t).$$

If $t \leq \frac{1}{\sqrt{L}}$, then we have

$$t^2 L\cosh(\sqrt{L}t) \leq \cosh(1) \leq 6\left(1 - \frac{1}{\sqrt{2}}\right).$$

Therefore, by triangle inequality,

$$\|Y_t\| \geq t\|\nabla f(X_0)\| - \|R_t\|$$
$$\geq t\|\nabla f(X_0)\| - \frac{1}{6}t^3 L\|\nabla f(X_0)\|\cosh(\sqrt{L}t)$$
$$= t\left(1 - \frac{1}{6}t^2 L\cosh(\sqrt{L}t)\right)\|\nabla f(X_0)\|$$
$$\geq \frac{t}{\sqrt{2}}\|\nabla f(X_0)\|.$$

$\square$

**Proof of Lemma 6.** We are now ready to prove Lemma 6.

*Proof.* Let $(x_{k+1}, y_{k+1}) = \mathsf{HF}_h(x_k, 0)$ be the solution of (HF) at time $h$ starting from $(x_k, 0)$. By Lemma 1, we have $f(x_{k+1}) = f(x_k) - \frac{1}{2}\|y_{k+1}\|^2$. Applying Lemma 9 with $h \leq \frac{1}{\sqrt{L}}$, we obtain

$$f(x_{k+1}) \leq f(x_k) - \frac{h^2}{4}\|\nabla f(x_k)\|^2.$$

$\square$

### D.1 Proof of Theorem 1

**Theorem 1.** *Assume $f$ is $L$-smooth. Along Algorithm 1 with $\eta_k = h \leq \frac{1}{\sqrt{L}}$, from any $x_0 \in \mathbb{R}^d$:*

*1. If $f$ is $\alpha$-gradient-dominated, then $f(x_k) - f(x^*) \leq \left(1 - \frac{1}{2}\alpha h^2\right)^k (f(x_0) - f(x^*))$.*

*2. If $f$ is weakly convex, then $f(x_k) - f(x^*) \leq \frac{34\|x_0 - x^*\|^2}{h^2 k}$.*

*Proof.* If $f$ is $\alpha$-gradient-dominated, using Lemma 6 yields

$$f(x_{k+1}) \leq f(x_k) - \frac{\alpha h^2}{2}(f(x_k) - f(x^*)) \iff f(x_{k+1}) - f(x^*) \leq \left(1 - \frac{\alpha h^2}{2}\right)(f(x_k) - f(x^*)),$$

which implies

$$f(x_k) - f(x^*) \leq \left(1 - \frac{\alpha h^2}{2}\right)^k (f(x_0) - f(x^*)).$$

Theorem 2 in Wilson et al. [2019] establish the convergence rates for weakly convex functions based on a descent lemma similar to Lemma 6. Thus we directly invoke that theorem to prove our Theorem 1. If $f$ is weakly convex and $h \leq \frac{1}{\sqrt{L}}$, then Theorem 2 in Wilson et al. [2019] implies

$$f(x_k) - f(x^*) \leq \frac{32\|x_0 - x^*\|^2 + 4h^2(f(x_0) - f(x^*))}{h^2 k} \leq \frac{(32 + 2Lh^2)\|x_0 - x^*\|^2}{h^2 k}$$
$$\leq \frac{34\|x_0 - x^*\|^2}{h^2 k}.$$

$\square$

### D.2 Implementation of HF-opt via leapfrog integrator as GD

Let $\mathsf{Leapfrog}_h : \mathbb{R}^d \times \mathbb{R}^d \to \mathbb{R}^d \times \mathbb{R}^d$ denote the leapfrog integrator map with stepsize $h \geq 0$, which is a numerical approximation of $\mathsf{HF}_h$. More specifically, if we denote $(x_{k+1}, y_{k+1}) = \mathsf{Leapfrog}_h(x_k, y_k)$, then it satisfies

$$y_{k+\frac{1}{2}} = y_k - \frac{h}{2}\nabla f(x_k), \tag{12a}$$

$$x_{k+1} = x_k + h y_{k+\frac{1}{2}}, \tag{12b}$$

$$y_{k+1} = y_{k+\frac{1}{2}} - \frac{h}{2} \nabla f(x_{k+1}). \tag{12c}$$

Replacing $y_{k+\frac{1}{2}}$ in (12b) and (12c) with with (12a), we can rewrite (12) as

$$x_{k+1} = x_k + h y_k - \frac{h^2}{2} \nabla f(x_k), \tag{13a}$$

$$y_{k+1} = y_k - \frac{h}{2} \left( \nabla f(x_k) + \nabla f(x_{k+1}) \right). \tag{13b}$$

Thus if we replace $\mathsf{HF}_{\eta_k}(x_k, 0)$ in Algorithm 1 (HF-opt) with $\mathsf{Leapfrog}_{\eta_k}(x_k, 0)$, we obtain

$$x_{k+1} = x_k - \frac{\eta_k^2}{2} \nabla f(x_k),$$

which is equivalent to gradient descent with stepsize $\eta_k^2/2$. Note that we refresh $y_{k+1}$ to be zero before the next iteration, and thus we only focus on the update of $x$. This shows that we can view gradient descent not only as a discretization of gradient flow, but also as a discretization of HF-opt with a one-step leapfrog integrator.

## E   Examples of HF-opt and RHF-opt on quadratic functions

**Example 1** (Quadratic functions)**.** *Consider the quadratic function $f(x) = \frac{1}{2} x^\top A x$, where $A \in \mathbb{R}^{d \times d}$ is symmetric and $\alpha I \preceq A \preceq L I$, then* HF-opt *with $\eta_k = h \leq \frac{1}{2\sqrt{L}}$ satisfies*

$$\|x_k - x^*\|^2 \leq \left( 1 - \frac{\alpha h^2}{4} \right)^k \|x_0 - x^*\|^2.$$

*If $h = \frac{1}{2\sqrt{L}}$, the total integration time to achieve $\|x_K - x^*\|^2 \leq \varepsilon$ satisfies $K \cdot h \geq \frac{8\sqrt{L}}{\alpha} \log \frac{\|x_0 - x^*\|^2}{\varepsilon}$.*

**Example 2** (RHF-opt for quadratic functions)**.** *Consider the quadratic function $f(x) = \frac{1}{2} x^\top A x$, where $A \in \mathbb{R}^{d \times d}$ is symmetric and $\alpha I \preceq A \preceq L I$, then* RHF-opt *with $\gamma(t) = \gamma > 0$ satisfies*

$$\mathbb{E} \left[ \|x_k - x^*\|^2 \right] \leq \left( 1 - \frac{2\alpha}{\gamma^2 + 4\alpha} \right)^k \mathbb{E} \left[ \|x_0 - x^*\|^2 \right].$$

*If $\gamma = 2\sqrt{\alpha}$, the expected total integration time to achieve $\mathbb{E} \left[ \|x_K - x^*\|^2 \right] \leq \varepsilon$ is $K \cdot \mathbb{E}[\tau_k] \geq \frac{2}{\sqrt{\alpha}} \log \frac{\mathbb{E}\left[ \|x_0 - x^*\|^2 \right]}{\varepsilon}$, matching that of accelerated gradient flow with refreshment and* HF-opt *with Chebyshev-based integration time [Wang, 2024]. Note that the expected total integration time of* RHF-opt *is smaller than the total integration time of* HF-opt *shown in Example 1.*

Before proving the conclusions in Examples 1 and 2, we first invoke two lemmas from Wang [2024]:

**Lemma 10** (Wang [2024] Lemma 2.2)**.** *For quadratic function $f(x) = \frac{1}{2} x^\top A x$,* RHF-opt *(Algorithm 2) satisfies*

$$x_K - x^* = \left( \prod_{k=0}^{K-1} \cos(\eta_k \sqrt{A}) \right) (x_0 - x^*),$$

*where $\sqrt{A}$ is the matrix square root of $A$, i.e., $\sqrt{A}\sqrt{A} = A$.*

**Lemma 11** (Wang [2024] Lemma A.4)**.** *The eigenvalues of the matrix $\prod_{k=1}^{K} \cos\left( \eta_k \sqrt{A} \right)$ are $\lambda_j := \prod_{k=0}^{K-1} \cos\left( \eta_k \sqrt{\sigma_j} \right)$, $j \in [d]$, where $\sigma_1, \sigma_2, \ldots, \sigma_d$ are the eigenvalues of $A$.*

**Proof of Example 1**

*Proof.* Using Lemmas 10 and 19, if $\eta_k = h \le \frac{1}{2\sqrt{\sigma_d}} = \frac{1}{2\sqrt{L}}$. we obtain

$$
\begin{aligned}
\|x_K - x^*\|^2 &= \left\| \left( \prod_{k=0}^{K-1} \cos(h\sqrt{A}) \right) (x_0 - x^*) \right\|^2 \\
&\le \left( \max_{j \in [d]} \prod_{k=0}^{K-1} \cos^2(h\sqrt{\sigma_j}) \right) \|x_0 - x^*\|^2 \\
&\le \left( 1 - \frac{\alpha h^2}{4} \right)^K \|x_0 - x^*\|^2.
\end{aligned}
$$

Thus the total integration time to generate a solution satisfying $\|x_K - x^*\|^2 \le \varepsilon$ is

$$
h \cdot K = h \cdot \frac{4}{\alpha h^2} \cdot \log \frac{1}{\varepsilon} = \frac{4}{\alpha h} \cdot \log \frac{1}{\varepsilon} \ge \frac{8\sqrt{L}}{\alpha} \cdot \log \frac{1}{\varepsilon},
$$

where the equality holds when $h = \frac{1}{2\sqrt{L}}$. $\qquad\square$

**Lemma 12.** *If $\tau \sim \mathrm{Exp}(\gamma)$, then we have $\mathbb{E}[\cos^2(\sqrt{\sigma}\tau)] = 1 - \frac{2\sigma}{\gamma^2 + 4\sigma}$.*

*Proof.* See the proof of Proposition 2 in Jiang [2023]. $\qquad\square$

**Proof of Example 2**

*Proof.* Using Lemma 10 and taking the expectation, we obtain

$$
\begin{aligned}
\mathbb{E}\|x_K - x^*\|^2 &= \mathbb{E} \left\| \left( \prod_{k=0}^{K-1} \cos(\tau_k \sqrt{A}) \right) (x_0 - x^*) \right\|^2 \\
&\le \left( \max_{j \in [d]} \prod_{k=0}^{K-1} \mathbb{E}[\cos^2(\tau_k \sqrt{\sigma_j})] \right) \mathbb{E}\|x_0 - x^*\|^2 \\
&= \left( 1 - \min_{j \in [d]} \frac{2\sigma_j}{\gamma^2 + 4\sigma_j} \right)^K \mathbb{E}\|x_0 - x^*\|^2 \\
&\le \exp \left( -\min_{j \in [d]} \frac{2\sigma_j K}{\gamma^2 + 4\sigma_j} \right) \mathbb{E}\|x_0 - x^*\|^2,
\end{aligned}
$$

where the second equality follows from Lemma 12. Thus the expected total integration time is

$$
\mathbb{E}[\tau_k] \cdot \max_j \frac{\gamma^2 + 4\sigma_j}{2\sigma_j} \cdot \log \frac{1}{\varepsilon} = \frac{1}{\gamma} \cdot \max_{j \in [d]} \frac{\gamma^2 + 4\sigma_j}{2\sigma_j} \cdot \log \frac{1}{\varepsilon} = \left( \frac{\gamma}{2\alpha} + \frac{2}{\gamma} \right) \cdot \log \frac{1}{\varepsilon} \ge \frac{2}{\sqrt{\alpha}} \cdot \log \frac{1}{\varepsilon},
$$

where equality holds when $\gamma = 2\sqrt{\alpha}$. $\qquad\square$

# F Convergence analysis of the randomized Hamiltonian flow

Let $\Pi_Y$ denote the operator that maps a function $\phi \colon \mathbb{R}^d \times \mathbb{R}^d \to \mathbb{R}$ to $\Pi_Y \phi \colon \mathbb{R}^d \times \mathbb{R}^d \to \mathbb{R}$ given by $(\Pi_Y \phi)(x, y) = \phi(x, 0)$. Then we provide the following continuity equation for any smooth test functions along the randomized Hamiltonian flow (RHF) defined in Section 3, which is the essential tool for establishing its convergence.

**Lemma 13.** *Let $Z_t = (X_t, Y_t) \sim \rho_t^Z$ evolve following* RHF. *For any smooth $\phi_t \colon \mathbb{R}^{2d} \to \mathbb{R}$,*

$$
\frac{\mathrm{d}}{\mathrm{d}t} \mathbb{E}[\phi_t(Z_t)] = \mathbb{E} \left[ \frac{\partial \phi_t}{\partial t}(Z_t) + \langle \nabla \phi_t(Z_t), b(Z_t) \rangle + \gamma(t)((\Pi_Y \phi_t)(Z_t) - \phi_t(Z_t)) \right], \quad (14)
$$

*where $b(Z_t) = (Y_t, -\nabla f(X_t))$ is the Hamiltonian vector field.*

*Proof.* The time derivative can be expressed as taking the limit:

$$\frac{\mathrm{d}}{\mathrm{d}t}\mathbb{E}[\phi_t(Z_t)] = \lim_{h\to 0}\frac{\mathbb{E}[\phi_{t+h}(Z_{t+h}) - \phi_t(Z_t)]}{h}.$$

Thus we first study the short time behavior of RHF. Using the interpretation at the beginning of Section 4, the probability of a refreshment occurring in a small interval with width $h$ is close to $\gamma(t)h$. Thus the RHF from $t$ to $t + h$ can be approximated by the following procedure for small $h$:

- Run (HF) with deterministic integration time $h$: $\tilde{Z}_{t+h} = (X_{t+h}, \tilde{Y}_{t+h}) = \mathsf{HF}_h(Z_t)$.

- Accept-refresh: $Z_{t+h} = \begin{cases} \tilde{Z}_{t+h} & \text{with probability } 1 - \gamma(t)h, \\ (X_{t+h}, 0) & \text{with probability } \gamma(t)h. \end{cases}$

We can decompose $\mathbb{E}[\phi_{t+h}(Z_{t+h}) - \phi_t(Z_t)]$ as

$$\mathbb{E}[\phi_{t+h}(Z_{t+h}) - \phi_t(Z_t)] = \mathbb{E}[\phi_{t+h}(Z_{t+h}) - \phi_t(Z_{t+h})] + \mathbb{E}[\phi_t(\tilde{Z}_{t+h}) - \phi_t(Z_t)]$$
$$+ \mathbb{E}[\phi_t(Z_{t+h}) - \phi_t(\tilde{Z}_{t+h})].$$

For the first two terms on the right hand side, we perform Taylor expansion and obtain

$$\mathbb{E}[\phi_{t+h}(Z_{t+h}) - \phi_t(Z_{t+h})] = \mathbb{E}\left[h\frac{\partial\phi_t}{\partial t}(Z_{t+h}) + O(h^2)\right],$$

$$\mathbb{E}[\phi_t(\tilde{Z}_{t+h}) - \phi_t(Z_t)] = \mathbb{E}[h\langle\nabla\phi_t(Z_t), b(Z_t)\rangle + O(h^2)].$$

For the last term, we obtain by the probabilistic accept-refresh step

$$\mathbb{E}[\phi_t(Z_{t+h}) - \phi_t(\tilde{Z}_{t+h})] = \mathbb{E}\left[(1 - \gamma(t)h)\cdot\phi_t(\tilde{Z}_{t+h}) + \gamma(t)h\cdot(\Pi_Y\phi_t)(\tilde{Z}_{t+h}) - \phi_t(\tilde{Z}_{t+h})\right]$$

$$= \gamma(t)h\cdot\mathbb{E}\left[(\Pi_Y\phi_t)(\tilde{Z}_{t+h}) - \phi_t(\tilde{Z}_{t+h})\right].$$

Thus we obtain

$$\frac{\mathrm{d}}{\mathrm{d}t}\mathbb{E}_{\rho_t^Z}[\phi_t(Z_t)] = \lim_{h\to 0}\frac{\mathbb{E}_{\rho_t^Z}[\phi_{t+h}(Z_{t+h}) - \phi_t(Z_t)]}{h}$$

$$= \lim_{h\to 0}\frac{\mathbb{E}_{\rho_t^Z}\left[h\frac{\partial\phi_t}{\partial t}(Z_{t+h}) + h\langle\nabla\phi_t(Z_t), b(Z_t)\rangle + \gamma(t)h\left((\Pi_Y\phi_t)(\tilde{Z}_{t+h}) - \phi_t(\tilde{Z}_{t+h})\right) + O(h^2)\right]}{h}$$

$$= \mathbb{E}_{\rho_t^Z}\left[\frac{\partial\phi_t}{\partial t}(Z_t) + \langle\nabla\phi_t(Z_t), b(Z_t)\rangle + \gamma(t)((\Pi_Y\phi_t)(Z_t) - \phi_t(Z_t))\right].$$

$\square$

### F.1 Proof of Theorem 2 (convergence of RHF under strong convexity)

**Theorem 2.** *Assume $f$ is $\alpha$-strongly convex. Let $(X_t, Y_t)$ evolve following* (RHF) *with the choice $\gamma(t) = \sqrt{\frac{16\alpha}{5}}$, from any $X_0 \in \mathbb{R}^d$ with $Y_0 = 0$. Then for any $t \geq 0$, we have*

$$\mathbb{E}[f(X_t) - f(x^*)] \leq \exp\left(-\sqrt{\frac{\alpha}{5}}t\right)\mathbb{E}\left[f(X_0) - f(x^*) + \frac{\alpha}{10}\|X_0 - x^*\|^2\right].$$

*Proof.* We construct the following Lyapunov function:

$$\mathcal{E}_t = \exp\left(\sqrt{\frac{\alpha}{5}}\right)\mathbb{E}\left[f(X_t) - f(x^*) + \frac{\alpha}{10}\left\|X_t - x^* + \sqrt{\frac{5}{\alpha}}Y_t\right\|^2\right]. \tag{15}$$

Let $\widetilde{\mathcal{E}}_t = \mathbb{E}\left[f(X_t) - f(x^*) + \frac{\alpha}{10}\left\|X_t - x^* + \sqrt{\frac{5}{\alpha}}Y_t\right\|^2\right]$. Apply the chain rule, and we have

$$\dot{\mathcal{E}}_t = \sqrt{\frac{\alpha}{5}}\exp\left(\sqrt{\frac{\alpha}{5}}t\right)\widetilde{\mathcal{E}}_t + \exp\left(\sqrt{\frac{\alpha}{5}}t\right)\dot{\widetilde{\mathcal{E}}}_t.$$

In order to show $\dot{\mathcal{E}}_t \leq 0$, it suffices to show $\dot{\widetilde{\mathcal{E}}}_t \leq -\sqrt{\frac{\alpha}{5}}\widetilde{\mathcal{E}}_t$. Apply Lemma 13 by taking $\phi_t = E_t$, and we obtain

$$
\dot{\widetilde{\mathcal{E}}}_t = \mathbb{E}\left[\langle \nabla f(X_t), Y_t\rangle + \left\langle \begin{pmatrix} \frac{\alpha}{5}\left(X_t - x^* + \sqrt{\frac{5}{\alpha}}Y_t\right) \\ \sqrt{\frac{\alpha}{5}}\left(X_t - x^* + \sqrt{\frac{5}{\alpha}}Y_t\right) \end{pmatrix}, \begin{pmatrix} Y_t \\ -\nabla f(X_t) \end{pmatrix}\right\rangle\right]
$$

$$
\qquad + \frac{\gamma(t)\alpha}{10}\mathbb{E}\left[\|X_t - x^*\|^2 - \left\|X_t - x^* + \sqrt{\frac{5}{\alpha}}Y_t\right\|^2\right]
$$

$$
= \mathbb{E}\left[\sqrt{\frac{\alpha}{5}}\langle x^* - X_t, \nabla f(X_t)\rangle + \left(\frac{\alpha}{5} - \gamma(t)\sqrt{\frac{\alpha}{5}}\right)\langle X_t - x^*, Y_t\rangle + \left(\sqrt{\frac{\alpha}{5}} - \frac{\gamma(t)}{2}\right)\|Y_t\|^2\right]
$$

$$
\leq \mathbb{E}\left[-\sqrt{\frac{\alpha}{5}}\left(f(X_t) - f(x^*)\right) - \frac{\alpha^{3/2}}{2\sqrt{5}}\|X_t - x^*\|^2 - \frac{3\alpha}{5}\langle X_t - x^*, Y_t\rangle - \sqrt{\frac{\alpha}{5}}\|Y_t\|^2\right]
$$

$$
= -\sqrt{\frac{\alpha}{5}}\widetilde{\mathcal{E}}_t - \mathbb{E}\left[\frac{2\alpha^{3/2}}{5\sqrt{5}}\|X_t - x^*\|^2 + \frac{2\alpha}{5}\langle X_t - x^*, Y_t\rangle + \frac{1}{2}\sqrt{\frac{\alpha}{5}}\|Y_t\|^2\right]
$$

$$
= -\sqrt{\frac{\alpha}{5}}\widetilde{\mathcal{E}}_t - 2\sqrt{\frac{\alpha}{5}}\mathbb{E}\left[\frac{\alpha}{5}\|X_t - x^*\|^2 + \sqrt{\frac{\alpha}{5}}\langle X_t - x^*, Y_t\rangle + \frac{1}{4}\|Y_t\|^2\right]
$$

$$
= -\sqrt{\frac{\alpha}{5}}\widetilde{\mathcal{E}}_t - 2\sqrt{\frac{\alpha}{5}}\mathbb{E}\left[\left\|\sqrt{\frac{\alpha}{5}}(X_t - x^*) + \frac{1}{2}Y_t\right\|^2\right]
$$

$$
\leq -\sqrt{\frac{\alpha}{5}}\widetilde{\mathcal{E}}_t,
$$

where the first inequality follows from $\alpha$-strong convexity:

$$
\langle x^* - X_t, \nabla f(X_t)\rangle \leq f(x^*) - f(X_t) - \frac{\alpha}{2}\|X_t - x^*\|^2.
$$

Thus we have $\mathcal{E}_t \leq \mathcal{E}_0$, which implies

$$
\mathbb{E}\left[f(X_t) - f(x^*)\right] \leq \exp\left(-\sqrt{\frac{\alpha}{5}}t\right)\mathbb{E}\left[f(X_0) - f(x^*) + \frac{\alpha}{10}\|X_0 - x^*\|^2\right].
$$

$\square$

## F.2 Proof of Theorem 3 (convergence of RHF under weak convexity)

**Theorem 3.** *Assume $f$ is weakly convex. Let $(X_t, Y_t)$ evolve following* (RHF) *with the choice $\gamma(t) = \frac{6}{t+1}$, from any $X_0 \in \mathbb{R}^d$ with $Y_0 = 0$. Then for any $t \geq 0$, we have*

$$
\mathbb{E}\left[f(X_t) - f(x^*)\right] \leq \frac{5 \cdot \mathbb{E}\left[f(X_0) - f(x^*) + \|X_0 - x^*\|^2\right]}{(t+1)^2}.
$$

*Proof.* We construct the following Lyapunov function:

$$
\mathcal{E}_t = \mathbb{E}\left[\frac{(t+1)^2}{4}(f(X_t) - f(x^*)) + \frac{1}{2}\left\|X_t - x^* + \frac{t+1}{2}Y_t\right\|^2 + \frac{3}{4}\|X_t - x^*\|^2\right]. \qquad (16)
$$

Apply Lemma 13 by taking $\phi_t = \mathcal{E}_t$, and we obtain

$$
\dot{\mathcal{E}}_t = \mathbb{E}\left[\frac{(t+1)^2}{4}\langle \nabla f(X_t), Y_t\rangle + \frac{t+1}{2}(f(X_t) - f(x^*))\right]
$$

$$
\quad + \mathbb{E}\left[\left\langle \begin{pmatrix} X_t - x^* + \frac{t+1}{2}Y_t \\ \frac{t+1}{2}(X_t - x^* + \frac{t+1}{2}Y_t) \end{pmatrix}, \begin{pmatrix} Y_t \\ -\nabla f(X_t) \end{pmatrix}\right\rangle + \frac{\gamma(t)}{2}\left(\|X_t - x^*\|^2 - \left\|X_t - x^* + \frac{t+1}{2}Y_t\right\|^2\right)\right]
$$

$$
\quad + \mathbb{E}\left[\left\langle X_t - x^* + \frac{t+1}{2}Y_t, \frac{1}{2}Y_t\right\rangle\right] + \mathbb{E}\left[\frac{3}{2}\langle X_t - x^*, Y_t\rangle\right]
$$

$$= \mathbb{E}\left[\frac{(t+1)^2}{4}\langle\nabla f(X_t), Y_t\rangle + \frac{t+1}{2}(f(X_t) - f(x^*))\right] + \mathbb{E}\left[\left\langle X_t - x^* + \frac{t+1}{2}Y_t, \frac{3}{2}Y_t - \frac{t+1}{2}\nabla f(X_t)\right\rangle\right]$$

$$- \mathbb{E}\left[\frac{\gamma(t)\cdot(t+1)}{2}\langle X_t - x^*, Y_t\rangle - \frac{\gamma(t)\cdot(t+1)^2}{8}\|Y_t\|^2\right] + \mathbb{E}\left[\frac{3}{2}\langle X_t - x^*, Y_t\rangle\right]$$

$$= \mathbb{E}\left[\left(3 - \frac{\gamma(t)\cdot(t+1)}{2}\right)\langle X_t - x^*, Y_t\rangle + \left(\frac{3(t+1)}{4} - \frac{\gamma(t)\cdot(t+1)^2}{8}\right)\|Y_t\|^2\right]$$

$$+ \frac{t+1}{2}\mathbb{E}\left[f(X_t) - f(x^*) + \langle x^* - X_t, \nabla f(X_t)\rangle\right] \le 0,$$

where the inequality above follows from the choice of $\gamma(t)$ and weak convexity:

$$\langle x^* - X_t, \nabla f(X_t)\rangle \le f(x^*) - f(X_t).$$

Thus we have $\mathcal{E}_t \le \mathcal{E}_0$, which implies

$$\mathbb{E}\left[f(X_t) - f(x^*)\right] \le \frac{5\mathbb{E}\left[f(X_0) - f(x^*) + \|X_0 - x^*\|^2\right]}{(t+1)^2}.$$

$\square$

# G  Discretization analysis

## G.1  Randomized proximal Hamiltonian descent

Based on the implicit integrator formulation (3), we propose the **randomized proximal Hamiltonian descent** (RPHD) algorithm:

---
**Algorithm 5 Randomized Hamiltonian Proximal Descent** (RPHD)
---
1: Initialize $x_0 \in \mathbb{R}^d$ and $y_0 = 0$. Choose refreshment parameter $\gamma_k > 0$, step size $h > 0$.
2: **for** $k = 0, 1, \ldots, K - 1$ **do**
3:    $x_{k+\frac{1}{2}} = x_k + hy_k$
4:    $x_{k+1} = \text{Prox}_{h^2 f}(x_{k+\frac{1}{2}})$
5:    $\tilde{y}_{k+1} = y_k - h\nabla f(x_{k+1})$
6:    $y_{k+1} = \begin{cases} \tilde{y}_{k+1} & \text{with probability } 1 - \min(\gamma_k h, 1) \\ 0 & \text{with probability } \min(\gamma_k h, 1) \end{cases}$
7: **end for**
8: **return** $x_K$

---

### G.1.1  For strongly convex functions

We now state the convergence rate of RPHD for minimizing strongly convex functions.

**Theorem 14.** *Assume $f$ is $\alpha$-strongly convex. Then for all $k \ge 0$, RPHD (Algorithm 5) with $0 < h \le 1$, $\gamma_k = \sqrt{\alpha}$ and any $x_0 \in \mathbb{R}^d$ satisfies*

$$\mathbb{E}[f(x_k) - f(x^*)] \le \left(1 + \frac{\sqrt{\alpha}h}{6}\right)^{-k} \mathbb{E}\left[f(x_0) - f(x^*) + \frac{\alpha}{72}\|x_0 - x^*\|^2\right].$$

*Proof.* We construct the Lyapunov function:

$$E_k = \left(1 + \frac{\sqrt{\alpha}h}{6}\right)^k \mathbb{E}\left[f(x_k) - f(x^*) + \frac{\alpha}{72}\left\|x_k - x^* + \frac{6}{\sqrt{\alpha}}y_k\right\|^2\right].$$

We also define

$$L_k = \mathbb{E}\left[f(x_k) - f(x^*) + \frac{\alpha}{72}\left\|x_k - x^* + \frac{6}{\sqrt{\alpha}}y_k\right\|^2\right].$$

In order to show $E_{k+1} - E_k \leq 0$, it suffices to show $L_{k+1} - L_k \leq -\frac{\sqrt{\alpha}h}{6}L_{k+1}$. By the accept-refresh step, we have

$$L_{k+1} = (1 - \gamma_k h) \cdot \mathbb{E}\left[f(x_{k+1}) - f(x^*) + \frac{\alpha}{72}\left\|x_{k+1} - x^* + \frac{6}{\sqrt{\alpha}}\tilde{y}_{k+1}\right\|^2\right]$$

$$+ \gamma_k h \cdot \mathbb{E}\left[f(x_{k+1}) - f(x^*) + \frac{\alpha}{72}\|x_{k+1} - x^*\|^2\right]$$

$$= \mathbb{E}\left[f(x_{k+1}) - f(x^*) + \frac{\alpha}{72}\left\|x_{k+1} - x^* + \frac{6}{\sqrt{\alpha}}\tilde{y}_{k+1}\right\|^2\right]$$

$$+ \frac{\alpha^{3/2}h}{72}\mathbb{E}\left[\|x_{k+1} - x^*\|^2 - \left\|x_{k+1} - x^* + \frac{6}{\sqrt{\alpha}}\tilde{y}_{k+1}\right\|^2\right].$$

**Note that we hide $\mathbb{E}$ in the calculation below for simplicity.** Taking difference between $L_{k+1}$ and $L_k$ and applying Lemma 15, we obtain

$$L_{k+1} - L_k = f(x_{k+1}) - f(x_k) + \frac{\alpha}{72}\left(\left\|x_{k+1} - x^* + \frac{6}{\sqrt{\alpha}}\tilde{y}_{k+1}\right\|^2 - \left\|x_k - x^* + \frac{6}{\sqrt{\alpha}}y_k\right\|^2\right)$$

$$+ \frac{\alpha^{3/2}h}{72}\left(\|x_{k+1} - x^*\|^2 - \left\|x_{k+1} - x^* + \frac{6}{\sqrt{\alpha}}\tilde{y}_{k+1}\right\|^2\right)$$

$$= \underbrace{f(x_{k+1}) - f(x_k)}_{\text{I}} + \underbrace{\frac{\alpha}{36}\left\langle x_{k+1} - x_k + \frac{6}{\sqrt{\alpha}}(\tilde{y}_{k+1} - y_k), x_{k+1} - x^* + \frac{6}{\sqrt{\alpha}}\tilde{y}_{k+1}\right\rangle}_{\text{II}}$$

$$\underbrace{- \frac{\alpha}{72}\left\|x_{k+1} - x_k + \frac{6}{\sqrt{\alpha}}(\tilde{y}_{k+1} - y_k)\right\|^2}_{\text{III}} - \frac{\alpha h}{6}\langle x_{k+1} - x^*, \tilde{y}_{k+1}\rangle - \frac{\sqrt{\alpha}h}{2}\|\tilde{y}_{k+1}\|^2.$$

For I, we apply $\alpha$-strong convexity and update (2a) to obtain

$$\text{I} \leq \langle \nabla f(x_{k+1}), x_{k+1} - x_k\rangle - \frac{\alpha}{2}\|x_{k+1} - x_k\|^2 = h\langle \nabla f(x_{k+1}), \tilde{y}_{k+1}\rangle - \frac{\alpha h^2}{2}\|\tilde{y}_{k+1}\|^2.$$

For II, we apply updates (2a) and (2b) to obtain

$$\text{II} = \frac{\alpha h}{36}\left\langle \tilde{y}_{k+1} - \frac{6}{\sqrt{\alpha}}\nabla f(x_{k+1}), x_{k+1} - x^* + \frac{6}{\sqrt{\alpha}}\tilde{y}_{k+1}\right\rangle$$

$$= \frac{\alpha h}{36}\langle \tilde{y}_{k+1}, x_{k+1} - x^*\rangle + \frac{\sqrt{\alpha}h}{6}\|\tilde{y}_{k+1}\|^2 + \frac{\sqrt{\alpha}h}{6}\langle \nabla f(x_{k+1}), x^* - x_{k+1}\rangle - h\langle \nabla f(x_{k+1}), \tilde{y}_{k+1}\rangle.$$

For III, we apply updates (2a) and (2b) and expand to obtain

$$\text{III} = -\frac{\alpha h^2}{72}\left\|\tilde{y}_{k+1} - \frac{6}{\sqrt{\alpha}}\nabla f(x_{k+1})\right\|^2$$

$$= -\frac{\alpha h^2}{72}\|\tilde{y}_{k+1}\|^2 + \frac{\sqrt{\alpha}h^2}{6}\langle \nabla f(x_{k+1}), \tilde{y}_{k+1}\rangle - \frac{h^2}{2}\|\nabla f(x_{k+1})\|^2.$$

Combining the calculation above, we obtain

$$L_{k+1} - L_k \leq \frac{\sqrt{\alpha}h^2}{6}\langle \nabla f(x_{k+1}), \tilde{y}_{k+1}\rangle - \left(\frac{\sqrt{\alpha}h}{3} + \frac{37\alpha h^2}{72}\right)\|\tilde{y}_{k+1}\|^2 - \frac{5\alpha h}{36}\langle x_{k+1} - x^*, \tilde{y}_{k+1}\rangle$$

$$+ \frac{\sqrt{\alpha}h}{6}\langle \nabla f(x_{k+1}), x^* - x_{k+1}\rangle - \frac{h^2}{2}\|\nabla f(x_{k+1})\|^2$$

$$\leq \frac{\sqrt{\alpha}h^2}{6}\langle \nabla f(x_{k+1}), \tilde{y}_{k+1}\rangle - \left(\frac{\sqrt{\alpha}h}{3} + \frac{37\alpha h^2}{72}\right)\|\tilde{y}_{k+1}\|^2 - \frac{5\alpha h}{36}\langle x_{k+1} - x^*, \tilde{y}_{k+1}\rangle$$

$$+ \frac{\sqrt{\alpha}h}{6}(f(x^*) - f(x_{k+1})) - \frac{\alpha^{3/2}h}{12}\|x^* - x_{k+1}\|^2 - \frac{h^2}{2}\|\nabla f(x_{k+1})\|^2. \tag{17}$$

Since $L_{k+1}$ consists of $f(x^*) - f(x_{k+1})$, we can write $f(x^*) - f(x_{k+1})$ in terms of $L_{k+1}$:

$$f(x^*) - f(x_{k+1}) = -L_{k+1} + \frac{\alpha}{72}\left\|x_{k+1} - x^* + \frac{6}{\sqrt{\alpha}}\tilde{y}_{k+1}\right\|^2$$
$$+ \frac{\alpha^{3/2}h}{72}\left(\|x_{k+1} - x^*\|^2 - \left\|x_{k+1} - x^* + \frac{6}{\sqrt{\alpha}}\tilde{y}_{k+1}\right\|^2\right)$$
$$= -L_{k+1} + \frac{\alpha}{72}\|x^* - x_{k+1}\|^2 + \left(\frac{\sqrt{\alpha}}{6} - \frac{\alpha h}{6}\right)\langle x_{k+1} - x^*, \tilde{y}_{k+1}\rangle$$
$$+ \left(\frac{1}{2} - \frac{\sqrt{\alpha}h}{2}\right)\|\tilde{y}_{k+1}\|^2$$

Substituting $f(x^*) - f(x_{k+1})$ in (17) with the relation above, we obtain

$$L_{k+1} - L_k \leq -\frac{\sqrt{\alpha}h}{6}L_{k+1} - \frac{35\alpha^{3/2}h}{432}\|x^* - x_{k+1}\|^2 - \frac{\alpha h}{9}\langle x_{k+1} - x^*, \tilde{y}_{k+1}\rangle$$
$$- \left(\frac{\sqrt{\alpha}h}{4} + \frac{43\alpha h^2}{72}\right)\|\tilde{y}_{k+1}\|^2 - \frac{h^2}{2}\|\nabla f(x_{k+1})\|^2$$
$$+ \frac{\sqrt{\alpha}h^2}{6}\langle \nabla f(x_{k+1}), \tilde{y}_{k+1}\rangle - \frac{\alpha^{3/2}h^2}{36}\langle x_{k+1} - x^*, \tilde{y}_{k+1}\rangle. \tag{18}$$

Now we control (18). Applying Lemma 16 and assuming $\alpha \leq 1$ and $h \leq 1$, we have

$$\frac{\sqrt{\alpha}h^2}{6}\langle \nabla f(x_{k+1}), \tilde{y}_{k+1}\rangle - \frac{\alpha^{3/2}h^2}{36}\langle x_{k+1} - x^*, \tilde{y}_{k+1}\rangle$$
$$\leq \frac{h^2}{12}\|\nabla f(x_{k+1})\|^2 + \frac{\alpha h^2}{12}\|\tilde{y}_{k+1}\|^2 + \frac{\alpha^{3/2}h^2}{72}\|x^* - x_{k+1}\|^2 + \frac{\alpha^{3/2}h^2}{72}\|\tilde{y}_{k+1}\|^2$$
$$\leq \frac{h^2}{12}\|\nabla f(x_{k+1})\|^2 + \frac{7\alpha h^2}{72}\|\tilde{y}_{k+1}\|^2 + \frac{\alpha^{3/2}h}{72}\|x^* - x_{k+1}\|^2.$$

Plugging this into (18), we obtain

$$L_{k+1} - L_k \leq -\frac{\sqrt{\alpha}h}{6}L_{k+1} - \frac{29\alpha^{3/2}h}{432}\|x^* - x_{k+1}\|^2 - \frac{\alpha h}{9}\langle x_{k+1} - x^*, \tilde{y}_{k+1}\rangle$$
$$- \left(\frac{\sqrt{\alpha}h}{4} + \frac{\alpha h^2}{2}\right)\|\tilde{y}_{k+1}\|^2 - \frac{5h^2}{12}\|\nabla f(x_{k+1})\|^2$$
$$\leq -\frac{\sqrt{\alpha}h}{6}L_{k+1} - \frac{\alpha^{3/2}h}{27}\|x^* - x_{k+1}\|^2 - \frac{\alpha h}{9}\langle x_{k+1} - x^*, \tilde{y}_{k+1}\rangle - \frac{\sqrt{\alpha}h}{12}\|\tilde{y}_{k+1}\|^2$$
$$= -\frac{\sqrt{\alpha}h}{6}L_{k+1} - \frac{\alpha^{3/2}h}{27}\left\|x_{k+1} - x^* + \frac{3}{2\sqrt{\alpha}}\tilde{y}_{k+1}\right\|^2$$
$$\leq -\frac{\sqrt{\alpha}h}{6}L_{k+1}.$$

Thus we have $E_k \leq E_0$, which implies

$$\mathbb{E}[f(x_k) - f(x^*)] \leq \left(1 + \frac{\sqrt{\alpha}h}{6}\right)^{-k}\mathbb{E}\left[f(x_0) - f(x^*) + \frac{\alpha}{72}\|x_0 - x^*\|^2\right].$$

$\square$

### G.1.2   For weakly convex functions

We now state the convergence rate of RPHD for minimizing weakly convex functions.

**Theorem 15.** *Assume $f$ is weakly convex. Then for all $k \geq 0$, RPHD (Algorithm 5) with $h > 0$, $\gamma_k = \frac{17}{2(k+9)h}$ and any $x_0 \in \mathbb{R}^d$ satisfies*

$$\mathbb{E}[f(x_k) - f(x^*)] \leq \frac{45\mathbb{E}\left[\|x_0 - x^*\|^2\right]}{4h^2(k+8)^2}.$$

*Proof.* We construct the following Lyapunov function:

$$E_k = \mathbb{E}\left[\frac{h^2(k+8)^2}{9}(f(x_k) - f(x^*)) + \frac{1}{2}\left\|x_k - x^* + \frac{(k+8)h}{3}y_k\right\|^2 + \frac{3}{4}\|x_k - x^*\|^2\right],$$

we derive $E_{k+1}$ by accept-refresh step:

$$E_{k+1} = (1 - \gamma_k h) \cdot \mathbb{E}\left[\frac{h^2(k+9)^2}{9}(f(x_{k+1}) - f(x^*)) + \frac{1}{2}\left\|x_{k+1} - x^* + \frac{(k+9)h}{3}\tilde{y}_{k+1}\right\|^2 + \frac{3}{4}\|x_{k+1} - x^*\|^2\right]$$

$$+ \gamma_k h \cdot \mathbb{E}\left[\frac{h^2(k+9)^2}{9}(f(x_{k+1}) - f(x^*)) + \frac{1}{2}\|x_{k+1} - x^*\|^2 + \frac{3}{4}\|x_{k+1} - x^*\|^2\right]$$

$$= \mathbb{E}\left[\frac{h^2(k+9)^2}{9}(f(x_{k+1}) - f(x^*)) + \frac{1}{2}\left\|x_{k+1} - x^* + \frac{(k+9)h}{3}\tilde{y}_{k+1}\right\|^2 + \frac{3}{4}\|x_{k+1} - x^*\|^2\right]$$

$$+ \frac{17}{4(k+9)}\mathbb{E}\left[\|x_{k+1} - x^*\|^2 - \left\|x_{k+1} - x^* + \frac{(k+9)h}{3}\tilde{y}_{k+1}\right\|^2\right].$$

**Note that we hide $\mathbb{E}$ in the calculation below for simplicity.** Taking difference between $E_{k+1}$ and $E_k$ and applying Lemma 15, we obtain

$$E_{k+1} - E_k = \frac{h^2(k+9)^2}{9}(f(x_{k+1}) - f(x^*)) - \frac{h^2(k+8)^2}{9}(f(x_k) - f(x^*))$$

$$+ \frac{1}{2}\left\|x_{k+1} - x^* + \frac{(k+9)h}{3}\tilde{y}_{k+1}\right\|^2 - \frac{1}{2}\left\|x_k - x^* + \frac{(k+8)h}{3}\tilde{y}_k\right\|^2$$

$$+ \frac{3}{4}\|x_{k+1} - x^*\|^2 - \frac{3}{4}\|x_k - x^*\|^2 + \frac{17}{4(k+9)}\left(\|x_{k+1} - x^*\|^2 - \left\|x_{k+1} - x^* + \frac{(k+9)h}{3}\tilde{y}_{k+1}\right\|^2\right)$$

$$= \underbrace{\left(\frac{(k+9)^2h^2}{9} - \frac{(k+8)^2h^2}{9}\right)(f(x_{k+1}) - f(x^*)) + \underbrace{\frac{(k+8)^2h^2}{9}(f(x_{k+1}) - f(x_k))}_{\text{II}}}_{\text{I}}$$

$$+ \underbrace{\left\langle x_{k+1} - x_k + \frac{(k+9)h}{3}\tilde{y}_{k+1} - \frac{(k+8)h}{3}y_k, x_{k+1} - x^* + \frac{(k+9)h}{3}\tilde{y}_{k+1}\right\rangle}_{\text{III}}$$

$$\underbrace{-\frac{1}{2}\left\|x_{k+1} - x_k + \frac{(k+9)h}{3}\tilde{y}_{k+1} - \frac{(k+8)h}{3}y_k\right\|^2}_{\text{IV}} + \underbrace{\frac{3}{2}\langle x_{k+1} - x_k, x_{k+1} - x^*\rangle - \frac{3}{4}\|x_{k+1} - x_k\|^2}_{\text{V}}$$

$$- \frac{17h}{6}\langle x_{k+1} - x^*, \tilde{y}_{k+1}\rangle - \frac{17(k+9)h^2}{36}\|\tilde{y}_{k+1}\|^2$$

For II, we apply weak convexity and update (2a) to obtain

$$\text{II} \leq \frac{(k+8)^2h^2}{9}\langle\nabla f(x_{k+1}), x_{k+1} - x_k\rangle = \frac{(k+8)^2h^3}{9}\langle\nabla f(x_{k+1}), \tilde{y}_{k+1}\rangle.$$

For III, we apply updates (2a) and (2b) to obtain

$$\text{III} = \left\langle h\tilde{y}_{k+1} + \frac{h}{3}\tilde{y}_{k+1} - \frac{(k+8)h^2}{3}\nabla f(x_{k+1}), x_{k+1} - x^* + \frac{(k+9)h}{3}\tilde{y}_{k+1}\right\rangle$$

$$= \frac{4h}{3}\langle\tilde{y}_{k+1}, x_{k+1} - x^*\rangle + \frac{4(k+9)h^2}{9}\|\tilde{y}_{k+1}\|^2 + \frac{(k+8)h^2}{3}\langle\nabla f(x_{k+1}), x^* - x_{k+1}\rangle$$

$$- \frac{(k+8)(k+9)h^3}{9}\langle\nabla f(x_{k+1}), \tilde{y}_{k+1}\rangle.$$

For IV, we apply updates (2a) and (2b) and expand to obtain

$$\text{IV} = -\frac{8h^2}{9}\|\tilde{y}_{k+1}\|^2 + \frac{4(k+8)h^3}{9}\langle\nabla f(x_{k+1}), \tilde{y}_{k+1}\rangle - \frac{(k+8)^2h^4}{18}\|\nabla f(x_{k+1})\|^2.$$

For V, we apply update (2a) to obtain

$$\mathsf{V} = \frac{3h}{2}\langle \tilde{y}_{k+1}, x_{k+1} - x^*\rangle - \frac{3h^2}{4}\|\tilde{y}_{k+1}\|^2$$

Combining the calculation above, we obtain

$$
\begin{aligned}
E_{k+1} - E_k &\le \frac{(2k+17)h^2}{9}\left(f(x_{k+1}) - f(x^*)\right) + \frac{(k+8)h^2}{3}\langle\nabla f(x_{k+1}), x^* - x_{k+1}\rangle \\
&\quad + \frac{(k+8)h^3}{3}\langle\nabla f(x_{k+1}), \tilde{y}_{k+1}\rangle - \left(\frac{(k+9)h^2}{36} + \frac{59h^2}{36}\right)\|\tilde{y}_{k+1}\|^2 - \frac{(k+8)^2h^4}{18}\|\nabla f(x_{k+1})\|^2 \\
&\le \frac{(k+8)h^2}{3}\left(f(x_{k+1}) - f(x^*)\right) + \frac{(k+8)h^2}{3}\langle\nabla f(x_{k+1}), x^* - x_{k+1}\rangle \\
&\quad + \frac{(k+8)h^3}{3}\langle\nabla f(x_{k+1}), \tilde{y}_{k+1}\rangle - \frac{59h^2}{36}\|\tilde{y}_{k+1}\|^2 - \frac{(k+8)^2h^4}{18}\|\nabla f(x_{k+1})\|^2 \\
&\le \frac{(k+8)h^3}{3}\langle\nabla f(x_{k+1}), \tilde{y}_{k+1}\rangle - \frac{59h^2}{36}\|\tilde{y}_{k+1}\|^2 - \frac{(k+8)^2h^4}{18}\|\nabla f(x_{k+1})\|^2 \\
&= \frac{h^2}{3}\langle (k+8)h\nabla f(x_{k+1}), \tilde{y}_{k+1}\rangle - \frac{59h^2}{36}\|\tilde{y}_{k+1}\|^2 - \frac{(k+8)^2h^4}{18}\|\nabla f(x_{k+1})\|^2 \\
&\le \frac{(k+8)^2h^4}{36}\|\nabla f(x_{k+1})\|^2 - \frac{(k+8)^2h^4}{18}\|\nabla f(x_{k+1})\|^2 + h^2\|\tilde{y}_{k+1}\|^2 - \frac{59h^2}{36}\|\tilde{y}_{k+1}\|^2 \\
&= -\frac{(k+8)^2h^4}{36}\|\nabla f(x_{k+1})\|^2 - \frac{23h^2}{36}\|\tilde{y}_{k+1}\|^2 \le 0.
\end{aligned}
$$

where the second inequality follows from Lemma 17; the third inequality follows from weak convexity, and the last inequality follows from Lemma 16. Thus we have $E_k \le E_0$, which implies

$$\mathbb{E}[f(x_k) - f(x^*)] \le \frac{45\mathbb{E}\left[\|x_0 - x^*\|^2\right]}{4h^2(k+8)^2}.$$

$\square$

## G.2 Approximation error

Starting from $x_{k+\frac{1}{2}}$, we run proximal point update with stepsize $h^2$ to obtain $\hat{x}_{k+1}$ and run gradient descent with stepsize $h^2$ to obtain $x_{k+1}$. The following proposition bound the error under smoothness.

**Proposition 1.** *Let $\hat{x}_{k+1} = \mathrm{Prox}_{h^2 f}(x_{k+\frac{1}{2}})$ and $x_{k+1} = x_{k+\frac{1}{2}} - h^2\nabla f(x_{k+\frac{1}{2}})$. If $f$ is $L$-smooth and $h < \frac{1}{2^{1/8}\cdot\sqrt{L}}$, then*

$$\|x_{k+1} - \hat{x}_{k+1}\|^2 \le \frac{2L^2h^8}{1 - 2L^4h^8}\|\nabla f(x_{k+1})\|^2.$$

*Proof.* We use the following updates to prove this proposition.

$$\hat{x}_{k+1} = x_{k+\frac{1}{2}} - h^2\nabla f(\hat{x}_{k+1}), \tag{19}$$

$$x_{k+1} = x_{k+\frac{1}{2}} - h^2\nabla f(x_{k+\frac{1}{2}}), \tag{20}$$

$$\|\nabla f(x) - \nabla f(y)\|^2 \le L^2\|x - y\|,, \tag{21}$$

where (20) follows from RHGD (Algorithm 3) and the last relation follows from $L$-smoothness of $f$ for any $x, y \in \mathbb{R}^d$. Subtracting (19) from (20), we obtain

$$\|x_{k+1} - \hat{x}_{k+1}\|^2 = h^4\|\nabla f(\hat{x}_{k+1}) - \nabla f(x_{k+\frac{1}{2}})\|^2 \overset{(21)}{\le} L^2h^4\|\hat{x}_{k+1} - x_{k+\frac{1}{2}}\|^2 \overset{(19)}{=} L^2h^8\|\nabla f(\hat{x}_{k+1})\|^2.$$

Using (40) in Lemma 18, we have

$$
\begin{aligned}
\|\nabla f(\hat{x}_{k+1})\|^2 &\le 2\|\nabla f(\hat{x}_{k+1}) - \nabla f(x_{k+1})\|^2 + 2\|\nabla f(x_{k+1})\|^2 \\
&\overset{(21)}{\le} 2L^2\|\hat{x}_{k+1} - x_{k+1}\|^2 + 2\|\nabla f(x_{k+1})\|^2.
\end{aligned}
$$

Thus we have

$$\|x_{k+1} - \hat{x}_{k+1}\|^2 \le 2L^4 h^8 \|\hat{x}_{k+1} - x_{k+1}\|^2 + 2L^2 h^8 \|\nabla f(x_{k+1})\|^2.$$

If $h \le \frac{1}{4^{1/8}\sqrt{L}}$, then rearranging the relation above yields

$$\|x_{k+1} - \hat{x}_{k+1}\|^2 \le \frac{2L^2 h^8}{1 - 2L^4 h^8} \|\nabla f(x_{k+1})\|^2.$$

$\square$

### G.3   Convergence analysis of RHGD

Since we lose update (2a) for RHGD we construct two dummy iterates $\hat{x}_{k+1}$ and $\hat{y}_{k+1}$ satisfying

$$\hat{x}_{k+1} - x_k = h\hat{y}_{k+1}, \tag{22}$$
$$\hat{y}_{k+1} - y_k = -h\nabla f(\hat{x}_{k+1}). \tag{23}$$

Substituting $\hat{y}_{k+1}$ in (22) with $\hat{y}_{k+1} = y_k - h\nabla f(\hat{x}_{k+1})$ from (23) yields $\hat{x}_{k+1} = \mathrm{Prox}_{h^2 f}(x_{k+\frac{1}{2}})$. We define the error term $\mathcal{G}_{k+1}^h$ as

$$\mathcal{G}_{k+1}^h := x_{k+1} - \hat{x}_{k+1} + h(\hat{y}_{k+1} - \tilde{y}_{k+1}).$$

Then we can write

$$\begin{aligned}
x_{k+1} - x_k &= \hat{x}_{k+1} - x_k + x_{k+1} - \hat{x}_{k+1} \\
&\overset{(22)}{=} h\hat{y}_{k+1} + x_{k+1} - \hat{x}_{k+1} \\
&= h\tilde{y}_{k+1} + x_{k+1} - \hat{x}_{k+1} + h(\hat{y}_{k+1} - \tilde{y}_{k+1}) \\
&= h\tilde{y}_{k+1} + \mathcal{G}_{k+1}^h. \tag{24}
\end{aligned}$$

The next lemma controls the error term $\mathcal{G}_{k+1}^h$.

**Proposition 2.** *If $f$ is $L$-smooth, then we have $\|\mathcal{G}_{k+1}^h\|^2 \le 2(1 + L^2 h^4)\|x_{k+1} - \hat{x}_{k+1}\|^2$.*

*Proof.* Using updates (2b), (23) and $L$-smoothness of $f$, we have

$$\begin{aligned}
\|\mathcal{G}_{k+1}^h\|^2 &= \|x_{k+1} - \hat{x}_{k+1} + h(\hat{y}_{k+1} - \tilde{y}_{k+1})\|^2 \\
&\le 2\|x_{k+1} - \hat{x}_{k+1}\|^2 + 2h^2\|\hat{y}_{k+1} - \tilde{y}_{k+1}\|^2 \\
&= 2\|x_{k+1} - \hat{x}_{k+1}\|^2 + 2h^4\|\nabla f(\hat{x}_{k+1}) - \nabla f(x_{k+1})\|^2 \\
&\le 2\|x_{k+1} - \hat{x}_{k+1}\|^2 + 2L^2 h^4\|x_{k+1} - \hat{x}_{k+1}\|^2 \\
&= 2(1 + L^2 h^4)\|x_{k+1} - \hat{x}_{k+1}\|^2.
\end{aligned}$$

$\square$

#### G.3.1   Proof of Theorem 4 (convergence of RHGD under strong convexity)

**Theorem 4.** *Assume $f$ is $\alpha$-strongly convex and $L$-smooth. Then for all $k \ge 0$, RHGD (Algorithm 3) with $h \le \frac{1}{4\sqrt{L}}$, $\gamma_k = \sqrt{\alpha}$ and any $x_0 \in \mathbb{R}^d$ satisfies*

$$\mathbb{E}[f(x_k) - f(x^*)] \le \left(1 + \frac{\sqrt{\alpha}h}{6}\right)^{-k} \mathbb{E}\left[f(x_0) - f(x^*) + \frac{\alpha}{72}\|x_0 - x^*\|^2\right].$$

*Proof.* Consider the following function

$$\mathcal{L}_k = \mathbb{E}\left[f(x_k) - f(x^*) + \frac{\alpha}{72}\left\|x_k - x^* + \frac{6}{\sqrt{\alpha}}y_k\right\|^2\right]. \tag{25}$$

We will bound $\mathcal{L}_{k+1} - \mathcal{L}_k$ in terms of $\mathcal{L}_{k+1}$, similar to the proof of Theorem 14 in Section G.1.1. We follow the calculation in Section G.1.1 and **hide $\mathbb{E}$ below for simplicity**:

$$\mathcal{L}_{k+1} - \mathcal{L}_k = f(x_{k+1}) - f(x_k) + \frac{\alpha}{72}\left(\left\|x_{k+1} - x^* + \frac{6}{\sqrt{\alpha}}\tilde{y}_{k+1}\right\|^2 - \left\|x_k - x^* + \frac{6}{\sqrt{\alpha}}y_k\right\|^2\right)$$

$$+ \frac{\alpha^{3/2} h}{72} \left( \|x_{k+1} - x^*\|^2 - \left\| x_{k+1} - x^* + \frac{6}{\sqrt{\alpha}} \tilde{y}_{k+1} \right\|^2 \right)$$

$$= \underbrace{f(x_{k+1}) - f(x_k)}_{\mathsf{I}} + \underbrace{\frac{\alpha}{36} \left\langle x_{k+1} - x_k + \frac{6}{\sqrt{\alpha}} (\tilde{y}_{k+1} - y_k), x_{k+1} - x^* + \frac{6}{\sqrt{\alpha}} \tilde{y}_{k+1} \right\rangle}_{\mathsf{II}}$$

$$\underbrace{- \frac{\alpha}{72} \left\| x_{k+1} - x_k + \frac{6}{\sqrt{\alpha}} (\tilde{y}_{k+1} - y_k) \right\|^2}_{\mathsf{III}} - \frac{\alpha h}{6} \langle x_{k+1} - x^*, \tilde{y}_{k+1} \rangle - \frac{\sqrt{\alpha} h}{2} \|\tilde{y}_{k+1}\|^2.$$

For I, we apply $\alpha$-strong convexity and (24) to obtain

$$\mathsf{I} \le \langle \nabla f(x_{k+1}), x_{k+1} - x_k \rangle - \frac{\alpha}{2} \|x_{k+1} - x_k\|^2$$

$$= \langle \nabla f(x_{k+1}), h\tilde{y}_{k+1} + \mathcal{G}_{k+1}^h \rangle - \frac{\alpha}{2} \|h\tilde{y}_{k+1} + \mathcal{G}_{k+1}^h\|^2$$

$$= h \langle \nabla f(x_{k+1}), \tilde{y}_{k+1} \rangle - \frac{\alpha h^2}{2} \|\tilde{y}_{k+1}\|^2 \tag{26}$$

$$+ \langle \nabla f(x_{k+1}), \mathcal{G}_{k+1}^h \rangle - \alpha h \langle \tilde{y}_{k+1}, \mathcal{G}_{k+1}^h \rangle - \frac{\alpha}{2} \|\mathcal{G}_{k+1}^h\|^2. \tag{27}$$

For II, we apply (24) and (2b) to obtain

$$\mathsf{II} = \frac{\alpha h}{36} \left\langle \tilde{y}_{k+1} + \frac{1}{h} \mathcal{G}_{k+1}^h - \frac{6}{\sqrt{\alpha}} \nabla f(x_{k+1}), x_{k+1} - x^* + \frac{6}{\sqrt{\alpha}} \tilde{y}_{k+1} \right\rangle$$

$$= \frac{\alpha h}{36} \langle \tilde{y}_{k+1}, x_{k+1} - x^* \rangle + \frac{\sqrt{\alpha} h}{6} \|\tilde{y}_{k+1}\|^2 + \frac{\sqrt{\alpha} h}{6} \langle \nabla f(x_{k+1}), x^* - x_{k+1} \rangle - h \langle \nabla f(x_{k+1}), \tilde{y}_{k+1} \rangle \tag{28}$$

$$+ \frac{\alpha}{36} \langle \mathcal{G}_{k+1}^h, x_{k+1} - x^* \rangle + \frac{\sqrt{\alpha}}{6} \langle \mathcal{G}_{k+1}^h, \tilde{y}_{k+1} \rangle. \tag{29}$$

For III, we apply updates (24) and (2b) and expand to obtain

$$\mathsf{III} = -\frac{\alpha h^2}{72} \left\| \tilde{y}_{k+1} + \frac{1}{h} \mathcal{G}_{k+1}^h - \frac{6}{\sqrt{\alpha}} \nabla f(x_{k+1}) \right\|^2$$

$$= -\frac{\alpha h^2}{72} \|\tilde{y}_{k+1}\|^2 + \frac{\sqrt{\alpha} h^2}{6} \langle \nabla f(x_{k+1}), \tilde{y}_{k+1} \rangle - \frac{h^2}{2} \|\nabla f(x_{k+1})\|^2$$

$$- \frac{\alpha h^2}{36} \left\langle \tilde{y}_{k+1} - \frac{6}{\sqrt{\alpha}} \nabla f(x_{k+1}), \frac{1}{h} \mathcal{G}_{k+1}^h \right\rangle - \frac{\alpha}{72} \|\mathcal{G}_{k+1}^h\|^2$$

$$= -\frac{\alpha h^2}{72} \|\tilde{y}_{k+1}\|^2 + \frac{\sqrt{\alpha} h^2}{6} \langle \nabla f(x_{k+1}), \tilde{y}_{k+1} \rangle - \frac{h^2}{2} \|\nabla f(x_{k+1})\|^2 \tag{30}$$

$$- \frac{\alpha h}{36} \langle \tilde{y}_{k+1}, \mathcal{G}_{k+1}^h \rangle + \frac{\sqrt{\alpha} h}{6} \langle \nabla f(x_{k+1}), \mathcal{G}_{k+1}^h \rangle - \frac{\alpha}{72} \|\mathcal{G}_{k+1}^h\|^2. \tag{31}$$

Note that (26), (28) and (30) keep the same as upper bounds of I, II and III in the proof of Theorem 14 (see Section G.1.1), and thus we have

$$(26) + (28) + (30) \le -\frac{\sqrt{\alpha} h}{6} \mathcal{L}_{k+1} - \frac{13\alpha^{3/2} h^2}{432} \|x^* - x_{k+1}\|^2 - \left( \frac{\sqrt{\alpha} h}{6} + \frac{\alpha h^2}{2} \right) \|\tilde{y}_{k+1}\|^2$$

$$- \frac{5h^2}{12} \|\nabla f(x_{k+1})\|^2.$$

(27), (29) and (31) can be viewed as the error terms. Collecting the error terms, we obtain

$$(27) + (29) + (31) = \langle \nabla f(x_{k+1}), \mathcal{G}_{k+1}^h \rangle + \frac{\sqrt{\alpha} h}{6} \langle \nabla f(x_{k+1}), \mathcal{G}_{k+1}^h \rangle \tag{32}$$

$$+ \frac{\sqrt{\alpha}}{6} \langle \tilde{y}_{k+1}, \mathcal{G}_{k+1}^h \rangle - \alpha h \langle \tilde{y}_{k+1}, \mathcal{G}_{k+1}^h \rangle - \frac{\alpha h}{36} \langle \tilde{y}_{k+1}, \mathcal{G}_{k+1}^h \rangle \tag{33}$$

$$+ \frac{\alpha}{36}\langle \mathcal{G}_{k+1}^h, x_{k+1} - x^* \rangle - \frac{37\alpha}{72}\|\mathcal{G}_{k+1}^h\|^2. \tag{34}$$

Applying Lemma 16, we can upper bound (32), (33) and (34) respectively:

$$(32) = \langle \nabla f(x_{k+1}), \mathcal{G}_{k+1}^h \rangle + \frac{1}{6}\langle h\nabla f(x_{k+1}), \sqrt{\alpha}\mathcal{G}_{k+1}^h \rangle$$

$$\leq \frac{h^2}{8}\|\nabla f(x_{k+1})\|^2 + \frac{2}{h^2}\|\mathcal{G}_{k+1}^h\|^2 + \frac{h^2}{12}\|\nabla f(x_{k+1})\|^2 + \frac{\alpha}{12}\|\mathcal{G}_{k+1}^h\|^2$$

$$= \frac{5h^2}{24}\|\nabla f(x_{k+1})\|^2 + \left(\frac{2}{h^2} + \frac{\alpha}{12}\right)\|\mathcal{G}_{k+1}^h\|^2.$$

$$(33) = \frac{\sqrt{\alpha}}{6}\langle \tilde{y}_{k+1}, \mathcal{G}_{k+1}^h \rangle + \alpha\left\langle (-h)\tilde{y}_{k+1}, \mathcal{G}_{k+1}^h \right\rangle + \frac{\alpha}{36}\langle (-h)\tilde{y}_{k+1}, \mathcal{G}_{k+1}^h \rangle$$

$$\leq \frac{\sqrt{\alpha}h}{6}\|\tilde{y}_{k+1}\|^2 + \frac{\sqrt{\alpha}}{24h}\|\mathcal{G}_{k+1}^h\|^2 + \frac{\alpha h^2}{4}\|\tilde{y}_{k+1}\|^2 + \alpha\|\mathcal{G}_{k+1}^h\|^2 + \frac{\alpha h^2}{72}\|\tilde{y}_{k+1}\|^2 + \frac{\alpha}{72}\|\mathcal{G}_{k+1}^h\|^2$$

$$= \left(\frac{\sqrt{\alpha}h}{6} + \frac{19\alpha h^2}{72}\right)\|\tilde{y}_{k+1}\|^2 + \left(\frac{\sqrt{\alpha}}{24h} + \alpha + \frac{\alpha}{72}\right)\|\mathcal{G}_{k+1}^h\|^2.$$

$$(34) = \frac{1}{36}\left\langle \alpha^{1/4}h^{-1}\mathcal{G}_{k+1}^h, \alpha^{3/4}h(x_{k+1} - x^*) \right\rangle - \frac{37\alpha}{72}\|\mathcal{G}_{k+1}^h\|^2$$

$$\leq \frac{\sqrt{\alpha}}{72h^2}\|\mathcal{G}_{k+1}^h\|^2 + \frac{\alpha^{3/2}h^2}{72}\|x^* - x_{k+1}\|^2 - \frac{37\alpha}{72}\|\mathcal{G}_{k+1}^h\|^2.$$

Since $\gamma_k \cdot h = \sqrt{\alpha}h \leq 1$ and $\alpha \leq 1$, we have

$$(27) + (29) + (31) \leq \frac{5h^2}{24}\|\nabla f(x_{k+1})\|^2 + \left(\frac{\sqrt{\alpha}h}{6} + \frac{19\alpha h^2}{72}\right)\|\tilde{y}_{k+1}\|^2 + \frac{\alpha^{3/2}h^2}{72}\|x^* - x_{k+1}\|^2$$

$$+ \left(\frac{2}{h^2} + \frac{7\alpha}{12} + \frac{\sqrt{\alpha}}{24h} + \frac{\sqrt{\alpha}}{72h^2}\right)\|\mathcal{G}_{k+1}^h\|^2$$

$$= \frac{5h^2}{24}\|\nabla f(x_{k+1})\|^2 + \left(\frac{\sqrt{\alpha}h}{6} + \frac{19\alpha h^2}{72}\right)\|\tilde{y}_{k+1}\|^2 + \frac{\alpha^{3/2}h^2}{72}\|x^* - x_{k+1}\|^2$$

$$+ \frac{144 + 42\alpha h^2 + 3\sqrt{\alpha}h + \sqrt{\alpha}}{72h^2}\|\mathcal{G}_{k+1}^h\|^2$$

$$\leq \frac{5h^2}{24}\|\nabla f(x_{k+1})\|^2 + \left(\frac{\sqrt{\alpha}h}{6} + \frac{19\alpha h^2}{72}\right)\|\tilde{y}_{k+1}\|^2 + \frac{\alpha^{3/2}h^2}{72}\|x^* - x_{k+1}\|^2$$

$$+ \frac{95}{36h^2}\|\mathcal{G}_{k+1}^h\|^2.$$

Combining the upper bounds of $(26) + (28) + (30)$ and $(27) + (29) + (31)$, we obtain

$$\mathcal{L}_{k+1} - \mathcal{L}_k \leq -\frac{\sqrt{\alpha}h}{6}\mathcal{L}_{k+1} - \frac{7\alpha^{3/2}h^2}{432}\|x^* - x_{k+1}\|^2 - \frac{17\alpha h^2}{72}\|\tilde{y}_{k+1}\|^2 - \frac{5h^2}{24}\|\nabla f(x_{k+1})\|^2$$

$$+ \frac{95}{36h^2}\|\mathcal{G}_{k+1}^h\|^2$$

$$\leq -\frac{\sqrt{\alpha}h}{6}\mathcal{L}_{k+1} + \frac{95}{36h^2}\|\mathcal{G}_{k+1}^h\|^2 - \frac{5h^2}{24}\|\nabla f(x_{k+1})\|^2. \tag{35}$$

For RHGD, we first choose $h \leq \frac{1}{4^{1/8}\sqrt{L}}$. Then applying Proposition 1 and Proposition 2 yields

$$\|\mathcal{G}_{k+1}^h\|^2 \leq 2 \cdot \left(1 + \frac{1}{2}\right)\|x_{k+1} - \hat{x}_{k+1}\|^2 \leq 3\|x_{k+1} - \hat{x}_{k+1}\|^2 \leq \frac{6L^2h^8}{1 - 2L^4h^8}\|\nabla f(x_{k+1})\|^2.$$

If we choose $h \leq \frac{1}{4\sqrt{L}}$, we have $\frac{95L^2h^4}{6(1-2L^4h^8)} \leq \frac{5}{24}$. Thus we obtain from (35):

$$\mathcal{L}_{k+1} - \mathcal{L}_k \leq -\frac{\sqrt{\alpha}h}{6}\mathcal{L}_{k+1} - h^2\left(\frac{5}{24} - \frac{95L^2h^4}{6(1 - 2L^4h^8)}\right)\|\nabla f(x_{k+1})\|^2 \leq -\frac{\sqrt{\alpha}h}{6}\mathcal{L}_{k+1},$$

which implies

$$\mathbb{E}[f(x_k) - f(x^*)] \leq \mathcal{L}_k \leq \left(1 + \frac{\sqrt{\alpha}h}{6}\right)^{-k} \mathcal{L}_0.$$

$\square$

### G.3.2  Proof of Corollary 1

*Proof.* If we choose $h = \frac{1}{4\sqrt{L}}$, then we obtain from Theorem 4

$$\mathbb{E}[f(x_K) - f(x^*)] \leq \left(1 + \frac{1}{24}\sqrt{\frac{\alpha}{L}}\right)^{-K} \mathcal{L}_0 = \left(1 + \frac{1}{24\sqrt{\kappa}}\right)^{-K} \mathcal{L}_0,$$

where $\mathcal{L}_0 = \mathbb{E}\left[f(x_0) - f(x^*) + \frac{\alpha}{72}\|x_0 - x^*\|^2\right]$. Let

$$\mathbb{E}[f(x_K) - f(x^*)] \leq \left(1 + \frac{1}{24\sqrt{\kappa}}\right)^{-K} \mathcal{L}_0 = \left(1 - \frac{1}{24\sqrt{\kappa} + 1}\right)^{K} \mathcal{L}_0 \leq \exp\left(-\frac{K}{24\sqrt{\kappa} + 1}\right) \mathcal{L}_0 \leq \varepsilon,$$

which solves

$$K \geq (24\sqrt{\kappa} + 1) \log \frac{\mathcal{L}_0}{\varepsilon}.$$

$\square$

### G.3.3  Proof of Theorem 5 (convergence of **RHGD** under weak convexity)

**Theorem 5.** *Assume $f$ is weakly convex and $L$-smooth. Then for all $k \geq 0$, RHGD (Algorithm 3) with $h \leq \frac{1}{8\sqrt{L}}$, $\gamma_k = \frac{17}{2(k+9)h}$ and any $x_0 \in \mathbb{R}^d$ satisfies*

$$\mathbb{E}[f(x_k) - f(x^*)] \leq \frac{14 \cdot \mathbb{E}\left[\|x_0 - x^*\|^2\right]}{h^2(k + 8)^2}.$$

*Proof.* Consider the following Lyapunov function

$$\tilde{E}_k = \mathbb{E}\left[\frac{h^2(k+8)^2}{9}(f(x_k) - f(x^*)) + \frac{1}{2}\left\|x_k - x^* + \frac{(k+8)h}{3}y_k\right\|^2 + \frac{3}{4}\|x_k - x^*\|^2\right].$$

We will bound $\tilde{E}_{k+1} - \tilde{E}_k$ in terms of $\tilde{E}_{k+1}$, similar to the proof of Theorem 15 in Section G.1.2. We follow the calculation in the proof of Theorem 15 and **hide $\mathbb{E}$ below for simplicity:**

$$\tilde{E}_{k+1} - \tilde{E}_k = \frac{h^2(k+9)^2}{9}(f(x_{k+1}) - f(x^*)) - \frac{h^2(k+8)^2}{9}(f(x_k) - f(x^*))$$

$$+ \frac{1}{2}\left\|x_{k+1} - x^* + \frac{(k+9)h}{3}\tilde{y}_{k+1}\right\|^2 - \frac{1}{2}\left\|x_k - x^* + \frac{(k+8)h}{3}\tilde{y}_k\right\|^2$$

$$+ \frac{3}{4}\|x_{k+1} - x^*\|^2 - \frac{3}{4}\|x_k - x^*\|^2 + \frac{17}{4(k+9)}\left(\|x_{k+1} - x^*\|^2 - \left\|x_{k+1} - x^* + \frac{(k+9)h}{3}\tilde{y}_{k+1}\right\|^2\right)$$

$$= \underbrace{\left(\frac{(k+9)^2h^2}{9} - \frac{(k+8)^2h^2}{9}\right)(f(x_{k+1}) - f(x^*))}_{\text{I}} + \underbrace{\frac{(k+8)^2h^2}{9}\left(f(x_{k+1}) - f(x_k)\right)}_{\text{II}}$$

$$+ \underbrace{\left\langle x_{k+1} - x_k + \frac{(k+9)h}{3}\tilde{y}_{k+1} - \frac{(k+8)h}{3}y_k, x_{k+1} - x^* + \frac{(k+9)h}{3}\tilde{y}_{k+1}\right\rangle}_{\text{III}}$$

$$\underbrace{-\frac{1}{2}\left\|x_{k+1} - x_k + \frac{(k+9)h}{3}\tilde{y}_{k+1} - \frac{(k+8)h}{3}y_k\right\|^2}_{\text{IV}} + \underbrace{\frac{3}{2}\langle x_{k+1} - x_k, x_{k+1} - x^*\rangle - \frac{3}{4}\|x_{k+1} - x_k\|^2}_{\text{V}}$$

$$-\frac{17h}{6}\langle x_{k+1} - x^*, \tilde{y}_{k+1}\rangle - \frac{17(k+9)h^2}{36}\|\tilde{y}_{k+1}\|^2.$$

For I, it simplifies to

$$\mathsf{I} = \frac{(2k+17)h^2}{9}(f(x_{k+1}) - f(x^*)).$$

For II, we apply weak convexity and (24) to obtain

$$\mathsf{II} \leq \frac{(k+8)^2h^2}{9}\langle \nabla f(x_{k+1}), x_{k+1} - x_k\rangle = \frac{(k+8)^2h^3}{9}\langle \nabla f(x_{k+1}), \tilde{y}_{k+1}\rangle + \frac{(k+8)^2h^2}{9}\langle \nabla f(x_{k+1}), \mathcal{G}_{k+1}^h\rangle.$$

For III, we apply (24) and (2b) to obtain

$$\mathsf{III} = \left\langle h\tilde{y}_{k+1} + \mathcal{G}_{k+1}^h + \frac{h}{3}\tilde{y}_{k+1} - \frac{(k+8)h^2}{3}\nabla f(x_{k+1}), x_{k+1} - x^* + \frac{(k+9)h}{3}\tilde{y}_{k+1}\right\rangle$$

$$= \frac{4h}{3}\langle \tilde{y}_{k+1}, x_{k+1} - x^*\rangle + \frac{4(k+9)h^2}{9}\|\tilde{y}_{k+1}\|^2 + \frac{(k+8)h^2}{3}\langle \nabla f(x_{k+1}), x^* - x_{k+1}\rangle$$

$$- \frac{(k+8)(k+9)h^3}{9}\langle \nabla f(x_{k+1}), \tilde{y}_{k+1}\rangle + \langle \mathcal{G}_{k+1}^h, x_{k+1} - x^*\rangle + \frac{(k+9)h}{3}\langle \mathcal{G}_{k+1}^h, \tilde{y}_{k+1}\rangle.$$

For IV, we apply updates (2a) and (2b) and expand to obtain

$$\mathsf{IV} = -\frac{8h^2}{9}\|\tilde{y}_{k+1}\|^2 + \frac{4(k+8)h^3}{9}\langle \nabla f(x_{k+1}), \tilde{y}_{k+1}\rangle - \frac{(k+8)^2h^4}{18}\|\nabla f(x_{k+1})\|^2$$

$$- \frac{4h}{3}\langle \mathcal{G}_{k+1}^h, \tilde{y}_{k+1}\rangle + \frac{(k+8)h^2}{3}\langle \mathcal{G}_{k+1}^h, \nabla f(x_{k+1})\rangle - \frac{1}{2}\|\mathcal{G}_{k+1}^h\|^2.$$

For V, we apply update (2a) to obtain

$$\mathsf{V} = \frac{3h}{2}\langle \tilde{y}_{k+1}, x_{k+1} - x^*\rangle - \frac{3h^2}{4}\|\tilde{y}_{k+1}\|^2 + \frac{3}{2}\langle \mathcal{G}_{k+1}^h, x_{k+1} - x^*\rangle - \frac{3h}{2}\langle \mathcal{G}_{k+1}^h, \tilde{y}_{k+1}\rangle - \frac{3}{4}\|\mathcal{G}_{k+1}^h\|^2.$$

Note that we obtain the same terms as in the proof of Theorem 15 (see Appendix G.1) and the error terms involving $\mathcal{G}_{k+1}^h$. For the error terms, we apply Lemma 16 to obtain

$$\mathsf{err} = \frac{(k+8)^2h^2}{9}\langle \mathcal{G}_{k+1}^h, \nabla f(x_{k+1})\rangle + \frac{(k+8)h^2}{3}\langle \mathcal{G}_{k+1}^h, \nabla f(x_{k+1})\rangle + \frac{(k+9)h}{3}\langle \mathcal{G}_{k+1}^h, \tilde{y}_{k+1}\rangle$$

$$- \frac{17h}{6}\langle \mathcal{G}_{k+1}^h, \tilde{y}_{k+1}\rangle + \frac{5}{2}\langle \mathcal{G}_{k+1}^h, x_{k+1} - x^*\rangle - \frac{5}{4}\|\mathcal{G}_{k+1}^h\|^2$$

$$= \frac{(k+8)^2}{9}\langle \mathcal{G}_{k+1}^h, h^2\nabla f(x_{k+1})\rangle + \frac{1}{3}\langle (k+8)\mathcal{G}_{k+1}^h, h^2\nabla f(x_{k+1})\rangle + \frac{1}{3}\langle (k+9)\mathcal{G}_{k+1}^h, h\tilde{y}_{k+1}\rangle$$

$$+ \frac{17}{6}\langle \mathcal{G}_{k+1}^h, (-h)\tilde{y}_{k+1}\rangle + \frac{5}{2}\left\langle (k+9)\mathcal{G}_{k+1}^h, \frac{1}{k+9}(x_{k+1} - x^*)\right\rangle - \frac{5}{4}\|\mathcal{G}_{k+1}^h\|^2$$

$$\leq \frac{4(k+8)^2}{9}\|\mathcal{G}_{k+1}^h\|^2 + \frac{(k+8)^2h^4}{144}\|\nabla f(x_{k+1})\|^2 + 4\|\mathcal{G}_{k+1}^h\|^2 + \frac{(k+8)^2h^4}{144}\|\nabla f(x_{k+1})\|^2$$

$$+ \frac{(k+9)^2}{6}\|\mathcal{G}_{k+1}^h\|^2 + \frac{h^2}{6}\|\tilde{y}_{k+1}\|^2 + \frac{17}{4}\|\mathcal{G}_{k+1}^h\|^2 + \frac{17h^2}{36}\|\tilde{y}_{k+1}\|^2 + \frac{25(k+9)^2}{12}\|\mathcal{G}_{k+1}^h\|^2$$

$$+ \frac{3}{4(k+9)^2}\|x_{k+1} - x^*\|^2 - \frac{5}{4}\|\mathcal{G}_{k+1}^h\|^2$$

$$\leq \frac{(k+8)^2h^4}{72}\|\nabla f(x_{k+1})\|^2 + \frac{23h^2}{36}\|\tilde{y}_{k+1}\|^2 + \frac{1}{(k+9)^2}\tilde{E}_{k+1} + \left(\frac{4(k+8)^2}{9} + \frac{9(k+9)^2}{4} + 7\right)\|\mathcal{G}_{k+1}^h\|^2$$

$$\leq \frac{(k+8)^2h^4}{72}\|\nabla f(x_{k+1})\|^2 + \frac{23h^2}{36}\|\tilde{y}_{k+1}\|^2 + \frac{1}{(k+9)^2}\tilde{E}_{k+1} + \left(\frac{40(k+8)^2}{9} + 7\right)\|\mathcal{G}_{k+1}^h\|^2$$

$$\leq \frac{(k+8)^2h^4}{72}\|\nabla f(x_{k+1})\|^2 + \frac{23h^2}{36}\|\tilde{y}_{k+1}\|^2 + \frac{1}{(k+9)^2}\tilde{E}_{k+1} + \frac{41(k+8)^2}{9}\|\mathcal{G}_{k+1}^h\|^2.$$

where the third inequality follows from the relation: $k + 9 \leq \frac{4}{3}(k + 8)$ and the last inequality follows from the relation: $7 \leq \frac{(k+8)^2}{9}$ for $k \geq 0$. Thus, combining the calculation in the proof of Theorem 15 and the calculation above, we obtain

$$\tilde{E}_{k+1} - \tilde{E}_k \leq -\frac{(k+8)^2 h^4}{36}\|\nabla f(x_{k+1})\|^2 - \frac{23h^2}{36}\|\tilde{y}_{k+1}\|^2 - \frac{(k+9)h^2}{36}\|\tilde{y}_{k+1}\|^2 + \text{err}$$

$$\leq -\frac{(k+8)^2 h^4}{72}\|\nabla f(x_{k+1})\|^2 + \frac{41(k+8)^2}{9}\|\mathcal{G}_{k+1}^h\|^2 + \frac{1}{(k+9)^2}\tilde{E}_{k+1}. \qquad (36)$$

Applying Propositions 2 and 1 with the choice $h \leq \frac{1}{7\sqrt{L}} \leq \frac{1}{4^{1/8}\sqrt{L}}$, we have

$$\frac{41(k+8)^2}{9}\|\mathcal{G}_{k+1}^h\|^2 \leq \frac{164(k+8)^2}{9} \cdot (1 + L^2 h^4) \cdot \frac{L^2 h^8}{1 - 2L^4 h^8} \cdot \|\nabla f(x_{k+1})\|^2$$

$$\leq \frac{82(k+8)^2 h^4}{3} \cdot \frac{L^2 h^4}{1 - 2L^4 h^8} \cdot \|\nabla f(x_{k+1})\|^2$$

$$\leq \frac{(k+8)^2 h^4}{72}\|\nabla f(x_{k+1})\|^2,$$

where the second inequality follows from $h \leq \frac{1}{4^{1/8}\sqrt{L}}$ and the third inequality follows from $h \leq \frac{1}{7\sqrt{L}}$. Combining with (36), we have

$$\tilde{E}_{k+1} - \tilde{E}_k \leq \frac{1}{(k+9)^2}\tilde{E}_{k+1} \quad \Rightarrow \quad \frac{\tilde{E}_{k+1}}{\tilde{E}_k} \leq \frac{(k+9)^2}{(k+8)(k+10)}.$$

Taking the product, we obtain

$$\tilde{E}_k = \left(\frac{\tilde{E}_k}{\tilde{E}_{k-1}} \cdot \frac{\tilde{E}_{k-1}}{\tilde{E}_{k-2}} \cdot \ldots \cdot \frac{\tilde{E}_1}{\tilde{E}_0}\right) \cdot \tilde{E}_0 \leq \left(\frac{(k+8)^2}{(k+7)(k+9)} \cdot \frac{(k+7)^2}{(k+6)(k+8)} \cdots \frac{9^2}{8 \cdot 10}\right) \cdot \tilde{E}_0$$

$$= \frac{9(k+8)}{8(k+9)}\tilde{E}_0 \leq \frac{9}{8}\tilde{E}_0.$$

This implies

$$\mathbb{E}[f(x_k) - f(x^*)] \leq \frac{9}{h^2(k+8)^2} \cdot \frac{9}{8} \cdot \tilde{E}_0$$

$$= \frac{9}{h^2(k+8)^2} \cdot \frac{9}{8} \cdot \mathbb{E}\left[\frac{64h^2}{9}(f(x_0) - f(x^*)) + \frac{5}{4}\|x_0 - x^*\|^2\right]$$

$$\leq \frac{\mathbb{E}\left[72h^2(f(x_0) - f(x^*)) + 13\|x_0 - x^*\|^2\right]}{h^2(k+8)^2}$$

$$\leq \frac{\mathbb{E}\left[36Lh^2\|x_0 - x^*\|^2 + 13\|x_0 - x^*\|^2\right]}{h^2(k+8)^2}$$

$$\leq \frac{14 \cdot \mathbb{E}\left[\|x_0 - x^*\|^2\right]}{h^2(k+8)^2},$$

where the third inequality follows from $L$-smoothness of $f$ and the last inequality follows from $h \leq \frac{1}{7\sqrt{L}}$. $\qquad \square$

### G.3.4   Proof of Corollary 2

*Proof.* If we choose $h = \frac{1}{7\sqrt{L}}$, we obtain from Theorem 5

$$\mathbb{E}[f(x_K) - f(x^*)] \leq \frac{686L \cdot \mathbb{E}\left[\|x_0 - x^*\|^2\right]}{(K+8)^2}.$$

Let

$$\mathbb{E}[f(x_K) - f(x^*)] \leq \frac{686L \cdot \mathbb{E}\left[\|x_0 - x^*\|^2\right]}{(K+8)^2} \leq \frac{686L \cdot \mathbb{E}\left[\|x_0 - x^*\|^2\right]}{K^2} \leq \varepsilon,$$

and the last inequality is satisfied whenever

$$K \geq \sqrt{\frac{686L \cdot \mathbb{E}\left[\|x_0 - x^*\|^2\right]}{\varepsilon}}.$$

$\square$

# H  Additional experimental details

In this section, we provide some experimental details as a supplement of Section 5. All experiments were implemented in Python 3.10.12 and executed with the default CPU runtime. No GPU or specialized hardware was used. Most experiments complete within one minute, while some high-condition-number or fine-resolution runs (e.g., quadratic minimization with $\kappa = 10^7$) take up to 5-6 minutes. The total compute required is modest and can be reproduced on standard CPU-based environments.

## H.1  Quadratic minimization

To generate a symmetric positive semi-definite (SPSD) matrix with a prescribed condition number, we construct a matrix $A = Q\Lambda Q^\top$, where $Q \in \mathbb{R}^{d \times d}$ is a random orthogonal matrix obtained via QR decomposition of a standard Gaussian matrix, and $\Lambda = \mathrm{diag}(\lambda_1, \ldots, \lambda_d)$ is a diagonal matrix with eigenvalues linearly spaced between $\alpha$ and $L$. This construction ensures that $A$ is SPSD with spectrum controlled by the given maximum and minimum eigenvalues.

### H.1.1  Comparison with baseline algorithms

In the main paper, due to space constraints, we reported only a subset of the experiments for quadratic minimization: (1) $\kappa = 10^7$ with exact $\alpha$, (2) $\kappa = 10^7$ with overestimated $\alpha = 0.01$, and (3) $\alpha = 0$ with $L = 500$. Here, we include the complete set of experiments to provide a more comprehensive evaluation. Specifically, we consider:

- (1) $\kappa \in \{10^3, 10^5, 10^7\}$, $L = 500$ with exact $\alpha = L/\kappa$;
- (2) $\kappa \in \{10^3, 10^5, 10^7\}$, $L = 500$ with overestimated $\hat{\alpha} \in \{0.01, 0.1, 1\}$;
- (3) $\alpha = 0$ with $L \in \{5 \times 10^2, 5 \times 10^3, 5 \times 10^4\}$.

These results allow us to evaluate how the performance of each algorithm scales with different condition numbers and parameter estimation errors.

Figure 3 shows the convergence results for setting (1) with exact $\alpha$ under condition numbers $\kappa \in \{10^3, 10^5\}$. RHGD consistently outperforms GD while being comparable to AGD and CAGD.

Figure 4 presents the results for setting (2), where $\hat{\alpha} \in \{0.1, 1\}$ is overestimated. We observe that while the performance of AGD and CAGD degrade significantly when $\alpha$ is poorly estimated, RHGD maintains robustness across different values of $\alpha$, demonstrating its practical advantage under parameter uncertainty.

Figure 5 provides the results for setting (3), i.e., when $\alpha = 0$ (weakly convex), with smoothness constants $L \in \{5 \times 10^3, 5 \times 10^4\}$. RHGD outperform AGD and CAGD in late iterations while being noticeably faster than GD.

### H.1.2  Refreshment behavior

To better understand the structure and dynamics of our randomized algorithm, we visualize the objective values as a function of Poisson time, reflecting the continuous-time intuition behind velocity refreshment.

In the case of a homogeneous Poisson process, the refreshment probability is constant across iterations, i.e., $\gamma_k = \sqrt{\alpha}$. This corresponds to the strongly convex setting, where refreshment occurs at a uniform rate. We simulate this process by drawing independent inter-arrival times $\Delta T_k \sim \mathrm{Exp}(\gamma_k)$ with fixed rate $\gamma_k = \sqrt{\alpha}$, and define the cumulative Poisson clock as $T_k = \sum_{i=1}^{k} \Delta T_i$. We then plot the objective value against $T_k$, rather than the iteration index. In contrast, the inhomogeneous Poisson

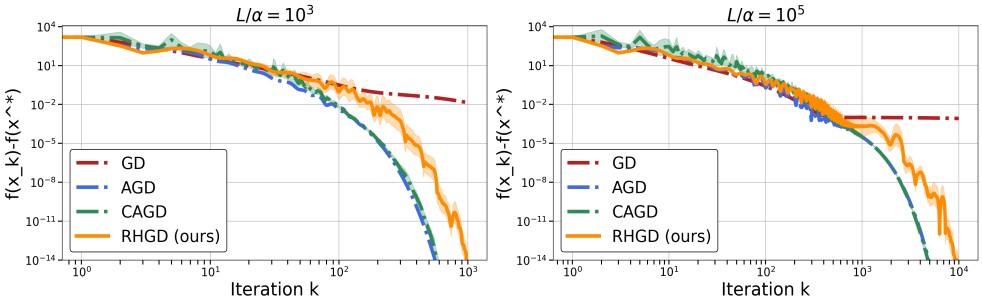

Figure 3: Comparison between GD, AGD, CAGD, and RHGD (ours) on minimizing quadratic functions with $\kappa \in \{10^3, 10^5\}$ and optimal stepsizes via grid search. All algorithms use exact $\alpha$. Each plot shows results averaged over 5 runs.

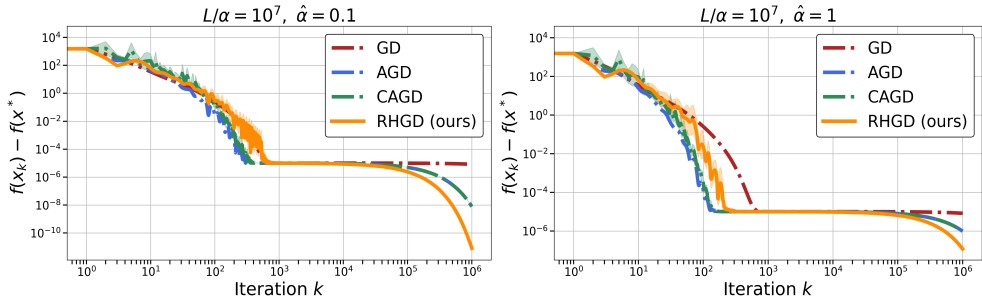

Figure 4: Comparison between GD, AGD CAGD and RHGD (ours) on minimizing the quadratic function with with $\kappa = 10^7$ and optimal stepsizes via grid search. All algorithms use misspecified $\hat{\alpha} \in \{0.1, 1\}$. Each plot shows results averaged over 5 runs.

process reflects the weakly convex setting, where the refreshment probability decays with iteration. Specifically, we consider a time-varying refreshment rate $\gamma_k = \frac{17}{2(k+9)h}$, and simulate the process using non-stationary exponential samples with rate $\gamma_k$, i.e., $\Delta T_k \sim \mathrm{Exp}(\gamma_k)$ and $T_k = \sum_{i=1}^{k} \Delta T_i$. This mimics the decaying refreshment frequency used in the weakly convex regime.

In the strongly convex setting, we choose $\kappa = 10^7$ with $L = 500$, and $h = \sqrt{1/L}$. In the weakly convex setting, we choose $L = 500$ and $h = \sqrt{1/L}$. Then we plot the objective value at each iteration versus the accumulated Poisson time $T_k$, thereby aligning our discrete-time algorithms with their continuous-time interpretations. We also overlay the actual refreshment events as markers on the plot, which allows us to visually compare the algorithm's progress with the stochastic timing of velocity refreshment. These visualizations confirm that our refreshment mechanisms closely match the intended Poisson-driven behavior and provide an intuitive connection between the discrete-time algorithm and their continuous-time limit flow.

### H.1.3 Choice of stepsize

When $\alpha > 0$, we fix the smoothness constant (largest eigenvalue) to be 500, and choose the optimal stepsizes via grid search summarized in Table 2. When $\alpha = 0$, we evaluate different smoothness constant $L$, and choose the optimal stepsizes via grid search summarized in Table 3.

### H.2 Logistic loss minimization

The feature vectors $a_i \in \mathbb{R}^d$ are generated with i.i.d. standard normal entries, and the ground-truth parameter vector $x^* \in \mathbb{R}^d$ is also sampled from standard Gaussian. Binary labels are assigned according to $b_i = \mathrm{sign}(a_i^\top x^* + 0.1 \cdot \xi_i)$, where $\xi_i \sim \mathcal{N}(0, 1)$ adds Gaussian noise to simulate label uncertainty.

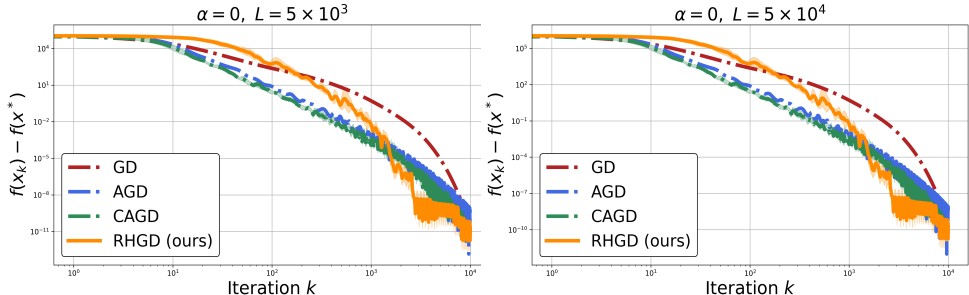

Figure 5: Comparison between GD, AGD CAGD and RHGD with their optimal stepsizes via grid search for minimizing quadratic functions with $\alpha = 0$ and $L \in \{5 \times 10^3, 5 \times 10^4\}$. Each plot shows results averaged over 5 runs.

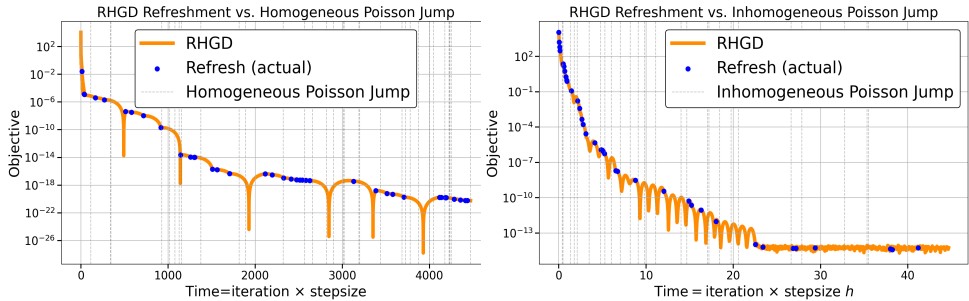

Figure 6: Comparison of actual refresh events in RHGD with theoretical Poisson jump times. Left: $\gamma_k = \sqrt{\alpha}$ vs. homogeneous Poisson process with rate $\sqrt{\alpha}$. Right: $\gamma_k = \frac{17}{2(k+9)h}$ vs. inhomogeneous Poisson process with rate $\gamma(t) = \frac{17}{2t+18h}$. The $x$-axis shows time $t = kh$ with $h = \sqrt{0.002}$. Blue dots: actual refreshes; gray dashed lines: Poisson jumps.

In the main paper, we only report the results of $\alpha \in \{10^{-4}, 0\}$. Here we provide the remaining results corresponding to $\{10^{-3}, 10^{-5}\}$. The setting $\alpha = 10^{-3}$ corresponds to a commonly used regularization level in practice, providing a well-conditioned objective. In contrast, $\alpha = 10^{-5}$ yields a more ill-conditioned problem, which serves as a stress test for the algorithms but is less typical in real-world applications. Figure 7 presents the convergence behavior of GD, AGD, CAGD, and RHGD under both settings. We observe that RHGD consistently outperforms GD and remains competitive with AGD and CAGD across both values of $\alpha$. Notably, RHGD using $\gamma_k = 2\sqrt{\alpha}$ consistently achieves faster convergence than $\gamma_k = \sqrt{\alpha}$. These results further confirm the robustness and practical efficiency of RHGD under varying degrees of strong convexity.

### H.3 Details of baseline algorithms

For each method, we assume the objective function $f$ is $L$-smooth and $\alpha$-strongly convex, where setting $\alpha = 0$ corresponds to weakly convex functions.

| $\kappa$ | GD | AGD | CAGD | RHGD |
|---|---|---|---|---|
| $10^3$ | $\eta = \frac{1}{L}$ | $\eta = \frac{1}{L}$ | $\eta = \frac{1}{L}$ | $h = \frac{1}{\sqrt{L}}$ |
| $10^5$ | $\eta = \frac{1}{L}$ | $\eta = \frac{1}{L}$ | $\eta = \frac{1}{L}$ | $h = \frac{1}{\sqrt{L}}$ |
| $10^7$ | $\eta = \frac{1}{L}$ | $\eta = \frac{1}{L}$ | $\eta = \frac{1}{L}$ | $h = \frac{1}{\sqrt{L}}$ |

Table 2: Stepsizes of GD, AGD, CAGD and RHGD in the strongly convex setting ($\alpha > 0$) under different condition numbers $\kappa$ with fixed $L = 500$.

| $L$ | GD | AGD | CAGD | RHGD |
|---|---|---|---|---|
| $5 \times 10^2$ | $\eta = \frac{1}{L}$ | $\eta = \frac{1}{L}$ | $\eta = \frac{1}{L}$ | $h = \frac{1}{\sqrt{L}}$ |
| $5 \times 10^3$ | $\eta = \frac{1}{8L}$ | $\eta = \frac{1}{16L}$ | $\eta = \frac{1}{8L}$ | $h = \frac{1}{8\sqrt{L}}$ |
| $5 \times 10^4$ | $\eta = \frac{1}{8L}$ | $\eta = \frac{1}{16L}$ | $\eta = \frac{1}{8L}$ | $h = \frac{1}{8\sqrt{L}}$ |

Table 3: Stepsizes of GD, AGD, CAGD and RHGD in the weakly convex setting ($\alpha = 0$) under different smoothness constants $L$.

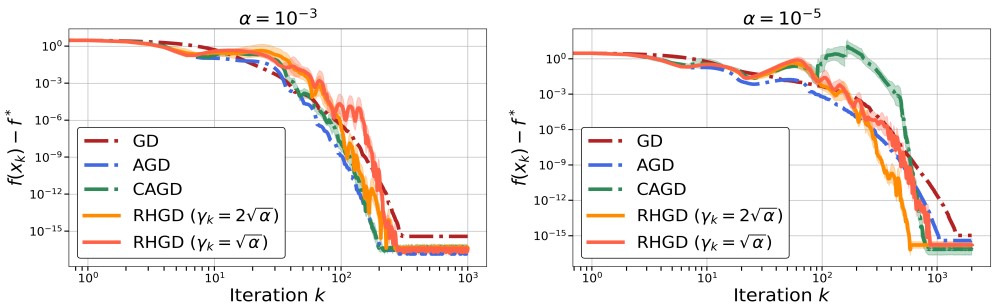

Figure 7: Comparison of GD, AGD, CAGD and RHGD on logistic regression with $\alpha \in \{10^{-3}, 10^{-5}\}$. We run each algorithm using adaptive stepsizes. For RHGD, we evaluate $\gamma_k = \sqrt{\alpha}$ and $\gamma_k = 2\sqrt{\alpha}$.

**Parameter formulas of Algorithm 7.** The update rules in Algorithm 7 depend on parameters $\beta_k$, which differs depending on whether the objective is strongly convex ($\alpha > 0$) or weakly convex ($\alpha = 0$). The specific formula is as follows: $\beta_k = \begin{cases} \frac{1-\sqrt{\alpha\eta}}{1+\sqrt{\alpha\eta}}, & \text{if } \alpha > 0, \\ \frac{k-1}{k+2}, & \text{if } \alpha = 0. \end{cases}$

**Parameter formulas of Algorithm 8.** The update rules in Algorithm 8 depend on parameters $\theta_k$, $\theta_k'$ and $\eta_k$ which differ depending on whether the objective is strongly convex ($\alpha > 0$) or weakly convex ($\alpha = 0$). The specific formulas are summarized in Table 4.

**Adaptive variants.** To improve robustness in practice, we also implement an adaptive stepsize scheme for GD, where the stepsize $\eta_k$ is adjusted multiplicatively based on the observed decrease in objective value:

$$\eta_{k+1} = \begin{cases} 1.1\eta_k, & \text{if } f(x_{k+1}) < f(x_k) - \frac{\eta_k}{2}\|\nabla f(x_k)\|^2 \\ 0.6\eta_k, & \text{otherwise} \end{cases}$$

The same mechanism can be applied to AGD, CAGD and RHGD variants. We present the adaptive variants of GD, AGD, CAGD and RHGD below, which are called Ada-GD, Ada-AGD, Ada-CAGD and Ada-RHGD. For all logistic regression experiments, we initialize $\eta_0 = 1$ and $h_0 = 1$. When $\alpha > 0$, the momentum parameter $\beta_k$ is computed using the updated stepsize $\eta_{k+1}$ as $\beta_k = \frac{1-\sqrt{\alpha\eta_{k+1}}}{1+\sqrt{\alpha\eta_{k+1}}}$, which reflects the dependence of acceleration on the current stepsize. When $\alpha = 0$, the momentum parameter simplifies to $\beta_k = \frac{k-1}{k+2}$, which is independent of the stepsize and follows from Nesterov's acceleration for weakly convex functions.

---

**Algorithm 6 Gradient Descent (GD)**

---
1: Initialize $x_0$. Choose stepsize $0 < \eta \le 1/L$.
2: **for** $k = 0, 1, \ldots, K - 1$ **do**
3:      $x_{k+1} = x_k - \eta\nabla f(x_k)$
4: **end for**
5: **return** $x_K$

---

**Algorithm 7 Accelerated Gradient Descent (AGD)**

---

1: Initialize $x_0$ and $y_0$. Choose stepsize $0 < \eta \le 1/L$.
2: **for** $k = 0, 1, \ldots, K - 1$ **do**
3: $\quad x_{k+1} = y_k - \eta \nabla f(y_k)$
4: $\quad y_{k+1} = x_{k+1} + \beta_k(x_{k+1} - x_k)$
5: **end for**
6: **return** $x_K$

---

**Algorithm 8 Continuized Accelerated Gradient Descent (CAGD)**

---

1: Initialize $x_0$ and $T_0 = 0$. Choose stepsize $0 < \eta \le 1/L$.
2: **for** $k = 0, 1, \ldots, K - 1$ **do**
3: $\quad$ Sample $\tau_k \sim \text{Exp}(1)$ and set $T_{k+1} = T_k + \tau_k$
4: $\quad y_k = x_k + \theta_k(z_k - x_k)$
5: $\quad x_{k+1} = y_k - \eta \nabla f(y_k)$
6: $\quad z_{k+1} = z_k + \theta_k'(y_k - z_k) - \eta_k \nabla f(y_k)$
7: **end for**
8: **return** $x_K$

---

| Parameter | $\alpha > 0$ | $\alpha = 0$ |
|-----------|-------------|-------------|
| $\theta_k$ | $\frac{1}{2}\left(1 - \exp(-2\sqrt{\alpha\eta}\,\tau_k)\right)$ | $1 - \left(\frac{T_k}{T_{k+1}}\right)^2$ |
| $\theta_k'$ | $\tanh\left(\sqrt{\alpha\eta}\,\tau_k\right)$ | $0$ |
| $\eta_k$ | $\sqrt{\frac{\eta}{\alpha}}$ | $\frac{T_k\eta}{2}$ |

Table 4: Parameter choices in Algorithm 8 for different regimes of $\alpha$.

---

**Algorithm 9 Adaptive Gradient Descent (Ada-GD)**

---

1: Initialize $x_0$ and stepsize $\eta_0 > 0$.
2: **for** $k = 0, 1, \ldots, K - 1$ **do**
3: $\quad \tilde{x}_{k+1} = x_k - \eta_k \nabla f(x_k)$
4: $\quad$ **if** $f(\tilde{x}_{k+1}) < f(x_k) - \frac{\eta_k}{2}\|\nabla f(x_k)\|^2$ **then**
5: $\quad\quad \eta_{k+1} = 1.1 \cdot \eta_k$ $\qquad\qquad\qquad\qquad\qquad\qquad$ ▷ Increase stepsize
6: $\quad\quad x_{k+1} = \tilde{x}_{k+1}$ $\qquad\qquad\qquad\qquad\qquad\qquad$ ▷ Accept trial step
7: $\quad$ **else**
8: $\quad\quad \eta_{k+1} = 0.6 \cdot \eta_k$ $\qquad\qquad\qquad\qquad\qquad\qquad$ ▷ Decrease stepsize
9: $\quad\quad x_{k+1} = x_k$ $\qquad\qquad\qquad\qquad\qquad\qquad\qquad$ ▷ Reject trial step
10: $\quad$ **end if**
11: **end for**
12: **return** $x_K$

---

**Algorithm 10 Adaptive Accelerated Gradient Descent (Ada-AGD)**

---

1: Initialize $x_0$, $y_0$, and stepsize $\eta_0 > 0$.
2: **for** $k = 0, 1, \ldots, K - 1$ **do**
3: $\quad \tilde{x}_{k+1} = y_k - \eta_k \nabla f(y_k)$
4: $\quad$ **if** $f(\tilde{x}_{k+1}) < f(y_k) - \frac{\eta_k}{2}\|\nabla f(y_k)\|^2$ **then**
5: $\quad\quad \eta_{k+1} = 1.1 \cdot \eta_k$ $\qquad\qquad\qquad\qquad\qquad\qquad$ ▷ Increase stepsize
6: $\quad\quad x_{k+1} = \tilde{x}_{k+1}$ $\qquad\qquad\qquad\qquad\qquad\qquad$ ▷ Accept trial step
7: $\quad$ **else**
8: $\quad\quad \eta_{k+1} = 0.6 \cdot \eta_k$ $\qquad\qquad\qquad\qquad\qquad\qquad$ ▷ Decrease stepsize
9: $\quad\quad x_{k+1} = x_k$ $\qquad\qquad\qquad\qquad\qquad\qquad\qquad$ ▷ Reject trial step
10: $\quad$ **end if**
11: $\quad y_{k+1} = x_{k+1} + \beta_k(x_{k+1} - x_k)$
12: **end for**
13: **return** $x_K$

---

---

**Algorithm 11** Adaptive Continuized Accelerated Gradient Descent (Ada-CAGD)

---

1: Initialize $x_0$, $z_0 = x_0$, $T_0 = 0$, and stepsize $\eta_0 > 0$.
2: **for** $k = 0, 1, \ldots, K - 1$ **do**
3:      Sample $\tau_k \sim \text{Exp}(1)$ and set $T_{k+1} = T_k + \tau_k$
4:      Compute $\theta_k$ using $\eta_k$, $\tau_k$, $T_k$ and $T_{k+1}$ as shown in Table 4
5:      $y_k = x_k + \theta_k(z_k - x_k)$
6:      $\tilde{x}_{k+1} = y_k - \eta_k \nabla f(y_k)$
7:      **if** $f(\tilde{x}_{k+1}) < f(y_k) - \frac{\eta_k}{2}\|\nabla f(y_k)\|^2$ **then**
8:          $\eta_{k+1} = 1.1 \cdot \eta_k$                                           ▷ Increase stepsize
9:          $x_{k+1} = \tilde{x}_{k+1}$                                      ▷ Accept trial step
10:      **else**
11:          $\eta_{k+1} = 0.6 \cdot \eta_k$                                       ▷ Decrease stepsize
12:          $x_{k+1} = x_k$                                           ▷ Reject trial step
13:      **end if**
14:      Compute $\theta'_k$ and $\eta_k$ using updated $\eta_{k+1}$, $\tau_k$ and $T_k$ as shown in Table 4
15:      $z_{k+1} = z_k + \theta'_k(y_k - z_k) - \eta_k \nabla f(y_k)$
16: **end for**
17: **return** $x_K$

---

---

**Algorithm 12** Adaptive Randomized Hamiltonian Gradient Descent (Ada-RHGD)

---

1: Initialize $x_0$, $y_0 = 0$ and $h_0 > 0$. Choose refreshment parameter $\gamma_k > 0$.
2: **for** $k = 0, 1, \ldots, K - 1$ **do**
3:      $x_{k+\frac{1}{2}} = x_k + h_k y_k$
4:      $\tilde{x}_{k+1} = x_{k+\frac{1}{2}} - h_k^2 \nabla f(x_{k+\frac{1}{2}})$
5:      **if** $f(\tilde{x}_{k+1}) < f(x_{k+\frac{1}{2}}) - \frac{h_k^2}{2}\|\nabla f(x_{k+\frac{1}{2}})\|^2$ **then**
6:          $h_{k+1} = \sqrt{1.1} \cdot h_k$                                     ▷ Increase stepsize
7:          $x_{k+1} = \tilde{x}_{k+1}$                                        ▷ Accept trial step
8:      **else**
9:          $h_{k+1} = \sqrt{0.6} \cdot h_k$                                   ▷ Decrease stepsize
10:          $x_{k+1} = x_k$                                       ▷ Reject trial step
11:      **end if**
12:      $\tilde{y}_{k+1} = y_k - h_{k+1} \nabla f(x_{k+1})$
13:      $y_{k+1} = \begin{cases} \tilde{y}_{k+1} & \text{with probability } 1 - \min\left(\gamma_k h_{k+1}, 1\right) \\ 0 & \text{with probability } \min\left(\gamma_k h_{k+1}, 1\right) \end{cases}$
14: **end for**
15: **return** $x_K$

---

# I    Helpful lemmas

**Lemma 14.** *Let $t \mapsto v_t \in \mathbb{R}^d$ be a family of vector fields and suppose that the random variables $t \mapsto Z_t$ evolve according to*

$$\dot{Z}_t = v_t(Z_t).$$

*Then, the law $\rho_t$ of $Z_t$ evolves according to the continuity equation*

$$\partial_t \rho_t + \nabla \cdot (\rho_t v_t) = 0. \tag{37}$$

*Proof.* Given a smooth test function $\phi : \mathbb{R}^d \to \mathbb{R}$, we obtain by the chain rule:

$$\frac{\mathrm{d}}{\mathrm{d}t}\mathbb{E}[\phi(Z_t)] = \mathbb{E}\left[\langle \nabla \phi(Z_t), \dot{Z}_t \rangle\right] = \mathbb{E}\left[\langle \nabla \phi(Z_t), v_t(Z_t) \rangle\right].$$

For the left hand side, we can write

$$\frac{\mathrm{d}}{\mathrm{d}t}\mathbb{E}[\phi(Z_t)] = \frac{\mathrm{d}}{\mathrm{d}t}\int \rho_t(z)\phi(z)\,\mathrm{d}z = \int \partial_t \rho_t(z)\phi(z)\,\mathrm{d}z.$$

For the right hand side, applying the integration by part, we have

$$\mathbb{E}\left[\langle \nabla\phi(Z_t), v_t(Z_t)\rangle\right] = \int \langle \nabla\phi(z), v_t(z)\rangle \rho_t(z)\,\mathrm{d}z = -\int \nabla\cdot(\rho_t(z)v_t(z))\phi(z)\,\mathrm{d}z.$$

Thus we have

$$\int \partial_t\rho_t(z)\phi(z)\,\mathrm{d}z = -\int \nabla\cdot(\rho_t(z)v_t(z))\phi(z)\,\mathrm{d}z.$$

Since this holds for every test function $\phi$, we obtain $\partial_t\rho_t + \nabla\cdot(\rho_t v_t) = 0$. $\qquad\square$

**Lemma 15** (Three-point identity). *For all $x, y, z \in \mathbb{R}^d$, we have*

$$\|x - y\|^2 - \|x - z\|^2 = -2\langle y - z, x - y\rangle - \|y - z\|^2. \tag{38}$$

*Proof.* Expanding the squared norm and rearranging yields the claimed identity. $\qquad\square$

**Lemma 16** (Young's inequality). *For all $x, y \in \mathbb{R}^d$ and $p > 0$, we have*

$$\langle x, y\rangle \le p\|x\|^2 + \frac{1}{4p}\|y\|^2. \tag{39}$$

*Proof.* We have

$$\left\|\sqrt{p}x - \frac{1}{2\sqrt{p}}y\right\|^2 = p\|x\|^2 + \frac{1}{4p}\|y\|^2 - \langle x, y\rangle \ge 0,$$

which implies the conclusion. $\qquad\square$

**Lemma 17.** *For all $k \ge 0$, we have $\frac{2k+17}{9} \le \frac{k+8}{3}$.*

*Proof.* It suffices to show $\frac{2k+17}{k+8} \le 3$. Since $\frac{1}{k+8} \le 1$, we have $\frac{2k+17}{k+8} = 2 + \frac{1}{k+8} \le 3$. $\qquad\square$

**Lemma 18.** *For all $x, y \in \mathbb{R}^d$, we have*

$$\|x + y\|^2 \le 2\|x\|^2 + 2\|y\|^2. \tag{40}$$

*Proof.* $\|x + y\|^2 - (2\|x\|^2 + 2\|y\|^2) = -\left(\|x\|^2 - 2\langle x, y\rangle + \|y\|^2\right) = -\|x - y\|^2 \le 0.$ $\qquad\square$

**Lemma 19.** *For $0 \le x \le \frac{1}{2}$, we have $\cos^2(x) \le 1 - \frac{x^2}{2}$.*

*Proof.* Let $g(x) = \cos^2(x) + \frac{1}{2}x^2 - 1$. Then $g'(x) = -\sin(2x) + x$ and $g''(x) = -2\cos(2x) + 1$. Since $0 \le x \le \frac{1}{2} \le \frac{\pi}{6}$, we have $\cos(2x) \ge \frac{1}{2}$ which implies $g''(x) \le 0$ for $x \in [0, \frac{1}{2}]$. Thus $g'(x)$ is monotonically decreasing for $x \in [0, \frac{1}{2}]$, which implies $g'(x) \le g'(0) = 0$. Thus $g(x)$ is also monotonically decreasing for $x \in [0, \frac{1}{2}]$. Then we have $g(x) = \cos^2(x) + \frac{1}{2}x^2 - 1 \le g(0) = 0$, which implies $\cos^2(x) \le 1 - \frac{x^2}{2}$ $\qquad\square$

