# OpenReview forum: "Hamiltonian Descent Algorithms for Optimization: Accelerated Rates via Randomized Integration Time"
_NeurIPS.cc/2025/Conference — NeurIPS 2025 spotlight_

### Official Review · Reviewer_ZMTX · 2025-06-26

**Clarity:** 2
**Significance:** 3
**Originality:** 3
**Rating:** 5
**Confidence:** 5

**Summary:**

Let me start with a statement of (what I think is) the main result.
That is that there is an implementable approximation of Hamiltonian dynamics for convex optimization that has a provably accelerated convergence rate.

Now for a summary. It is shown that an idealized version of Hamiltonian flow for convex optimization in discrete time achieves the convergence rate of gradient descent on the same class of functions when the step size (or integration time) is small. Inspired by the work of other authors, who have demonstrated that randomizing the integration time can speed up convergence, the authors show that a continuous-time limit of Hamiltonian flow with appropriately randomized integration times achieves accelerated convergence on convex functions. Presumably motivated by this continuous-time hint, the authors then go on to present a discretization of the continuous-time flow--that is, an implementable algorithm--and prove that the discrete-time dynamics achieves an accelerated convergence rate.

**Questions:**

**Experiments.**
There are a few things wrong with the experiments.
One problem is the range on the y-axes. In the most egregious example, the y-axis goes all the way down to $10^{-77}$, in another it goes to $10^{-23}$, and in many others it goes to $10^{-15}$.
What bit precision is being used in the experiments?
In single precision, machine epsilon is merely $10^{-7}$. In double precision that improves to something like $10^{-15}$.
Values of $f_k-f^{\star}$ that are smaller than machine epsilon are not sensible. At that point it is not the performance of the algorithm so much as cancellation effects that are being measured.
This needs to be fixed.
The second problem is that many of the experiments don't actually show faster convergence for the accelerated algorithm. For convex or strongly convex functions with the strong convexity constant correctly specified, we should definitely see the accelerated algorithm converging faster on average than gradient descent. The story seems to be more complicated than that, in fact it looks in Figure 1 like the two types of algorithms are converging at the same rate. I'm wondering if there is something wrong with the code, or if perhaps the vast range of the y-axis is obscuring any differentials in behavior.
Because of these two problems, there is also a third problem, which is that the experiments as presented simply don't support the claim in the main text (see for example, the paragraph at line 280) that RHGD converges faster than GD.

**Readability**. My other main issue with this paper has to do with readability, which I'm going to break into two pieces: lack of overall narrative, and other choices in the writing.
With what follows and the suggestions I make, my goal is simply to be helpful. My concerns regarding readability did not influence how I scored the paper. I am of the opinion that the quality of the written paper is not on par with the quality of the mathematical results, but I think this can be fixed without much trouble.

Narrative. There isn't a narrative arc in this paper. In the summary, I wrote my impression of what I think the story might be. In the beginning of the paper the authors do a good job of identifying a gap in the literature---how to derive accelerated optimization algorithms from ideas in sampling---that serves as motivation for the main results. But after that the paper is mostly a list of theorem statements. Is the narrative I put in the summary what the authors had in mind (namely, that the fact that the continuous-time limit of randomized integration-time HMC turned out to have an accelerated rate motivated the search for a discretization that could preserve that rate)? If so, it would be worth slipping in a few sentences here and there that make it clear. This would be helpful to the reader in a paper with several algorithms and proofs of convergence. Speaking for myself, although they're cool, I'm just not sure what the place of the continuous-time results (Theorems 2 and 3) is if not to serve as motivation to search for Theorems 3 and 4.

Also along the lines of narrative, I wonder if the authors would consider promoting the discussion of lift-conserve-project schemes to the main paper? Descent lemmas are valuable in optimization. The observation that Hamiltonian dynamics gives a descent lemma for free is really nice. I think it would be worth putting that and the proof of the descent lemma in the main paper. One possibility would be to relegate some of the material in the experiments section to the appendices to make room.

And while I'm talking about LCP schemes---the authors claim that this perspective is new. My question is, new to whom? If they mean the optimization/ML community then that is certainly plausible. But in physics the idea is folklore to the point where I don't think it even has a name. And I would be surprised if it wasn't also known in numerical analysis (but maybe it isn't!). Can the authors please clarify what they meant? Secondly, I think we should be careful not to imply (I think it's happening accidentally) that we can obtain a descent lemma for absolutely any $f$. For Hamilton's equations to be well-defined I believe we need $f$ to be differentiable at the very least.

More about the writing.
None of the math in this paper was new to me but it was still a difficult read. I think that part of the reason why is that the paper is absolutely teeming with acronyms. This really slows a reader down (for example, I got totally lost trying to keep RHMC, RHF, RHF-opt, RPHD, and RHGD straight). I think it would be worth dropping some of the acronyms, reserving them perhaps only for implementable algorithms such as GD and RHGD. There is a similar problem with excessive use of lemmas for well-known results such as Young's inequality. A sentence in a proof that reads ``we go from step A to step B via an application of Young's inequality'' is much faster to read than ''we go from step A to step B via Lemma 16'' and then I must flip several pages to check the statement of Lemma 16.

Apart from that, there are specific sentences and paragraphs that could use some rewriting for clarity. For example, no convergence result is provided for Algorithm 2 (and that's fine), but then the sentence in line 154 should be edited. Another example are the two paragraphs that begin at lines 176 and 185, respectively. We can't directly compare rates of a discrete-time dynamics with a continuous-time dynamics (like HF-opt and RHF in line 176).

**Questions.**
HF-opt is not an implementable algorithm. But having used an approximation of the Prox operator to approximate the Hamiltonian flow in the construction of RHGD, do the authors know what might happen if they did the same thing in HF-opt? Would the convergence rates of Theorem 1 be preserved?

I was surprised by the discussion following the statements of Theorems 2 and 3, specifically the bit about smoothness. And are we sure that smoothness is really needed to prove a convergence rate for HF-opt? The relevant descent lemma does not require it. Could the authors please clarify what they are trying to say there?

**One last thing.** Markov's inequality is quite simple to state and apply. It may be worth doing so in the paper to quantify deviations from the expected convergence rate of RHGD since the possibility of doing so is mentioned twice anyway.

**Ethical Concerns:**

["NO or VERY MINOR ethics concerns only"]

**Final Justification:**

This is mainly a theory paper. The theorems look solid and are relevant to the community. I had some concerns regarding the clarity of the exposition and a number of issues with the experiments, but the authors seem to have understood my concerns and are addressing them. I am raising my score to an accept.

**Limitations:**

Yes.

**Paper Formatting Concerns:**

None.

**Quality:**

3

**Strengths And Weaknesses:**

The mathematical results of this paper are interesting, original, and relevant, and deserve to be published. The experiments need some work (they are not properly implemented or interpreted) and the whole paper could do with some rewriting here and there for clarity. I'll expand on these two points in the next section. Here, let me just say that the reason for my tepid score are the issues with the experiments. If the authors fix the experiments and the associated discussion in the experiments section, I am happy to raise my score to a firm accept.

---

> ### Author Rebuttal · Authors · 2025-07-28
>
> We thank the reviewer for the detailed and constructive feedback. Our responses to the questions and concerns are provided below.
>
> **1. Issues on the experiment**
>
> Regarding the y-axis scale, we agree that values below numerical precision are less meaningful. We re-ran the experiment with y-axis truncation at $10^{-14}$ to reflect numerical precision limits, and as the reviewer suggested, the acceleration of RHGD over GD became more evident. This confirms that the original plots already demonstrate accelerated convergence before reaching $10^{-14}$, particularly in the later stages of optimization. We will incorporate this change in the revised version by truncating the y-axis appropriately to improve visual clarity. Finally, we note that in better-conditioned settings, the advantage of RHGD over GD is more visible even in the early iterations (see Figure 3 in Appendix H.1).
>
> **2. Issue on readability**
>
> We sincerely appreciate the reviewer’s thoughtful and generous feedback regarding the narrative and presentation.
>
> Your interpretation is essentially correct. More precisely, our motivation stemmed from the observation that the randomized HMC exhibits acceleration for Gaussian sampling, and this inspired us to search for an optimization analogue that could preserve similar accelerated behavior to bridge the gap between optimization and sampling. We will revise the introduction and technical sections to make this narrative more explicit and to help guide the reader through the progression of results.
>
> We also appreciate the suggestion to promote the discussion of lift-conserve-project (LCP) schemes and the associated descent lemma to the main body. We agree that this is a compelling and intuitive idea, and we will move that discussion (including the proof) into the main text. To make space, we are happy to shift some experimental details to the appendix as suggested.
>
> We fully agree that the lift-conserve-project (LCP) idea is well known in physics, and possibly in numerical analysis as well. Our claim of novelty is intended relative to the optimization and machine learning communities, where this perspective is, to our knowledge, not widely recognized or explicitly used in algorithm design. One of our goals is to make this viewpoint explicit and show how it leads to useful guarantees. Even within the MCMC literature, while HMC is often motivated by physical principles (e.g., symplecticity), the LCP interpretation offers a clean and concrete explanation for its behavior that is not always emphasized. We also agree that a descent lemma requires regularity in $f$, and we will clarify that differentiability is assumed.
>
> We agree that the excessive use of acronyms may hinder readability, and we will revise the paper to reduce or eliminate non-essential acronyms, keeping them only for key algorithms like GD and RHGD. We also agree that referencing standard results like Young’s inequality through numbered lemmas is unnecessarily indirect. In the revision, we will inline such standard tools where appropriate to improve the flow of proofs.
>
> **3. No convergence result is provided for Algorithm 2**
>
> While we do not provide a direct convergence result for Algorithm 2 as written, we note that it involves exact simulation of Hamiltonian flow and is therefore a piecewise continuous-time process. To enable analysis, we equivalently reformulate it as the RHF process (a combination of Hamiltonian flow and a Poisson clock) and provide convergence guarantees for it in Theorems 2 and 3.
>
> **4. Do the authors know what might happen if they did the same thing in HF-opt? Would the convergence rates of Theorem 1 be preserved?**
>
> If we use the implicit integrator with Prox approximation to discretize HF-opt, we will get gradient descent (similar to the leapfrog integrator shown in Appendix D.2) and then preserve the same convergence rates of Theorem 1. We will include this scheme explicitly in the appendix in the revised version.
>
> **5. And are we sure that smoothness is really needed to prove a convergence rate for HF-opt? The relevant descent lemma does not require it. Could the authors please clarify what they are trying to say there?”**
>
> While Lemma 2 provides an initial descent property, it is not sufficient to establish the convergence rates of HF-opt. For that, we rely on a stronger descent lemma (Lemma 6 in Appendix D), which does require smoothness. We will clarify this point in the revised version.
>
> **6. Markov's inequality is quite simple to state and apply. It may be worth doing so in the paper to quantify deviations from the expected convergence rate of RHGD since the possibility of doing so is mentioned twice anyway.**
>
> We appreciate the reviewer’s observation. We agree that Markov’s inequality yields loose bounds, and our intention was not to claim tightness, but to demonstrate that the convergence in expectation can be converted into a high-probability guarantee without additional assumptions, which is consistent with some prior works, e.g., [1] and [2]. Given the randomized velocity refresh in our algorithm, stronger concentration results would require assumptions such as bounded variance or martingale structure. To keep the analysis general and assumption-free, we opted for the Markov-based bound. We agree that deriving tighter bounds under additional structure is an interesting direction for future work and will clarify this in the revision.
>
> [1] Mathieu Even, et al. "Continuized accelerations of deterministic and stochastic gradient descents, and of gossip algorithms." NeurIPS 2021.
>
> [2] Jun-Kun Wang, and Andre Wibisono. "Continuized Acceleration for Quasar Convex Functions in Non-Convex Optimization." ICLR 2023.

---

> ### Comment · Reviewer_ZMTX · 2025-08-04
> **Responses to authors**
>
> Thank you for responding to my review, answering all my questions, and promising various clarifications in the paper.
>
> Regarding the figures---I am glad you will truncate the y-axes at $10^{-14}$ in the next iterations of the figures. I am still somewhat confused by the observed behavior. In the convex setting, for example, if we plot $\log (f_k-f^{\star})$ as a function of $\log k$, we should expect to see a slope of $-1$ for gradient descent and $-2$ for an accelerated method (or at the very least the slope of the line corresponding to the accelerated method should be twice as steep as the slope of the line corresponding to gradient descent). It's not obvious to me that we are seeing such a clear separation of behaviors between the two types of algorithms in your figures but I'm wondering if perhaps I just don't have the resolution to tell. Can you please clarify? (Are you satisfied that you are seeing what you expect to see?) Thank you.
>
> Regarding the point about Markov's inequality---I didn't mean to comment on the tightness or lack thereof of Markov's inequality, actually. The suggestion was that you may want to state the inequality as it applies to your result for RHGD. But I read through your response to Reviewer DNKQ's questions and I understand that you feel it may not be useful to do this without making additional assumptions that would enable the application of tighter inequalities.

---

> > ### Author Response · Authors · 2025-08-04
> >
> > Thank you for your observation. We would like to clarify that in the weakly convex case experiment, we intentionally constructed a matrix with a zero minimum eigenvalue by setting the smallest eigenvalue to zero during matrix generation. However, due to floating-point precision limitations, the numerical minimum eigenvalue is on the order of $10^{-14}$, which is effectively zero but still allows for linear convergence of gradient descent in practice.
> >
> > To confirm the behavior in the truly weakly convex case, we have re-run the experiment with the smallest eigenvalue explicitly truncated to zero post-construction. This results in the expected sublinear convergence with two distinct slopes for the objective gap, aligning well with our theoretical predictions. We also note that in our logistic regression experiments (Figure 2, $\alpha=0$), the convergence slope aligns more closely with the behavior the reviewer expected, which further supports our interpretation. The slight curvature in Figure 2 is due to the adaptive step size; when we re-ran the experiment with a constant step size, the result aligned precisely with the theoretical behavior.
> >
> > We appreciate your comment, which prompted us to doublecheck and refine our experimental setup! Let us know if you have any further questions or if any clarification would be helpful.

---

> > > ### Comment · Area_Chair_R5ce · 2025-08-06
> > >
> > > Dear Reviewer ZMTX,
> > >
> > > The (extended) deadline for the Author–Reviewer discussion is approaching. Please let us know whether your concerns have been satisfactorily addressed. If not, kindly point out what remains inadequate as soon as possible, so that the authors have time to respond.
> > >
> > > Thanks,
> > > AC

---

> > > ### Comment · Reviewer_ZMTX · 2025-08-06
> > > **Response to the authors**
> > >
> > > Thank you for your note, I'm glad my comment was helpful. It's interesting (and good) that you obtained the expected behavior in the weakly convex case after correcting for numerical imprecision in the smallest eigenvalue.
> > >
> > > I was thinking more about the figures, and I wondered the following. In all or most of your figures I think you are plotting the performance of various optimizers for a fixed number of iterations, so the curves are all cut off at a specific value of $k$. It's possible that this is obscuring differential behaviors at early stages in optimization. If instead you replot the data (in the cases where $\alpha$ is not misspecified) after cutting it off at some reasonably small value of $f_{k}-f^{\star}$, do you see the two types of trajectories (accelerated vs. not) separate out right from init? I hope it's clear what I mean, please ask if not.

---

> ### Author Response · Authors · 2025-08-06
>
> Thank you for the constructive suggestion and insightful question. We re-ran the code for the strongly convex quadratic functions with correct $\alpha$ and **precision truncation**. Our updated observations are as follows:
>
> - For $\kappa = 10^7$, the acceleration becomes noticeable once the function gap $f(x_k) - f^*$ drops below $10^{-4}$, which occurs around iteration $k \approx 1000$.
> - For $\kappa = 10^5$, the acceleration becomes noticeable once the function gap $f(x_k) - f^*$ drops below $10^{-3}$, which occurs around iteration $k\approx 400$.
> - For $\kappa = 10^3$, the acceleration becomes noticeable once the function gap $f(x_k) - f^*$ drops below $10^{-1}$ which occurs around iteration $k\approx 100$.
>
> These results suggest that RHGD benefits from acceleration when the target precision $\varepsilon$ is smaller than these thresholds. We would also like to point out that this delayed acceleration phenomenon is not unique to our method. Similar behavior can be observed in prior works comparing accelerated methods and gradient descent, such as Figure 1(a) in [1] and Figure 3 in [2], where acceleration does not manifest clearly during the very early optimization phase. We hope this addresses your concern, and we are happy to provide further details if needed! Let us know if you have any other questions or concerns.
>
> [1] Wilson, Mackey and Wibisono, "Accelerating Rescaled Gradient Descent: Fast Optimization of Smooth Functions", NeurIPS 2019.
>
> [2] Kim, Yang, "Unifying Nesterov’s Accelerated Gradient Methods for Convex and Strongly Convex Objective Functions", ICML 2023

---

> > ### Comment · Reviewer_ZMTX · 2025-08-06
> > **Response to the authors**
> >
> > Thank you for your note.
> >
> > This is good. I am glad we are now seeing behaviors in the optimization trajectories that are consistent with known effects. Thinking about all our discussion regarding the experiments, in the final version of your manuscript, I would advise paying attention to numerical precision issues in all the experiments, and including more detailed discussions of what is actually being observed. You may also want to play with cutting off the trajectories after a certain tolerance is reached (as in your most recent run) as opposed to after a certain number of iterations to make the plots clearer. I trust you will think through how to do this. I am going to raise my score and sign off.

---

> > > ### Author Response · Authors · 2025-08-06
> > >
> > > Thank you very much for your thoughtful feedback and for raising your score.
> > >
> > > We truly appreciate your engagement throughout the discussion. In the final version of the manuscript, we will make sure to carefully address the numerical precision issues in all experiments. In particular, we will include a more detailed discussion of what is being observed in the optimization trajectories and clarify the impact of numerical artifacts.
> > >
> > > Following your suggestion, we also plan to adjust the stopping criteria to be based on achieving a target tolerance (e.g., $f(x_k)-f^* < \varepsilon$), rather than a fixed number of iterations, in order to make the plots more informative and easier to interpret.
> > >
> > > Thank you again for your constructive input and support.

---

### Official Review · Reviewer_DNKQ · 2025-07-01

**Clarity:** 4
**Significance:** 4
**Originality:** 4
**Rating:** 5
**Confidence:** 4

**Summary:**

This paper presents new connections between optimization and Hamiltonian dynamics. Specifically the authors study the Hamiltonian version of the gradient flow dynamics. They show that for periodic resetting of the velocity, the exact integration yields convergence rates that match deterministic gradient descent, in both weakly and strongly convex/PL condition cases. Aspiring to improve these rates to match those of accelerated gradient descent (AGD), the authors consider the idea of randomizing the reset/integration time. Typically they propose a non-homogeneous Poisson process to govern the resets. Within each period, they further disco ties the Hamiltonian dynamics to obtain and implementable algorithm (RHGD), which has expected convergence rates matching AGD. I am quadratic opt simulation and logistic regression problems, the performance of RHGD is similar to other accelerated methods and more robust in some circumstances.

**Questions:**

Major:
1. In the paragraph "Approximate Poisson process." an acceptance-rejection scheme is described. However, I can think of alternative approach that first samples a path/trace of the Poisson process for $K$ steps. Indeed, this can be done totally independently of the dynamics I think. This gives a sequence of inter-arrival times. Then, within each subinterval of that sequence, you can discretize the HF dynamics. Will this approach also work? If not, shat is the drawback of this approach compared to yours?
2. You propose an implicit (backward Euler) integrator. Is explicit Euler not stable here? Can you elaborate?
3. You mention "Nevertheless, convergence in expectation can still imply high-probability bounds...via Markov’s inequality." However, traditionally in optimization, Markov is not enough and you need a light-tail assumption and stronger concentration inequalities to get high-probability bounds. I think this is firmly beyond the scope of this paper, but I would still appreciate further comment (or correction) on this brief comment you made.
4. All of your quadratic instances in Section 5.1 have minimizer $x^\ast = 0$. This is a bit non-standard (usually a linear term is added), but perhaps WLOG. Can you elaborate on this choice?

Minor:
1. Is the phrase ``$\alpha$-dominated graident" very common in the literature? I more commonly have seen it referred to as the Polyak-Lojasiewicz (PL) Condition, and I might suggest that nomenclature.
2. Why is the constraint $\alpha \leq 1$ needed? Most common in the literature is to simply assume $0 \leq \alpha \leq L$.
3. Although they are defined in the notation section, are big O and big Theta ever actually used in the paper? I could not find them. (And in fact, I really appreciate how your bounds are explicit!)
4. I think $HF_{\eta}(x, y)$ is first used in Algorithm 1 and needs a proper definition before that.
5. Why does Theorem 1 need $L$-smoothness but Theorem 2 as well as results for the exact traditional gradient flow do not?

**Ethical Concerns:**

["NO or VERY MINOR ethics concerns only"]

**Final Justification:**

The authors' response was primarily to questions that I had and my score/evaluation of the paper was already positive (5). Based on their response and my assessment of the other reviews, I maintain my score.

**Limitations:**

yes

**Paper Formatting Concerns:**

None.

**Quality:**

4

**Strengths And Weaknesses:**

Strengths:
1. In my view this is a high-quality submission. The results are technically sound and seem mature. Although I did not read the appendix in detail, the claims are well supported by extensive proofs. The paper is also written well, well organized, and clearly articulates the main ideas and results.
2. To my knowledge the results of this paper and quite significant and interesting. The most similar previous results are Wang [2024], who use Chebyshev polynomials to set the reset times, and achieve results only on strongly convex quadratic functions. This paper presents a significant leap forward, with a new idea of randomized reset times, and results that hold across a much wider class of problems -- problems where the PL condition holds and weakly convex problems. Furthermore, the results significantly strengthen the connections between sampling and optimization, both of which are core topics in machine learning.
3. To the best of my knowledge, there are no overlapping works in the literature and this work appears to present original new insights, which I think will be valuable.

Weaknesses:
1. My reading of this paper is quite positive and I don't have many strong weaknesses. However I will mention that I would appreciate if the authors could provide more insight/intuition on *why* the idea of randomized reset/integration times following a Poisson process actually leads to acceleration. I understand the "real" details are in the proof, but to strengthen the impact of this work some further intuition in the main body (which I could not find) would be quite helpful.
2. I furthermore have several questions below, which I don't inherently view as weaknesses per se, but I would appreciate to see addressed.

---

> ### Author Rebuttal · Authors · 2025-07-28
>
> We truly appreciate the reviewer’s insightful and encouraging comments. Below we provide our responses to the questions raised.
>
> **1. Alternative scheme for simulating the Poisson process**
>
> Thank you for the interesting suggestion. Indeed, pre-sampling the Poisson process trace and then simulating the dynamics within each interval is a valid and intriguing alternative. While our current method uses an accept-refresh approach that tightly couples refresh times with the dynamics, exploring pre-sampled refresh traces as a potential alternative is an interesting direction, and we would be excited to investigate it in future work.
>
> **2. Is explicit Euler not stable here? Can you elaborate?**
>
> Theoretically, using the explicit Euler scheme to discretize our flow does not yield accelerated convergence rates. This is consistent with prior work:  [1] shows that when the Euler method is applied to discretize high-resolution ODEs for acceleration, the resulting algorithm fails to exhibit acceleration. The key reason is that the stepsize must scale with the strong convexity constant, which limits the convergence rate and prevents acceleration. In contrast, more carefully designed discretization schemes such as symplectic or implicit integrators can better preserve the geometric structure of the flow and support accelerated behavior.
>
> [1] Shi, Bin, et al. "Acceleration via symplectic discretization of high-resolution differential equations." Advances in Neural Information Processing Systems 32 (2019).
>
> **3. Rough high probability bound by Markov’s inequality**
>
> You’re absolutely right. While Markov’s inequality can technically yield a high-probability bound from expectation, such bounds are loose and not meaningful without additional assumptions. We’ve revised the text to clarify that stronger concentration requires light-tail or boundedness assumptions, which are beyond the scope of this work.
>
> **4. $x^*=0$ for the quadratic minimization experiments**
>
> Thank you for pointing this out. We chose $x^*=0$ for simplicity, as one can always shift a quadratic function to have its minimizer at the origin without loss of generality. This normalization simplifies the presentation and does not affect the dynamics or convergence behavior of the algorithms.
>
>
> **5. Intuition on why randomization leads to acceleration**
>
> As we show in Theorem 1, the short-time integration only recovers the same non-accelerated convergence rates as GD, which implies that the short-time integration doesn't efficiently decrease the function value, and we need potentially longer integration time. Another intuition comes from sampling, namely there is a conjecture that Hamiltonian Monte Carlo with randomized integration time can achieve accelerated mixing time for sampling. This has been shown in the idealized continuous-time setting in [Lu and Wang, 2022], and discussed in further details in [Jiang, 2023]. Our work can be seen as an optimization analogue of this question, and we show that randomized integration time does indeed lead to acceleration, both in continuous and discrete times.
>
> [Lu and Wang, “On explicit $L^2$-convergence rate estimate for piecewise deterministic Markov processes in MCMC algorithms’’, Annals of Applied Probability, 2022]
>
> [Jiang, "On the dissipation of ideal Hamiltonian Monte Carlo sampler." Stat 12.1 (2023): e629.]
>
> **6. $\alpha$-gradient domination vs PL inequality**
>
> The condition we referred to as “$\alpha$-dominated gradient” is indeed also known as the Polyak–Łojasiewicz (PL) condition, which is a more commonly used term in the optimization literature. We used the term “gradient dominated” following some prior works, but we agree that “PL condition” is more standard and widely recognized. We are happy to update the terminology in the revision to reflect this, and clarify the equivalence where appropriate.
>
> **7. Why assume $\alpha \leq 1$?**
>
> Thank you for the observation. In our analysis (e.g., line 1445), we assume $\alpha \leq 1$ to simplify the expression involving step size $h$ and to ensure terms incorporating $\alpha$ remain bounded. This assumption is natural in practice, as most real-world problems satisfy $\alpha\leq 1$, and it helps keep constants manageable. That said, the analysis can be extended to the general case $\alpha \leq L$ via rescaling. We will clarify this in the revision.
>
> **8. Where are big O and big Theta used?**
>
> We used big O in the summary of contribution (line 79-line 87) and we used big Theta in related work discussion (line 1047).
>
> **9. The definition of $HF_{\eta}$**
>
> We defined $HF_{\eta}$ in Section 2.1 (Notations and definitions, line 105). We will make this more prominent before Algorithm 1 in the revision to improve clarity.
>
> **10. Why does Theorem 1 need $L$-smoothness but Theorem 2 as well as results for the exact traditional gradient flow do not?**
>
> The key to establishing Theorem 1 is  Lemma 6 in Appendix D, which is a stronger descent lemma and requires $L$-smoothness to bound the magnitude of velocity. Intuitively, HF-opt needs small integration time, which can be viewed as a discretization, and thus we need smoothness. In contrast, neither gradient flow nor randomized Hamiltonian flow needs small integration time and thus we don’t need smoothess in our analysis.

---

> ### Comment · Reviewer_DNKQ · 2025-08-05
> **Response to Rebuttal**
>
> I thank the authors for their detailed responses to my questions, which helped clarify the paper further for me. I appreciate their plans to make the mentioned changes to the camera ready version of the paper. I maintain my evaluation/score of 5.

---

### Official Review · Reviewer_QLTJ · 2025-07-01

**Clarity:** 3
**Significance:** 3
**Originality:** 3
**Rating:** 4
**Confidence:** 2

**Summary:**

This paper studies the Hamiltonian flow for optimization (HF-opt). The authors show that HF-opt has the same convergence rates as gradient descent for minimizing strongly and weakly convex functions. The authors also propose the randomized Hamiltonian flow (RHF) by randomizing the integration time, which achieves accelerated convergence rates in continuous time. This paper also studies a discrete-time implementation of RHF as the randomized Hamiltonian gradient descent (RHGD) algorithm, which achieves the same accelerated convergence rates as Nesterov’s accelerated gradient descent (AGD) for minimizing smooth strongly and weakly convex functions.

**Questions:**

1. Theorems 2 and 3 establish the convergence rate in continuous time without smoothness assumption. For the discrete time case, Theorems 4 and 5 need the L-smooth assumption. What is role of L-smooth when discretizing a continuous system?

2. Why can randomness accelerate Hamiltonian flow? What is the intuition?

3. Can the authors elucidate the physical principles underlying the descrete-time Algorithm 3? For example, what does each of the four steps on lines 3-6 mean? There are many accelerated algorithms developed from the continuous-to-descrete view, providing an intuitive physical interpretation would significantly advance our understanding of acceleration mechanism.

**Ethical Concerns:**

["NO or VERY MINOR ethics concerns only"]

**Final Justification:**

I agree to accept this paper

**Limitations:**

The proposed descrete RHGD algorithm needs two gradient computations at each iteration, which may make RHGD two times slower than the standard AGD.

**Quality:**

3

**Strengths And Weaknesses:**

Strengths:

1. This paper propses the randomized Hamiltonian flow (RHF) for optimization and stablishes its accelerated convergence rates of $O(exp(-\sqrt{a/5}t))$ for $a$-strongly convex functions and $O(1/t^2)$ for weakly convex functions.

2. This paper proposes the randomized Hamiltonian gradient descent (RHGD), a discretization of RHF. Under L-smoothness assumption, The authors establish the $\widetilde O(\sqrt{L/a})$ complexity for a-strongly convex functions and $O(\sqrt{L/\epsilon})$ for weakly convex functions. These complexities matches the optimal accelerated rates of AGD.

3. The RHGD is a new algorithmic formulation of acceleration, expanding the family of accelerated algorithms with a new member. It may advance the understanding of acceleration mechanism from the view of Hamiltonian flow.

4. The theory is solid and the proofs are organized well.

5. The paper is well-written.

Weaknesses:

1. The RHGD algorithm in Algorithm 3 needs two gradient computations at each iteration. In most machine learning applications, gradient computation constitutes the most time-consuming operation. This makes each step of Algorithm 3 two times slower than the standard AGD. Especially, in Figures 1 and 2 which uses the iteration as the X-axis, RHGD performs similar to AGD, making RHGD almost two times slower than AGD when using time as the X-axis.

2. The convergence rate of RHGD holds in expectation. The authors argue that  convergence in expectation can still imply high-probability bounds for via Markov’s inequality. However, Markov’s inequality makes the algorithm $1/\gamma$ (or $log\frac{1}{\gamma}$) times slower when the convergence holds with probability at least $1-\gamma$.

In conclusion, this paper is well-written. The theory is solid. The Hamiltonian flow perspective of acceleration seems interesting. However, it seems slower than AGD and lacks practical utility.

---

> ### Author Rebuttal · Authors · 2025-07-28
>
> We thank the reviewer for the positive and constructive feedback. Our responses to the questions and concerns are provided below.
>
> **1. What is the role of L-smooth when discretizing a continuous system?**
>
> $L$-smoothness plays a central role in our discretization and analysis. Specifically, we assume that the objective function is $L$-smooth to ensure that the discretized dynamics accurately approximate the behavior of the continuous system. In the context of optimization, $L$-smoothness is a standard assumption when transitioning from continuous-time dynamics to discrete-time algorithms. For instance: discretizing gradient flow to gradient descent requires smoothness to ensure stability and guarantee convergence under appropriate step sizes (see Theorems 6 and 8 in Appendix C.1.1). Similarly, discretizing accelerated gradient flow into accelerated gradient descent also relies on smoothness to control discretization error and preserve acceleration (see Theorems 7 and 9 in Appendix C.1.1). In our work, L-smoothness enables us to bound the approximation error (see Proposition 1 in Appendix G.2), which is the key to the convergence of Algorithm 3 (RHGD).
>
> **2. Why can randomness accelerate Hamiltonian flow? What is intuition?**
>
> As we show in Theorem 1, the short-time integration only recovers the same non-accelerated convergence rates as GD, which implies that the short-time integration doesn't efficiently decrease the function value, and we need potentially longer integration time. Another intuition comes from sampling, namely there is a conjecture that Hamiltonian Monte Carlo with randomized integration time can achieve accelerated mixing time for sampling. This has been shown in the idealized continuous-time setting in [Lu and Wang (check)], and discussed in further details in [Jiang, 2023]. Our work can be seen as an optimization analogue of this question: we show that randomized integration time does indeed lead to acceleration for acceleration, both in continuous and discrete times.
>
> [Lu and Wang, “On explicit $L^2$-convergence rate estimate for piecewise deterministic Markov processes in MCMC algorithms’’, Annals of Applied Probability, 2022]
>
> [Jiang, "On the dissipation of ideal Hamiltonian Monte Carlo sampler." Stat 12.1 (2023): e629.]
>
> **3. Can the authors elucidate the physical principles underlying the discrete-time Algorithm 3?**
>
> We thank the reviewer for this excellent question. Since our method is derived from discretizing a randomized Hamiltonian flow, the algorithm naturally admits a physical interpretation inspired by classical mechanics, where $x_k$ represents position and $y_k$ corresponds to velocity. Below we explain each step (lines 3–6) from this perspective:
>
> - Line 3 performs a half-step position update using the current velocity, consistent with the Leapfrog integrator in Hamiltonian dynamics. It reflects how a particle’s position changes under its inertial motion, prior to any forces being applied.
>
> - Line 4 is a force-induced correction to the position, using the gradient of the potential energy (objective function). Physically, it corresponds to the second half of the position update, now taking into account the effect of the force field. The form resembles a position Leapfrog step and is critical for stability and accuracy.
>
> - Line 5 updates the velocity according to Newton’s second law $\dot{Y}_t=-\nabla f(X_t)$, which reflects how the gradient affects the velocity, pushing the system toward low-energy regions.
>
> - Line 6 corresponds to a random velocity reset. With a small probability proportional to the refresh rate $\gamma_k$, the velocity is zeroed out to inject dissipation and prevent the system from accumulating excessive momentum or oscillations. This mechanism plays a key role in balancing inertia (acceleration) and stability, and is crucial for escaping flat regions or poorly conditioned directions.
>
> We will incorporate this interpretation into the final version to clarify the physical intuition behind RHGD.
>
> **4. Two gradient evaluations of Algorithm 3**
>
> We appreciate the reviewer’s observation regarding the number of gradient evaluations. It is true that Algorithm 3 (RHGD) requires two gradient evaluations per iteration, which arises naturally from the discretization of the underlying Hamiltonian flow. In fact, this two-gradient structure appears to be inherent in many established discretizations of Hamiltonian dynamics, such as the Leapfrog integrator widely used in Hamiltonian Monte Carlo and related methods. From our perspective, this structure is not merely a cost, but a design choice that contributes to the robustness and stability of RHGD. In our experiments, we observe that RHGD is more robust to parameter misspecification compared to other first-order methods, potentially due to the richer feedback provided by evaluating the gradient at multiple points. We recognize that gradient cost is a valid concern in practice. An interesting direction for future work is to explore gradient-efficient variants of RHGD while retaining its acceleration and stability properties.
>
> **5. Worse high probability bound via Markov’s inequality**
>
> We appreciate the reviewer’s observation. We agree that Markov’s inequality yields loose bounds, and our intention was not to claim tightness, but to demonstrate that the convergence in expectation can be converted into a high-probability guarantee without additional assumptions, which is consistent with some prior works, e.g., [1] and [2]. Given the randomized velocity refresh in our algorithm, stronger concentration results would require assumptions such as bounded variance or martingale structure. To keep the analysis general and assumption-free, we opted for the Markov-based bound. We agree that deriving tighter bounds under additional structure is an interesting direction for future work and will clarify this in the revision.
>
> [1] Mathieu Even, et al. "Continuized accelerations of deterministic and stochastic gradient descents, and of gossip algorithms." NeurIPS 2021.
>
> [2] Jun-Kun Wang, and Andre Wibisono. "Continuized Acceleration for Quasar Convex Functions in Non-Convex Optimization." ICLR 2023.

---

> > ### Comment · Reviewer_QLTJ · 2025-08-04
> >
> > Thank you for your response. I have read through the rebuttal and have decided to keep my rating.

---

> > > ### Comment · Area_Chair_R5ce · 2025-08-04
> > >
> > > Dear Reviewer QLTJ,
> > >
> > > Could you please clarify whether the rebuttal satisfactorily addresses your concerns? If not, could you specify what you still find inadequate? This will help us evaluate the paper and assist the authors in improving their work.
> > >
> > > Thanks,
> > > AC

---

> > > > ### Comment · Reviewer_QLTJ · 2025-08-04
> > > >
> > > > Dear AC,
> > > >
> > > >     The authors acknowledge in their response that their algorithm requires two gradient evaluations per iteration while the classical AGD only requires one. So I decide not to increase my score. However, I am positive about this paper overall.

---

> > > > > ### Comment · Area_Chair_R5ce · 2025-08-04
> > > > >
> > > > > Thank you for the clarification.
> > > > >
> > > > > AC

---

### Official Review · Reviewer_9m2g · 2025-07-02

**Clarity:** 3
**Significance:** 3
**Originality:** 4
**Rating:** 5
**Confidence:** 3

**Summary:**

The paper proposes to add a downward jump term to the velocity to the Hamiltonian flow such that at random Poisson times (which can be non-homogeneous), the velocity is reset to zero. By randomizing the integration time, the paper shows that the resulting randomized Hamiltonian flows achieves accelerated convergence rates in continuous time, similar to those for accelerated gradient flow. The paper also studies the discrete-time algorithm based on randomized Hamiltonian flows, and obtained theoretical guarantees for expected suboptimality,  which enjoy the same convergence rates as Nesterov's accelerated gradient descent in both strongly convex and weakly convex settings.  Numerical results are also provided to show the efficiency of the proposed algorithm.

**Questions:**

(1) In your equation (RHF), I think you should write $Y_{t-}dN_{t}$ instead to be more rigorous.

(2) The Poisson intensity $\gamma(t)$ is constant for strongly convex setting and decreasing for weakly convex setting. That reminds me of choosing constant stepsize for strongly convex setting and decreasing stepsize for Nesterov's accelerated gradient descent. I am wondering if there are any connections here.

(3) It would be helpful if the author(s) can add some discussions (if not already) the intuitions why the proposed method can accelerate. Also, $\gamma(t)$ is a function of time in general, and that is indeed the choice in the weakly convex setting. But $\gamma(t)$ is independent of the Poisson arrivals in the model. I am wondering what is the intuition to choose $\gamma(t)$ this way, instead of for example choosing $\gamma(t)=\phi(t-\tau_{i})$ between two Poisson arrivals $\tau_{i}$ and $\tau_{i+1}$, i.e. reset the Poisson intensity after the velocity hits zero.

(4) It seems that you missed quite a few references when you discuss underdamped Langevin algorithms, for example, Chau and Rasonyi (2022) Stochastic Gradient Hamiltonian Monte Carlo for Non-convex Learning, Dalalyan and Riou-Durand (2020) On sampling from a log-concave density using kinetic Langevin diffusions, Gao et al. (2022) Global Convergence of Stochastic Gradient Hamiltonian Monte Carlo for Nonconvex Stochastic Optimization.

(5) In the reference Andre Wibisono, Ashia C Wilson, and Michael I Jordan, proceedings should be Proceedings.

**Ethical Concerns:**

["NO or VERY MINOR ethics concerns only"]

**Final Justification:**

I am satisfied with author(s)' response. I am keeping the score.

**Limitations:**

Yes.

**Paper Formatting Concerns:**

No.

**Quality:**

4

**Strengths And Weaknesses:**

Strengths:

(1) The continuous-time and discrete-time methods proposed in this paper are novel to the best of my knowledge. Interestingly, the proposed method can achieve the same accelerated convergence rates as Nesterov's accelerated method.

(2) The paper is very well written, and the analysis seems to be rigorous.

(3) Numerical results are promising.

Weaknesses:

(1) It would be great if the author(s) can provide some guidance when it is better to use Nesterov's accelerated method and when it is better to use your proposed method, especially because it works under expectations only.

(2) In practice, stochastic gradient is often used, and it would be interesting to investigate the sensitivity to gradient noise for the proposed method. But I understand that this is beyond the scope of the paper, and should be left as a future research direction.

---

> ### Author Rebuttal · Authors · 2025-07-28
>
> Thank you for the positive comments and feedback. Our responses to the questions and concerns are provided below.
>
> **1. It would be great if the author(s) can provide some guidance when it is better to use Nesterov's accelerated method and when it is better to use your proposed method**
>
> We appreciate the reviewer’s helpful suggestion. While we do not yet have a complete theoretical characterization, our experiments suggest that the proposed method is more robust to parameter misspecification. In particular, when the strong convexity constant is small, unknown and must be estimated heuristically, our method tends to perform better than Nesterov’s accelerated method, which can be sensitive to such inaccuracies.
>
> **2. Stochastic gradient variants**
>
> This is a very interesting direction, especially since the stochastic gradient variants are more applicable and scalable in large-scale settings. We would be happy to explore this in future work.
>
> **3. In your equation (RHF), I think you should write $Y_{t^-}dN_t$**
>
> Thank you for pointing out. We followed the same notation about the Poisson process as [Even et al., 2021] in the submission. We will change it as the reviewer suggested to be more rigorous.
>
> [Even, et al. "Continuized accelerations of deterministic and stochastic gradient descents, and of gossip algorithms." NeurIPS 2021.]
>
> **4. Connection between Nesterov's AGF (AGD) and RHF (RHGD)**
>
> This is a very good catch. The choice of $\gamma$ is similar to the choice of damping parameter. However, the constants are not totally the same. We are interested in exploring the connection between RHF (RHGD) and AGF (AGD) in the future.
>
> **5. Intuition on why randomization can lead to acceleration**
>
> By energy conservation, if we initialize the velocity to be zero, we can decrease the function value as we evolve the Hamiltonian flow (HF). Then we study how long we should evolve HF. As we show in Theorem 1, the short-time integration only recovers the same non-accelerated convergence rates as GD, which implies that the short-time integration doesn't efficiently decrease the function value, and we need potentially longer integration time. Another intuition comes from sampling, namely there is a conjecture that Hamiltonian Monte Carlo with randomized integration time can achieve accelerated mixing time for sampling. This has been shown in the idealized continuous-time setting in [Lu and Wang, 2022], and discussed in further details in [Jiang, 2023]. Our work can be seen as an optimization analogue of this question, and we show that randomized integration time does indeed lead to acceleration, both in continuous and discrete times.
>
>  [Lu and Wang, “On explicit $L^2$-convergence rate estimate for piecewise deterministic Markov processes in MCMC algorithms’’, Annals of Applied Probability, 2022.]
>
>  [Jiang, "On the dissipation of ideal Hamiltonian Monte Carlo sampler." Stat 12.1 (2023): e629.]
>
> **6. Intuition on the choice of $\gamma(t)$**
>
> In our setup, $\gamma(t)$ serves as the intensity of the Poisson process, and we chose it in this specific form primarily to make the convergence proof work. While this choice may seem somewhat technical, its resemblance to the time-varying damping coefficient in Nesterov's accelerated gradient flow [Su et al, 2016] lends further plausibility. Exploring alternative designs is indeed interesting, and we leave this for future investigation.
>
> [Su, Boyd and Candes, "A differential equation for modeling Nesterov's accelerated gradient method: Theory and insights." Journal of Machine Learning Research 17.153 (2016): 1-43.]
>
> **7. Reference missing and wrong capitalization**
>
> Thank you for pointing out these issues, we will correct them in the revision.

---

> > ### Comment · Reviewer_9m2g · 2025-08-04
> > **response**
> >
> > Thanks for the detailed response.

---

### Official Review · Reviewer_KAWd · 2025-07-02

**Clarity:** 4
**Significance:** 3
**Originality:** 3
**Rating:** 5
**Confidence:** 2

**Summary:**

This paper proposes an optimization method inspired by Hamiltonian Monte Carlo (HMC), which follows the system’s equations of motion. At random intervals, the momentum is reset to zero, causing the state to converge to a (local) minimum rather than oscillating around it. The approach aligns with recent Hamiltonian-based optimization techniques, such as [Wang 2024]. To the best of my understanding, the primary difference is that in [Wang 2024], momentum truncation occurs after fixed evolution times, whereas in this work, those truncation times are sampled at random.

The authors provide theoretical convergence rate bounds which, as far as I can tell, are comparable to those established in prior work on related algorithms. Given these similarities, I find it difficult to clearly assess the novelty of the proposed method relative to the existing literature.

**Questions:**

1. As you mentioned, most existing Hamiltonian-based optimization methods use a damping term for dissipation energy. You highlight that your works (along with [Wang 24]) is not so. My question is that why should that be an important feature? Since, as far as I can say, setting the momentum to zero (either at random or fixed intervals) in effect plays the role of friction and damping, causing the state to converge to a (local) minimum rather than oscillating around it. Can you comment on this?

**Ethical Concerns:**

["NO or VERY MINOR ethics concerns only"]

**Final Justification:**

In the rebuttal discussions, I realised that the authors have presented an interesting theoretical bound between their approach and HMC via a scheme that they call LCP.

Remaining insufficiencies:
The authors should spell out the comparisons with [Wang 2024].

**Limitations:**

One limitation is that the established bounds hold in expectation due tot he randomness of the algorithm which is specified in the paper.

**Paper Formatting Concerns:**

No concern

**Quality:**

3

**Strengths And Weaknesses:**

Strengths:
1. The paper is well-written. The notations are clear, and the formal derivations are rigorous.
2. In practice, (on the toy examples that they provide), their algorithm is comparable to continuized AGD (CAGD).
3. Theoretical bounds on the convergence rate are proved.
4. [After rebuttal] An Interesting and deep relation between the proposed optimizer and HMC is established.

Weaknesses:
1. The significance of the contributions, compared with similar methods such as "Frictionless Hamiltonian Descent and Coordinate Hamiltonian Descent for strongly convex quadratic problems and beyond" (https://arxiv.org/pdf/2402.13988) is either not clarified or the contributions are minimal.
2. No comparison is made with any existing Hamiltonian-descent-based optimization algorithms, such as [Wang 2024].
3. The comparison is only carried out on toy models.

---

> ### Author Rebuttal · Authors · 2025-07-28
>
> Thank you for the comments and feedback. Please kindly find our response to the questions and concerns below.
>
> **1. The significance of the contributions, compared with similar methods such as [wang 2024] is either not clarified or the contributions are minimal.**
>
> As we claim in line 69-70 and line 141-142, [wang 2024] only establishes the accelerated convergence of HF-opt via Chebyshev-based integration time for **quadratic functions** whereas our paper establishes accelerated convergence rates via randomized integration time for **general strongly and weakly convex functions**. In general, the algorithm in [Wang 2024] is not implementable due to the exact simulation of Hamiltonian flow, whereas we propose an implementable algorithm in discrete time that preserves the accelerated convergence rates. We believe that this is a significant contribution.
>
> **2. No comparison is made with any existing Hamiltonian-descent-based optimization algorithms, such as [Wang 2024].**
>
> We will clarify the comparisons with [Wang 2024]. We would like to note that [Wang 2024] focuses on quadratic functions, while our work treats general strongly/weakly convex functions. We would also like to highlight that in the case of the quadratic objective function, our result shows RHF-opt achieves a sharper accelerated convergence rate without having to assume smoothness (finite L), while the result of [Wang 2024] requires smoothness (finite L), see Appendix E.
>
> **3. The comparison is only carried out on toy models.**
>
> We would like to clarify that our experiments include two widely studied and representative convex optimization problems: quadratic minimization and logistic regression. These are standard benchmarks in the literature and are commonly used to evaluate the behavior of optimization algorithms, especially in the context of understanding convergence and acceleration. We agree that evaluating the method on larger-scale or nonconvex problems would be an interesting direction for future work.
>
> **4. Why should setting the velocity to zero be an important feature?**
>
> Our work is inspired by the Hamiltonian Monte Carlo (HMC) algorithm for sampling, where refreshing the velocity is the key to decreasing the KL divergence (see Appendix B.2). The HF-opt we propose is a direct translation of this principle (that we call the Lift-Conserve-Flow/LCP principle, see Appendix B) to optimization. Both our work and [Wang 2024] strengthen the link between HMC and optimization. Our work also study the acceleration in optimization from a new perspective and proposes novel accelerated dynamics and algorithms as candidates of accelerated optimization methods.
>
> **5. For novelty**
>
> - Our work is the first to study the Hamiltonian flow for minimizing general strongly convex, gradient dominated and weakly convex functions.
> - Our work is the first to establish the acceleration for strongly and weakly convex functions based on Hamiltonian flow with randomized integration time in both continuous-time and discrete-time.

---

> > ### Comment · Reviewer_KAWd · 2025-08-06
> > **Response**
> >
> > Thanks for your response, and sorry for replying late. You clarified some of my concerns -- notably, how it differs from [Wang 24].
> > Other concerns are not addressed or are addressed only partially:
> > 1. No comparison with any existing Hamiltonian-descent-based optimization algorithms is provided. (though promised in the final version).
> > 2. Evaluating the method on larger-scale or nonconvex problems is not given but postponed to future work.
> > 3. You have not provided any reasoning why setting the momentum to 0 should play a role that is different from assuming a friction term. It seems to me that they should have the same effect. You mention that your work "is inspired by the Hamiltonian Monte Carlo (HMC) algorithm for sampling, where refreshing the velocity is the key to decreasing the KL divergence", but the inspiration is totally superficial and misleading. In HMC, in each iteration, the momentum is drawn from a marginal normal distribution (not truncated) to guarantee the detailed balance condition that is required for MCMC sampling. This is irrelevant to your optimization setting.  If you can demonstrate a deeper connection between your Hamiltonian-based approach for optimization and HMC, I will gladly increase my score -- though I would be surprised if such a deeper link exists.

---

> > > ### Author Response · Authors · 2025-08-06
> > >
> > > We thank the reviewer for engaging in this discussion with us.
> > >
> > > We would respectfully disagree with the reviewer's claim (regarding the connection between HF-opt for optimization and HMC for sampling) that "the inspiration is totally superficial and misleading". We believe there is a precise connection between them, via what we call the Lift-Conserve-Project (LCP) scheme, as we explain in Appendix B (and also mention in line 124 in the main text).
> > >
> > > To briefly summarize: The LCP scheme works by (1) lifting the problem to a phase space that preserves the current objective value; (2) evolving under a conservative flow; and (3) projecting back to the original space which reduces the objective function.
> > >
> > > Our proposed HF-opt is an instantiation of this principle, where
> > >
> > > - (1) we lift the objective function $f(x)$ to a Hamiltonian function $H(x,y)=f(x)+\frac{1}{2}||y||^2$, starting with zero initial velocity ($y=0$) so the function value is preserved ($H(x,0) = f(x)$);
> > >
> > > - (2) evolving under the Hamiltonian flow which conserves the Hamiltonian function;
> > >
> > > - (3) projecting back to the $x$ space which reduces the objective ($f(x) \leq H(x,y)$). When iterated, this gives the HF-opt algorithm which "evolves following Hamiltonian flow and resetting velocity to $0$ every once in a while".
> > >
> > > In the precise sense, HMC is also an instance of this principle. Here the goal is to sample from a distribution $\nu(x)$, which can be equivalently formulated as minimizing the KL divergence $F(\rho) = KL(\rho || \nu)$ over the space of distributions $\rho(x)$. The HMC algorithm that "evolves following Hamiltonian flow and resetting velocity to a fresh Gaussian every once in a while", can be seen as implementing the LCP scheme as follows:
> > >
> > > - (1) Lifting step: We lift the base KL divergence to be a joint KL divergence $\tilde{F}(\tilde{\rho}) = KL(\tilde{\rho} || \tilde{\nu})$ with respect to the joint distribution $\tilde{\nu}(x,y) = \nu(x) \times \text{Gaussian}(y)$, and lifting the current distribution $\rho(x)$ to $\tilde{\rho}(x,y) = \rho(x) \times \text{Gaussian}(y)$ (this is "refreshing velocity from a fresh Gaussian"), so that $KL(\tilde{\rho} || \tilde{\nu}) = KL(\rho || \nu)$.
> > >
> > > - (2) Conserving step: We evolve following the Hamiltonian flow in the $(x,y)$ space; when translated to the space of distributions, this flow conserves the KL divergence $KL(\tilde{\rho} || \tilde{\nu})$.
> > >
> > > - (3) Projection step: We drop the $y$ component and project back to the $x$ space; when translated to the space of distributions, we project from the joint distribution $\tilde{\rho}(x,y)$ to the $x$-marginal $\rho(x)$, which has the effect of reducing the KL divergence: $KL(\rho || \nu) \le KL(\tilde{\rho} || \tilde{\nu})$.
> > >
> > > The LCP scheme provides a clear view of why HMC works for sampling, by minimizing KL divergence. We would like to note that LCP is not the only view of why HMC works, as the common wisdom in the MCMC community proceeds via detailed balance consideration, as the reviewer astutely noted. However, we believe the LCP scheme provides a valuable alternative view that may be useful for other algorithmic purposes, and one of our main goals in this paper is to derive a pure optimization algorithm, the HF-opt, based on this LCP scheme.
> > >
> > > We would also like to point out that in the MCMC literature, there is indeed an intuitive connection between continual friction as in the  kinetic (underdamped) Langevin dynamics, and periodic velocity refreshment as in HMC; however, at the precise technical level, the analyses of mixing times for kinetic Langevin and HMC require quite distinct tools. Similarly, for optimization, the continual friction gives the classical heavy-ball dynamics and Nesterov acceleration, which are well-studied; however, the analogous algorithm derived from periodic velocity resetting (made precise via the LCP scheme) was still missing, which we aim to fill in via this work with HF-opt.

---

> > > > ### Comment · Reviewer_KAWd · 2025-08-07
> > > > **On linking with HMC**
> > > >
> > > > With all respect, what you call "Lift-Conserve-Project" does not make any sense to me. You are just borrowing concepts from the HMC literature that cannot be linked to the optimization task.
> > > > In HMC, the sampling is done in the augmented space of position (original space) and momentum simply because in this space evolution of the current state using the equations of motion conserves the Hamiltonian; as such, the Metropolis-Hastings acceptance probability will be close to 1.
> > > > In your setting, preserving the Hamiltonian does not have an equivalent advantage. And you are truncating the momentum regularly anyway!
> > > > What you are doing is following the gradient (as optimizers typically do), plus an initial random momentum/velocity that acts as an additive noise. By approaching the local optimum points (low-energy points), the momentum increases (which, arguably, is a bad property and makes the state escape from the optimal point and oscillate around it). To avoid this problem, you are truncating the momentum regularly, which has the same effect as friction. In the optimization setting, you are finding a local optimum point (in the original parameter space). You are not minimizing any KL divergence. And even if (somehow) you would minimize the KL divergence in the augmented space (or as you call it the "lifted space")  by truncating the momentum, you are regularly changing the Hamiltonian! This is very different from HMC, where the momentum is sampled from its marginal.

---

> > > > > ### Author Response · Authors · 2025-08-07
> > > > >
> > > > > We thank the reviewer for the response. We respect the reviewer's intuition regarding HMC, and we do not wish to enforce our understanding to the reviewer. However, we would like to briefly respond to clarify what we believe is a misconception regarding our explanation, just in case this discussion is also useful for other reviewers or readers.
> > > > >
> > > > > - **"You are just borrowing concepts from the HMC literature that cannot be linked to the optimization task."**
> > > > >
> > > > > Our explanation of HMC uses the perspective of "sampling as optimization in the space of probability distributions"; in particular, sampling from a target distribution $\nu(x)$ (a task that is by nature stochastic in the state space $\mathbb{R}^d$) is equivalent to the optimization task of minimizing the KL divergence $F(\rho) = KL(\rho || \nu)$ on the space of probability distributions $\mathcal{P}(\mathbb{R}^d)$. Perhaps the most well-known of this perspective is the correspondence that the Langevin dynamics (a stochastic process) for sampling, is in fact running the gradient flow of KL divergence (a deterministic optimization dynamics) on the space of distributions. This perspective has been fruitfully used to translate many optimization algorithms and analyses to derive new sampling algorithms.
> > > > >
> > > > > Here, we view HMC as also minimizing KL divergence in the space of distributions, not by running gradient flow, but by performing the LCP scheme, as we explain above. Thus, **HMC is---in a precise sense, not merely at an intuitive sense---an optimization algorithm on the space of distributions.** In our work, we extract this optimization principle (the LCP scheme) more explicitly, and study the arguably simpler setting of pure optimization on $\mathbb{R}^d$, which results in our HF-opt method.
> > > > >
> > > > > - **"In your setting, preserving the Hamiltonian does not have an equivalent advantage."**
> > > > >
> > > > > Preserving the Hamiltonian by itself indeed does not help for optimization; however, it is only one component (the C = Conservation part) of the LCP scheme, and it is the P = Projection part that is helping for optimization, by reducing the objective function.
> > > > >
> > > > > - **"You are not minimizing any KL divergence."**
> > > > >
> > > > > In our optimization problem, we are indeed not minimizing KL divergence. We merely explained how HMC (a sampling algorithm) is actually minimizing KL divergence (via the LCP scheme); our optimization algorithm translates this principle (the LCP scheme) in a rigorous way to derive a new optimization algorithm with provable convergence rates in the convex and strongly convex setting.
> > > > >
> > > > > We sincerely thank the reviewer again for the discussion.

---

> > > > > > ### Comment · Reviewer_KAWd · 2025-08-07
> > > > > >
> > > > > > Let's be more precise:
> > > > > >
> > > > > > In HMC the potential energy function, $f(x)$, is the negative log of a target density that should be approximated by MCMC. So the task of HMC is to approximate the target density $\pi(x) = exp(-f(x))$ with a set of particles, in the sense that the number of particles from each region $A$ be proportional the target probability mass of that region $Pr(A) = \int_A exp(-f(x)) dx$.
> > > > > > In your setting, $f(x)$ is an arbitrary function that you want to minimize; that is, your task is to find $x^* = argmin f(x)$. These two tasks are totally different.
> > > > > > In your setting, you do not have a **target distribution**, as you are not approximating a distribution. You are trying to find a point $x^*$.
> > > > > >
> > > > > > In your previous response, you talked about:
> > > > > >
> > > > > > "lifting the **current distribution** $\rho(x)$ to $\tilde{\rho}(x,y) = \rho(x) \times $ Gaussian$ (y)$ (this is "refreshing velocity from a fresh Gaussian)"
> > > > > >
> > > > > > But in your setting, you do not have a **current distribution** either. You just have a state $(x,y)$ that evolves and its $x$ component tries to find the optimum point $x^*$.
> > > > > > You are not "refreshing velocity from a fresh Gaussian" either. You just truncate $y$ at random points. Unlike HMC (which draws momentum from its marginal Gaussian), your algorithm has nothing to do with a "fresh Gaussian" (as you put it).
> > > > > >
> > > > > >  Reframing HMC (or any MCMC algorithm) as an optimization task is just an over-generalization of the term "optimization" that I am not sure would be helpful for our discussion: HMC approximates a target distribution that does not exist in your setting and does not match your optimization task.
> > > > > >
> > > > > > It is good that we agree that "Preserving the Hamiltonian by itself indeed does not help for optimization".
> > > > > > As mentioned above, what you call the "Projection part", is also totally different between HMC and your algorithm.
> > > > > > In your setting, a "momentum" vector is assigned to the state (similar to any Momentum-based Gradient Optimizer) and evolves by following the Hamiltonian equations of motion.
> > > > > > As such, in your algorithm, the momentum keeps increasing near the optimal point $x^*$ rather than being decreased, which is an undesired property. To mitigate this issue, you simply truncate the momentum at random times.
> > > > > >
> > > > > > There are other incorrect statements in your response, too. For example, you mention the Hamiltonian flow "conserves the KL divergence", which does not make any sense. Maybe you meant it conserves the Hamiltonian (?).

---

> > > > > > > ### Author Response · Authors · 2025-08-07
> > > > > > >
> > > > > > > We thank the reviewer for the response. Respectfully:
> > > > > > >
> > > > > > > (1) In sampling, the task is to sample from a target distribution with density function $\nu(x) = \exp(-f(x))$ (or can be proportional to it). We agree that here, $f(x) = -\log \nu(x)$ is the negative log-density.
> > > > > > >
> > > > > > > In our setting of optimization, our objective function $f(x)$ is arbitrary, and not related to the target distribution above. Indeed, there is no target distribution in our optimization problem.
> > > > > > >
> > > > > > > The reviewer's dismissal of our HMC-based motivation overlooks the conceptual clarity it provides. We do not claim a direct equivalence between HMC and HF-opt, but rather present a principled inspiration for HF-opt from structure of Hamiltonian systems. **We are not claiming that HF-opt is the same algorithm as HMC, but only that it is an instantiation of the same principle (LCP) applied to optimization.**
> > > > > > >
> > > > > > > When we wrote: "lifting the current distribution ...", we were trying to explain how HMC works from an LCP perspective. This is for sampling, not for optimization. So indeed in HF-opt, we do not have a current distribution, but only a current point.
> > > > > > >
> > > > > > > (2) Regarding our discussion of KL divergence in sampling: we would like to clarify that there are multiple well-accepted formulations of the sampling objective. One formulation seeks to output a single sample $X$ with distribution $\rho$ such that $KL(\rho \Vert \nu) \leq \epsilon$.  Another -- as the reviewer mentions -- involves generating multiple samples to approximate $\nu$ via the empirical distribution. Both are valid goals. We focused on the first to make our use of KL divergence and the analogy to HMC more precise.
> > > > > > >
> > > > > > > (3) **"There are other incorrect statements in your response, too. For example, you mention the Hamiltonian flow "conserves the KL divergence", which does not make any sense. Maybe you meant it conserves the Hamiltonian (?)."**
> > > > > > >
> > > > > > > We would like to assure the reviewer that our statement is totally correct. To be more precise, what we meant was the following property, which we stated as Lemma 4 in Appendix B.2 in our submission: If $(X_0,Y_0) \sim \tilde{\rho}_0$ is an arbitrary joint random variable, and if $(X_t,Y_t)$ evolves following the Hamiltonian flow (a deterministic dynamic) starting from $(X_0,Y_0)$, then $(X_t,Y_t) \sim \tilde{\rho}_t$ is also a random variable, and the KL divergence to the joint target distribution $\tilde{\nu}(x,y) = \nu(x) \times \text{Gaussian}(y)$ is conserved, i.e., $KL(\tilde{\rho}_t || \tilde{\nu}) = KL(\tilde{\rho}_0 || \tilde{\nu})$.
> > > > > > >
> > > > > > > For the proof, please kindly see our derivation in the proof of Lemma 4 (page 27 in our submission), which proceeds via continuity equation. (Another derivation comes from the decomposition of KL divergence as the sum of negative entropy plus the expected value of the Hamiltonian function; combined with the properties that
> > > > > > > - the Hamiltonian flow preserves the Hamiltonian function
> > > > > > > - the Hamiltonian flow preserves the entropy of the joint distribution (since the Hamiltonian vector field is divergence-free), this shows the KL divergence is also conserved.)
> > > > > > >
> > > > > > > We hope this clarifies our earlier points. We stand by the mathematical correctness of our statements and believe the broader unifying perspective provided by the LCP framework contributes meaningful conceptual clarity. We appreciate the continued discussion and the opportunity to address the reviewer’s concerns.

---

> > > > > > > ### Comment · Area_Chair_R5ce · 2025-08-08
> > > > > > >
> > > > > > > Dear Reviewer KAWd and Authors,
> > > > > > >
> > > > > > > It appears that a primary concern raised by Reviewer KAWd is how strong the correspondence between Hamiltonian Monte Carlo and Hamiltonian descent is.
> > > > > > >
> > > > > > > @Reviewer KAWd: Besides this issue, do you think this paper has any other flaws from an optimization-theoretic perspective?
> > > > > > >
> > > > > > > @Authors: If the proposed Hamiltonian descent algorithms are only inspired by Hamiltonian Monte Carlo, perhaps you should consider toning down the introduction a little bit. What do you think?
> > > > > > >
> > > > > > > Best,
> > > > > > > AC

---

> > > > > > > > ### Author Response · Authors · 2025-08-08
> > > > > > > >
> > > > > > > > Dear AC,
> > > > > > > >
> > > > > > > > Thank you very much for your thoughtful suggestion.
> > > > > > > >
> > > > > > > > We would like to clarify that while our algorithm is not a direct adaptation of HMC, the connection is more than superficial inspiration. Our design follows a shared Lift-Conserve-Project (LCP) framework with HMC, and we provide a rigorous formulation of this perspective in Appendix B. The LCP view highlights a structural similarity between Hamiltonian dynamics used in sampling and our optimization method, which we believe is both novel and meaningful.
> > > > > > > >
> > > > > > > > That said, we understand the importance of clear framing. In the final version, we are happy to revise the introduction to better reflect the conceptual distinction between HMC and HF-opt, and to avoid any potential misinterpretation that we are claiming a formal equivalence.
> > > > > > > >
> > > > > > > > Best,
> > > > > > > > The authors

---

> > > > > > > > > ### Comment · Area_Chair_R5ce · 2025-08-08
> > > > > > > > >
> > > > > > > > > Dear Authors,
> > > > > > > > >
> > > > > > > > > Thank you for the quick response. Your proposal looks reasonable.
> > > > > > > > >
> > > > > > > > > Best,
> > > > > > > > > AC

---

> > > > > > > > ### Comment · Reviewer_KAWd · 2025-08-08
> > > > > > > >
> > > > > > > > Dear AC and Authors,
> > > > > > > >
> > > > > > > > Thanks to this conversation, I carefully read Appendix B — which, regrettably, I had previously skipped. This resolved many of my misunderstandings. Most notably, I now realize that, in the context of HMC, the LCP scheme operates on the space of probability distributions, rather than on the space on which the target distribution is defined. The authors have indeed established a strong and insightful connection between their optimization approach and HMC in Appendix B. Therefore, I am increasing my score to 5.

---

> > > > > > > > > ### Author Response · Authors · 2025-08-08
> > > > > > > > >
> > > > > > > > > We sincerely thank the reviewer for the thoughtful discussion throughout the rebuttal process, which has helped us clarify and better communicate our ideas. We also appreciate the AC for stepping in and providing helpful guidance. We will incorporate your suggestions into the revised version.

---

> > > ### Author Response · Authors · 2025-08-07
> > >
> > > Dear Reviewer KAWd,
> > >
> > > We followed up in our previous response to clarify the connection between HF-opt and HMC via the Lift-Conserve-Project (LCP) framework, and we hope it addressed your concerns.
> > >
> > > Since the review period ends tomorrow, we would greatly appreciate it if you could kindly let us know whether further clarification would be helpful. We would be happy to provide more details.
> > >
> > > Given the importance of your review and your thoughtful comments, we hope our clarifications may lead you to consider updating your score if you find the response satisfactory.
> > >
> > > Thank you again for your time and engagement.
> > >
> > > The authors

---

> ### Author Response · Authors · 2025-08-05
> **Gentle Reminder Regarding Our Response to Your Review**
>
> Thank you again for taking the time to review our paper. We’ve provided a detailed response to your comments, especially clarifying the key differences between our work and [Wang 2024], as well as addressing your concerns about novelty. We believe some misunderstandings may have affected the initial assessment, and we would greatly appreciate it if you could take another look and let us know your thoughts.
>
> Please let us know if any clarification is needed. Thank you for your time!

---

> ### Comment · Area_Chair_R5ce · 2025-08-06
>
> Dear Reviewer KAWd,
>
> Please read the rebuttal and let us know whether your concerns have been satisfactorily addressed. If not, please point out what remains inadequate. In particular, does your concern about the novelty still remain?
>
> Please respond as soon as possible, as the (extended) deadline for the Author-Reviewer discussion is approaching. Otherwise, I will need to apply the “insufficient review” flag in accordance with the NeurIPS 2025 policy.
>
> > You can flag non-participating reviewers with the InsufficientReview button, and we will use the flag in accordance with possible penalties of this year's responsible reviewing initiative and future reviewing invitations.
>
> Best,
> AC

---

### Comment · Area_Chair_R5ce · 2025-08-04
**Author-Reviewer Discussion Period Ending Soon**

Dear Reviewers,

The Author–Reviewer discussion period has begun and will end on August 6. Please read the rebuttal and let us know whether it satisfactorily addresses your concerns. If not, could you specify what remains inadequate? Your response will help us evaluate the paper and assist the authors in improving their work.

Please avoid responding at the last minute, as the authors may not have sufficient time to clarify your concerns.

Thank you!

Best,
AC

---

### Decision · Program_Chairs · 2025-09-17

**Decision:**

Accept (spotlight)

**Comment:**

This paper studies Hamiltonian flow approaches to optimization, rather than to sampling, where the approach has been more popular. In particular, the paper shows that a randomized Hamiltonian flow achieves an accelerated rate in expected optimization error for both the strongly convex and weakly convex cases. It then shows that a discretized version also achieves accelerated rates with an additional smoothness assumption, which is standard in convex optimization.

The approach is interesting, given the surprising fact that very few existing works study Hamiltonian flow for optimization. Among these, this paper appears to be the most satisfactory, providing new instances of accelerated first-order convex optimization algorithms (albeit in terms of expected optimization error). Moreover, Appendix B establishes a correspondence between sampling and optimization. I believe this work makes a fundamental contribution to convex optimization theory.

All reviewers are positive toward this paper. Reviewer QLTJ gave the lowest score of 4, noting that the proposed algorithm is slower than AGD in practice. Nevertheless, I believe the theoretical contribution is already sufficient, and the reviewer also explicitly stated that they agreed with accepting this paper. Reviewer KAWd was initially confused by the correspondence between optimization and sampling, but raised their score after the confusion was resolved during the author–reviewer discussion period.

Given the above, I recommend accepting this paper as a spotlight. I would not mind if my recommendation were bumped up.